# Minimum Description Length and Generalization Guarantees for Representation Learning

**Milad Sefidgaran** [†], **Abdellatif Zaidi** [† †], **Piotr Krasnowski** [†]

[†] Paris Research Center, Huawei Technologies France     [†] Université Gustave Eiffel, France

{milad.sefidgaran2,piotr.g.krasnowski}@huawei.com, abdellatif.zaidi@univ-eiffel.fr

## Abstract

A major challenge in designing efficient statistical supervised learning algorithms is finding representations that perform well not only on available training samples but also on unseen data. While the study of representation learning has spurred much interest, most existing such approaches are heuristic; and very little is known about theoretical generalization guarantees. For example, the information bottleneck method seeks a good generalization by finding a minimal description of the input that is maximally informative about the label variable, where minimality and informativeness are both measured by Shannon's mutual information.

In this paper, we establish a compressibility framework that allows us to derive upper bounds on the generalization error of a representation learning algorithm in terms of the "Minimum Description Length" (MDL) of the labels or the latent variables (representations). Rather than the mutual information between the encoder's input and the representation, which is often believed to reflect the algorithm's generalization capability in the related literature but in fact, falls short of doing so, our new bounds involve the "multi-letter" relative entropy between the distribution of the representations (or labels) of the training and test sets and a fixed prior. In particular, these new bounds reflect the structure of the encoder and are not vacuous for deterministic algorithms. Our compressibility approach, which is information-theoretic in nature, builds upon that of Blum-Langford for PAC-MDL bounds and introduces two essential ingredients: block-coding and lossy-compression. The latter allows our approach to subsume the so-called *geometrical compressibility* as a special case. To the best knowledge of the authors, the established generalization bounds are the first of their kind for Information Bottleneck type encoders and representation learning. Finally, we partly exploit the theoretical results by introducing a new *data-dependent* prior. Numerical simulations illustrate the advantages of well-chosen such priors over classical priors used in IB.

## 1 Introduction

A key performance indicator of stochastic learning algorithms is their capability to generalize, i.e., perform equally well on training and unseen data. However, designing learning algorithms with good generalization guarantees remains a major challenge. A popular approach involves learning an encoder part and a decoder part. The encoder aims at generating a suitable representation of the input (referred to as "latent variable") by extracting relevant features from the input data. The decoder aims at optimizing the performance of the learning task for the given training dataset, known as empirical risk minimization (ERM), based only on the learned latent variables. This approach is grounded on the idea that performing ERM on the latent variable (instead of the input itself) prevents overfitting.

**Information Bottleneck.** Several approaches have attempted to formalize the concept of a "good representation" [SST10, AFDM17, VDOV+17, DKSV20]. Perhaps, the most prominent is the *information bottleneck (IB)* method which was first introduced in [TPB00] and then extended in several directions [SST10, AFDM17, AZ19, KTW19, Fis20, RGTS20, KASK22]. The IB approach was deemed useful to analyze deep neural networks [SZT17, ZEAS20, GP20, Gei21] and guide their training process. For

instance, in supervised learning tasks the IB seeks representations that capture "minimum" information about the input data (shoots for good generalization power) while providing "maximum" information about the label (shoots for small empirical risk), where the amounts of captured and provided information are measured using Shannon mutual information. More precisely, denoting by $Y$ the label variable and by $X$ the input data, the IB latent variable $U$ is chosen so as to maximize the mutual information $I(U; Y)$ while minimizing the mutual information $I(U; X)$. Equivalently, this can be formulated as maximizing the Lagrange cost

$$I(U; Y) - \beta I(U; X),$$

where $\beta \geqslant 0$ designates a Lagrange multiplier. Let $(X^m, U^m)$ denote a vector, or block, of $m$ independent and identically distributed (i.i.d.) realizations $(X_i, U_i), i = 1, \dots, m$, of discrete variables $(X, U) \sim P_{X,U}$. Fix a distribution $P_{\hat{U}}$ such $P_{X,\hat{U}} = P_{X,U}$. Essentially by means of the rate-distortion theoretic *covering lemma* [CT06], it is easy to see that one can get a suitable description $\hat{U}$ of $X$ using only $I(U; X)$ bits per symbol. Precisely, generating a codebook of roughly $l_m \approx 2^{mI(U;X)}$ vectors $\hat{u}^m[j] \in \mathcal{U}^m, j = 1, \dots, m$, all drawn i.i.d. according to $P_U$, the covering lemma states that for large $m$ there exists at least one index $j$ for which the empirical distribution of $(X^m, \hat{u}^m[j])$ is arbitrary close to $P_{X,U}$ in some suitable sense. That is, the empirical distributions of $(X^m, U^m)$ and $(X^m, \hat{u}^m[j])$ are close. Hence, an "equivalent" version $\hat{U}^m$ of $U^m$ can be described with roughly $mI(U; X)$ bits. Intuitively, this makes a connection between the mutual information $I(U; X)$ and the concept of *Minimal Description Length* (MDL) [Ris78, GMP05] when one considers MDL of the latent variables, not that of model parameters [VPV18, ZEAS20]. Moreover, it is now relatively well known that there exists a connection between the generalization error of a learning model and the MDL of the parameters of that model, see, e.g., [BEHW87, BO18, GR19]. A notable work [BL03] has considered MDL of the predicted labels of a super-sample of training and test data.

**Critics to IB.** The aforementioned connections perhaps formed a belief that $I(U; X)$ is closely related to the generalization performance of representation learning algorithms. This belief, however, is controversial and conflicting evidence has been reported [GK19]. For instance, using the term $I(U; X)$ as a regularizer has been criticized for four main reasons [KTVK18, RG19, AG19, DKSV20]: **i.** The few existing upper bounds on the generalization error which involve (among other terms) the mutual information $I(U; X)$ reported in [VPV18] and the very recent and concurrent work [KDJH23] are not convincing. For example, the bound of [VPV18] holds only when the alphabets of the input and latent spaces are finite; and, in that case, it states that for any $\delta > 0$, with probability $1 - \delta$ it holds that: $\text{gen}(S, W) \leqslant \mathcal{O}\left(\frac{\log(n)}{n}\right)\sqrt{I(U; X)} + C_\delta$, where $\text{gen}(S, W)$ is the generalization error of model $W$, $C_\delta := \mathcal{O}\big(|\mathcal{U}|/\sqrt{n}\big)$, $|\mathcal{U}|$ is the size of the latent space and $n$ is the size of the training dataset. For reasonable setups, however, the term $C_\delta$ dominates and their bound becomes vacuous [RG19, LLS$^+$23]. The bound reported in [KDJH23] suffers similar shortcomings – For further details on this, see Appendix B.2. **ii.** Experimental evidence shows dependence of the generalization error on the so-called *geometrical compression* rather than on $I(U; X)$ [GK19]. Geometrical compression occurs when the latent variables are concentrated around a limited number of clusters. See [GK19, Fig. 2] for a visual representation. **iii.** $I(U; X)$ is invariant to bijection; and, as such, it does not favor learning algorithms/representations with simple decision boundaries and does not reflect the "structure" or "simplicity" of the encoder/decoder. Please refer to [AG19, Section 4.3] for a detailed discussion and examples. **iv.** Finally, for deterministic algorithms $I(U; X)$ can take large or even infinite values, especially for continuous variables or high-dimensional data, hence limiting the usefulness of IB [KTVK18, AG19].

**MDL/Compressibility.** Several works have studied MDL/compressibility to establish generalization bounds. In this context, key is the "Occam's razor" principle [LW86, BEHW87, BL03] which, e.g., for binary classification tasks, states that if the labels of the training data $S = \{Z_1, \dots, Z_n\}$, $Z_i = (X_i, Y_i)$ predicted by a model $W$ can be described by $k \ll n$ number of bits, then the learned model $W$ has a good generalization performance. There exist three closely related lines of work that use different means to describe the dataset labels: **i.** The first describes the predicted labels via the hypothesis (parameter models). It includes the works of [BEHW87, AGNZ18, SAM$^+$20, HJTW21, BSE$^+$21, SGRS22]. Recently, it was shown in [SGRS22, SZ23] that information-theoretic bounds [RZ16, XR17, SZ20], PAC-Bayes bounds [McA98] and intrinsic dimension-based approaches [ŞSDE20] also fall into this category. The proof uses so-called "fixed-size" [SGRS22] and "variable-size" [SZ23] compressibility frameworks introduced therein. **ii.** The second formulates the minimal description of labels as follows: if a learning algorithm can predict all labels of the training data $S$ by using only samples from a small subset of $S$, then the learning algorithm is guaranteed to generalize well. This second approach was initiated by [LW86] and followed by others including [HK19, HKS19, BHMZ20, HK21, HKSW20, CK22]. **iii.** The third, initiated by [BL03], deals directly with the compression of the predicted labels. This approach is shown to be related to the previous two lines, and also to PAC-Bayes and VC-dimension-based results. Moreover, it is the closest to the problem

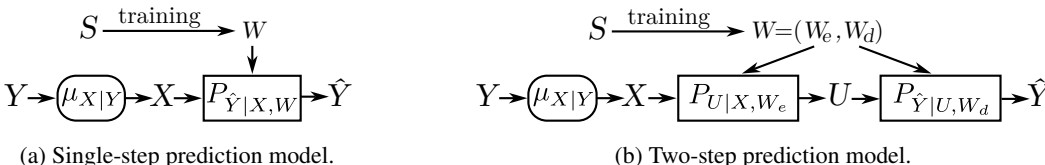

(a) Single-step prediction model.

(b) Two-step prediction model.

Figure 1: Considered learning frameworks.

of establishing bounds on the generalization error of representation learning algorithms in terms of the compressibility of the latent variables. This approach is detailed in Appendix B.1.

In this work we aim at understanding the theory of representation learning through a rigorous investigation of the connection between the MDL (or compressibility) of the latent variable and the generalization performance of representation learning algorithms. In doing so, we first study the single-step prediction model of Fig. 1a; and, then, we leverage the developed tools to study the two-step (encoder-decoder) prediction model of Fig. 1b. We also examine the claimed relationship between MDL and $I(U; X)$.

**Contributions.** Specifically, the main contributions of this work are as follows.

- For the prediction model of Fig. 1a, inspired by [BL03] we establish an *information-theoretic* framework that allows us to measure the compressibility (MDL) of the predicted labels in terms of a new object which is the KL-divergence of a vector of labels and any arbitrary symmetric prior. For a proper choice of prior, this new measure reduces to the sample-wise mutual information and hence inherits the associated properties. However, unlike the sample-wise mutual information, it can also reflect the structure/simplicity of the learning algorithm. Furthermore, by extending the framework to "lossy compressibility", this new measure does not become vacuous when one considers continuous variables instead of discrete labels. Moreover, it subsumes geometrical compressibility as a special case.

- We also establish both in-expectation and tail generalization bounds in terms of this compressibility measure of the predicted labels. In a simple case, the in-expectation bound can be cast into the form

$$\sqrt{\frac{2 \times \text{MDL(Predicted Labels)}}{n}}.$$

  The tail bound involves a similar expression. This generalization bound easily recovers the VC-dimension bound as a special case; and, hence, it also shows how the introduced compressibility measure depends on the complexity of the hypothesis class.

- Our results make a connection between the compressibility framework of [BL03] and the functional conditional mutual information (f-CMI) of [HRVSG21, HD22], which itself is an extension of the CMI-framework developed by [SZ20]. In particular, this connection shows how the f-CMI framework can be leveraged to study the compressibility of the predicted labels.

- For the encoder-decoder prediction model of Fig. 1b, the framework is extended non-trivially to the case in which instead of the compressibility of the predicted labels it is the compressibility of the latent variable which is considered. The results for this prediction model, which are our *main results* in this paper as given in Section 3, are generalization bounds for the representation learning algorithms in terms of the complexity of the latent variable. The established in-expectation and tail bounds hold for *any* decoder; and, for the $K$-class classification task for example, take the form

$$2\sqrt{\frac{2 \times \text{MDL(Latent Variables)} + K + 2}{n}}.$$

  These bounds appear to be the first of their kind for the studied representation learning setup. In part, their utility lies in that: i) they reflect the structure of the encoder class, ii) they do not become vacuous for deterministic encoders with continuous latent space and iii) they can explain the geometrical compressibility phenomenon. Thus, our framework reveals that the "joint" MDL of the latent space is more related to the encoder structure, rather than the mutual information $I(U; X)$.

- Finally, our results suggest that *data-dependent* priors can be used in place of the data-independent prior of the popular variational IB (VIB) method. We conduct experiments that illustrate the advantage brought up by proper choices of *data-dependent* priors over VIB.

Our results also open up several future directions as discussed in Appendix C.4.

**Notation.** Random variables, their realizations, and their alphabets are denoted respectively by upper-case letters, lower-case letters, and Calligraphy fonts, *e.g.,* $X$, $x$, and $\mathcal{X}$. Their distributions and expectations are denoted by $P_X$ and $\mathbb{E}[X]$. For ease of presentation, when $\mathcal{X}$ is a discrete set, $P_X$ is a *probability mass function*. Otherwise, $P_X$ is a *probability density function*. A collection of $n$ random variables $(X_1, \ldots, X_n)$ is denoted by $X^n$ or $\mathbf{X}$. We use the notation $\{x_i\}_{i=1}^m$ to denote a sequence of $m$ real or natural numbers. The set $\{1, \ldots, K\}$, $K \in \mathbb{N}$, is denoted by $[K]$. Finally, $\mathbb{R}^+$ denotes the set of non-negative real numbers. Our results are expressed in terms of information-theoretic functions. For two distributions $P$ and $Q$, the Kullback–Leibler (KL) divergence is defined as $D_{KL}(P\|Q) := \mathbb{E}_P[\log(P/Q)]$ if $P \ll Q$, and $\infty$ otherwise. The mutual information between two random variables $X, Y \sim P_{X,Y}$ with marginals $P_X$ and $P_Y$ is defined as $I(X; Y) := D_{KL}(P_{X,Y}\|P_X P_Y)$. The reader is referred to [CT06, CK11] for further information.

Our results in this paper will be expressed in terms of *symmetric* conditional priors. The following definition formalizes three types of symmetry.

**Definition 1** (Symmetric Priors). *For $U, V \sim P_{U,V}$, let $(U^{2n}, V^{2n})$ be vectors composed of $2n$ i.i.d. instances $(U_i, V_i) \sim P_{U,V}$, $i \in [2n]$. For any permutation $\pi \colon [2n] \to [2n]$, denote $U_\pi^{2n} := (U_{\pi(1)}, \ldots, U_{\pi(2n)})$ and $V_\pi^{2n} := (V_{\pi(1)}, \ldots, V_{\pi(2n)})$.*

- ***Type-I symmetry:*** *Define type-I permutations as the set of permutations $\pi \colon [2n] \to [2n]$ having the property that for any $i \in [n]$, the sets $\{\pi(i), \pi(i + n)\}$ and $\{i, i + n\}$ are equal. The conditional prior $\mathbf{Q}(U^{2n}|V^{2n})$ has type-I symmetry if $\mathbf{Q}(U_\pi^{2n}|V_\pi^{2n})$ is invariant to arbitrary type-I permutations. This definition first appeared in [Aud04] (called "almost exchangeable prior" therein) and was used for the CMI framework in [GSZ21].*

- ***Type-II symmetry:*** *The conditional prior $\mathbf{Q}(U^{2n}|V^{2n})$ has type-II symmetry if $\mathbf{Q}(U_\pi^{2n}|V_\pi^{2n})$ is invariant to any arbitrary permutations $\pi \colon [2n] \to [2n]$.*

- ***Type-III symmetry:*** *For a random variable $Z \sim P_{Z|U^{2n}, V^{2n}}$, the conditional prior $\mathbf{Q}(U^{2n}|V^{2n}, Z)$ has type-III symmetry if $\mathbf{Q}(U_\pi^{2n}|V^{2n}, Z)$ is invariant to any permutation $\pi \colon [2n] \to [2n]$ having the property that $V_i = V_{\pi(i)}$ for every $i \in [2n]$.*

*Throughout, if the underlying $(U^{2n}, V^{2n})$ is clear from the context, for ease of the notation the corresponding sets of Type-I and Type-II priors will be denoted simply as $\mathcal{Q}_i$ and $\mathcal{Q}_{ii}$ respectively.*

**Problem setup.** Unless indicated otherwise, we consider the $K$-class classification setup. Let $Z = (X, Y)$ be some *input data* taking value over the *input space* $\mathcal{Z} = \mathcal{X} \times \mathcal{Y}$ according to an unknown distribution $\mu$. We call $X \in \mathcal{X}$ the *features* of the data, and $Y \in \mathcal{Y}$ its *label*, where $\mathcal{Y} = [K]$. We assume a *training dataset* $S = \{Z_1, \ldots, Z_n\} \sim \mu^{\otimes n} =: P_S$, composed of $n$ i.i.d. samples $Z_i = (X_i, Y_i)$ of the input data, is available. We denote the features and labels of $S$ by $\mathbf{X} := X^n \sim \mu_X^{\otimes n}$ and $\mathbf{Y} := Y^n \sim \mu_Y^{\otimes n}$, respectively. We often use also a *ghost* or *test* dataset $S' = \{Z_1', \ldots, Z_n'\} \sim \mu^{\otimes n} =: P_{S'}$, where $Z_i' = (X_i', Y_i')$. Similarly, we denote the features and labels of $S'$ by $\mathbf{X}' := X'^n \sim \mu_X^{\otimes n}$ and $\mathbf{Y}' := Y'^n \sim \mu_Y^{\otimes n}$, respectively.

## 2 Generalization bounds in terms of predicted label complexity

In this section, we formulate a compressibility framework that allows us to derive upper bounds on the generalization error of a representation learning algorithm. Our proposed framework can be seen as a suitable generalization of the framework of Blum and Langford [BL03], in which the generalization encompasses *lossy compression* of the predicted labels and which exploits *block-coding*. However, unlike in the original framework based on a (compression) game between two agents Alice and Bob (see Appendix B.1 for details), this work adopts a rate-distortion theoretic perspective.

For ease of exposition, the results are presented for classification problems and the 0-1 loss function, but they can be extended trivially to any continuous $\mathcal{Y}$ and bounded loss function. Consider the setup in Fig. 1a. Let $\mathcal{A} \colon \mathcal{Z}^n \to \mathcal{W}$ be a possibly stochastic learning algorithm. That is, for a given $S = (Z_1, \ldots, Z_n) \in \mathcal{Z}^n$ the algorithm picks a hypothesis or model $W = \mathcal{A}(S) \in \mathcal{W}$. Also, let the induced joint distribution over $S \times \mathcal{W}$ be denoted as $P_{S,W}$; and the induced conditional distribution over $\mathcal{W}$ given $S$ be denoted as $P_{W|S}$. For every input data $z = (x, y) \in \mathcal{Z}$, every choice of hypothesis $w \in \mathcal{W}$ induces a conditional distribution $P_{\hat{Y}|X,W}(\hat{Y}|x, w)$ on $\hat{\mathcal{Y}} = \mathcal{Y}$. For convenience, we use the following handy notations:

$$P_{\hat{Y}|X,W}^{\otimes n}(\hat{\mathbf{y}}|\mathbf{x}, w) =: \prod_{i \in [n]} P_{\hat{Y}|X,W}(\hat{y}_i|x_i, w),$$

$$P_{\hat{Y}|X,W}^{\otimes n}(\hat{\mathbf{y}}'|\mathbf{x}', w) =: \prod_{i \in [n]} P_{\hat{Y}|X,W}(\hat{y}_i'|x_i', w),$$

$$P_{\hat{Y}|X,W}^{\otimes 2n}(\hat{\mathbf{y}}, \hat{\mathbf{y}}'|\mathbf{x}, \mathbf{x}', w) =: \prod_{i \in [n]} \left( P_{\hat{Y}|X,W}(\hat{y}_i|x_i, w) P_{\hat{Y}|X,W}(\hat{y}_i'|x_i', w) \right).$$

The quality of the prediction is measured by the loss function $\ell\colon \mathcal{Z} \times \mathcal{W} \to [0,1]$ given by

$$\ell(z,w) := \mathbb{E}_{\hat{Y} \sim P_{\hat{Y}|X,W}(\hat{Y}|x,w)}[\mathbb{1}_{\{y \neq \hat{Y}\}}], \tag{1}$$

where $\mathbb{1}$ stands for the indicator function. The associated empirical and population risks for this loss are defined as $\hat{\mathcal{L}}(s,w) := \frac{1}{n}\sum_{i \in [n]} \ell(z_i, w)$ and $\mathcal{L}(w) := \mathbb{E}_{Z \sim \mu}[\ell(Z,w)]$, respectively. Finally, the generalization error is defined as $\mathrm{gen}(s,w) := \mathcal{L}(w) - \hat{\mathcal{L}}(s,w)$.

## 2.1 Compressibility framework

Now, we introduce briefly the joint compression of a *block* of the predicted labels. Further details can be found in Appendix C.1. Consider $m$ i.i.d. pairs of train and test datasets $S_j := (Z_{j,1}, \ldots, Z_{j,n})$ and $S'_j := (Z'_{j,1}, \ldots, Z'_{j,n})$, where $Z_{j,i} = (X_{j,i}, Y_{j,i})$ and $Z'_{j,i} = (X'_{j,i}, Y'_{j,i})$. Let $S^m = (S_1, \ldots, S_m)$, $S'^m = (S'_1, \ldots, S'_m)$ and $W^m = (W_1, \ldots, W_m)$, where $W_j \sim P_{W_j|S_j}$. Denote the predicted labels using model $W_j$ for inputs $X_{j,i}$ and $X'_{j,i}$ as $\hat{Y}_{j,i}$ and $\hat{Y}'_{j,i}$, for $j \in [m]$ and $i \in [n]$. Moreover, let $\mathfrak{Z}_j^{2n} \in \mathcal{Z}^{2n}$ denote a rearrangement of the elements of $(S_j, S'_j)$ in a way that makes it indistinguishable whether a given sample $z$ is from $S_j$ or $S'_j$. In the following two subsections, we investigate two such rearrangements of a given $(S, S')$ as $\mathfrak{Z}^{2n}$. Then, for the collection $(S^m, S'^m)$, each pair $(S_j, S'_j)$ for $j \in [m]$ is rearranged independently and the matrix of all rearrangements is denoted by $\mathfrak{Z}^{2mn}$. Finally, given some $\mathfrak{Z}^{2mn}$, let denote the rearranged versions of $(Y^{mn}, Y'^{mn})$ and $(\hat{Y}^{mn}, \hat{Y}'^{mn})$ respectively by $\mathfrak{Y}^{2mn}$ and $\hat{\mathfrak{Y}}^{2mn}$.

Our approach is based on studying the *compressibility* of the rearranged model-predicted labels $\hat{\mathfrak{Y}}^{2mn}$, from an information-theoretic point of view. The rationale is as follows: since the (rearranged) predicted-labels vector $\hat{\mathfrak{Y}}^{2mn}$ agrees mostly with the true labels $\mathfrak{Y}^{2mn}$ on the dataset $S^m$, then in accordance with "Occam's Razor" theorem [LW86, BEHW87] the model $W$ is guaranteed to generalize well if $\hat{\mathfrak{Y}}^{2mn}$ can be described using only a few bits (or nats). We leverage source coding arguments to measure the *compressibility* of $\hat{\mathfrak{Y}}^{2mn}$. In our block-coding rate-distortion theoretic framework, a compression rate $R \in \mathbb{R}^+$ is said to be *achievable* if there exists a compression codebook of size $\approx e^{mR}$ (fixed a priori) which *covers* the space spanned by the model-predicted labels $\hat{\mathfrak{Y}}^{2mn}$ with high probability. That is, if $R$ is achievable then $\hat{\mathfrak{Y}}^{2mn}$ can be described using $R \in \mathbb{R}^+$ nats. Formally, $R$ is achievable if there exists a sequence of label books $\{\hat{\mathcal{Y}}_m\}_{m \in \mathbb{N}}$, with $\hat{\mathcal{Y}}_m := \{\hat{\mathbf{y}}[r], r \in [l_m]\} \subseteq \mathcal{Y}^{2mn}$, $l_m \in \mathbb{N}$, $\hat{\mathbf{y}}[r] = (\hat{\mathbf{y}}_1[r], \ldots, \hat{\mathbf{y}}_m[r])$ and $\hat{\mathbf{y}}_j[r] = (\hat{y}_{j,1}[r], \ldots, \hat{y}_{j,2n}[r]) \in \mathcal{Y}^{2n}$, such that: **i.** $l_m \leqslant e^{mR}$ and **ii.** there exist a sequence $\{\delta_m\}_{m \in \mathbb{N}}$ for which $\lim_{m \to \infty} \delta_m = 0$ such that with probability at least $(1 - \delta_m)$ over the choices of $S^m, S'^m$ one can find at least one index $r \in [l_m]$ whose associated $\hat{\mathbf{y}}[r]$ equals $\hat{\mathfrak{Y}}^{2mn}$.

As explained further in Appendix C.1.3, this *fixed-size* compressibility framework, which is suitable to upper bound the expectation of the generalization error, can be extended to *variable-size* compressibility in a way that is similar to [SZ23] in order to upper bound the generalization error with high probability. We notice that, in essence, the core idea of the compressibility framework of Blum-Langford, which we recall in Appendix B.1, can be *seen* as a *one-shot* counterpart of our approach. As discussed in [SGRS22, SZ23], in the one-shot case one deals with the "worst case" scenario; and this results in bounds that involve combinatorial terms [BL03]. In contrast, our information-theoretic framework allows us to bound the compression rate $R$ in terms of a simpler new quantity: the relative entropy of the joint conditional $P(\hat{Y}^n, \hat{Y}'^n | Y^n, Y'^n)$ and a (symmetric) conditional prior $\mathbf{Q}$ over $\hat{Y}^{2n}$ given $Y^{2n}$.

Our approach to establishing a bound on the compressibility of $\hat{\mathfrak{Y}}^{2mn}$ in terms of information-theoretic measures starting from the combinatorial approach of [BL03] and by using block-coding essentially consists in a suitable combination of the following key proof steps: *(i)* introduce and use the symmetries of Definition 1 to rearrange $(S^m, S'^m)$, *(ii)* generate the codewords using symmetric priors and *(iii)* analyze the minimum codebook size needed to "cover" reliably $\hat{\mathfrak{Y}}^{2mn}$, essentially by use of the *covering lemma*. As shown in [SGRS22, SZ23] and also our proofs, the last two ingredients can be merged via the Donsker-Varadhan's variational representation lemma. The application of this lemma is reminiscent of information-theoretic works on the generalization error such as [RZ16, XR17, SZ20].

## 2.2 Generalization bounds using type-I symmetric priors

One way to rearrange indistinguishably $(S, S')$ as $\mathfrak{Z}^{2n}$ is as follows: Let $\mathbf{J} = (J_1, \ldots, J_n)$ be a vector of $n$ i.i.d. Bernoulli$(\frac{1}{2})$ random variables $J_i \in \{i, i+n\}$, $i \in [n]$. For every $i \in [n]$, let the random variable $J_i^c \in \{i, i+n\}$ be defined such that $J_i^c = i+n$ if $J_i = i$ and $J_i^c = i$ if $J_i = i+n$. Also, let the vector $\mathbf{J}^c = (J_1^c, \ldots, J_n^c)$. For $i \in [n]$, we let the random variables $\mathfrak{Z}_{J_i}$ and $\mathfrak{Z}_{J_i^c}$ de defined as $\mathfrak{Z}_{J_i} = Z_i$ and

$\mathfrak{Z}_{J_i^c} = Z_i'$. Observe that the vector $\mathfrak{Z}^{2n} = (\mathfrak{Z}_1, \ldots, \mathfrak{Z}_{2n})$ is a $\mathbf{J}$-dependent random re-arrangement of the samples of the training and ghost datasets $S$ and $S'$. Without knowledge of $\mathbf{J}$ every element of the vector $\mathfrak{Z}^{2n}$ has equal likelihood to be from $S$ or $S'$. A similar construction was used in the context of the analysis of Rademacher complexity and the CMI of [SZ20]. We use Type-I symmetric priors, block coding and information-theoretic *covering* arguments in order to analyse the space spanned by the random vector $\mathfrak{Z}^{2n}$. This yields the generalization bound stated in the next theorem whose proof is deferred to Appendix E.2.

**Theorem 1.** *i. Let $\mathcal{Q}_i$ be the set of type-I symmetric conditional priors on $(\hat{\mathbf{Y}}, \hat{\mathbf{Y}}')$ given $(\mathbf{Y}, \mathbf{Y}')$. Then,* $\mathbb{E}_{S,W}[\mathrm{gen}(S, W)] \leqslant \sqrt{2R/n}$, *with*

$$R \leqslant \inf_{\mathbf{Q} \in \mathcal{Q}_i} \mathbb{E}_{\mathbf{Y},\mathbf{Y}'}\Big[ D_{KL}\Big( \mathbb{E}_{\mathbf{X}',\mathbf{X},W}\big[ P_{\hat{Y}|X,W}^{\otimes 2n}(\hat{\mathbf{Y}}, \hat{\mathbf{Y}}'|\mathbf{X}, \mathbf{X}', W) \big] \Big\| \mathbf{Q} \Big) \Big] = I\Big( \mathbf{J}; \hat{Y}^{2n} | Y^{2n} \Big) \quad (2)$$

*where $\mathbf{Y}, \mathbf{Y}' \sim \mu_Y^{\otimes 2n}$ and $\mathbf{X}', \mathbf{X}, W \sim P_{\mathbf{X}'|\mathbf{Y}'} P_{\mathbf{X},W|\mathbf{Y}}$. Also the mutual information is calculated with respect to the joint distribution $P_{\mathbf{J}, \hat{Y}^{2n}, Y^{2n}} = Bern(1/2)^{\otimes n} \mu_Y^{\otimes 2n} P_{\hat{Y}_{\mathbf{J}}^{2n}, \hat{Y}_{\mathbf{J}^c}^{2n} | Y_{\mathbf{J}}^{2n} Y_{\mathbf{J}^c}^{2n}}$ and the latter term is defined as $\mathbb{E}_{\mathbf{X}',\mathbf{X},W}\big[ P_{\hat{Y}|X,W}^{\otimes 2n}(\hat{Y}_{\mathbf{J}}^{2n}, \hat{Y}_{\mathbf{J}^c}^{2n}|\mathbf{X}, \mathbf{X}', W) \big]$ in which $\mathbf{X}', \mathbf{X}, W \sim P_{\mathbf{X}'|Y_{\mathbf{J}^c}^{2n}} P_{\mathbf{X},W|Y_{\mathbf{J}}^{2n}}$.*

*ii. For any $\delta \in \mathbb{R}^+$ and any conditional type-I symmetric prior $\mathbf{Q}$ on $(\hat{\mathbf{Y}}, \hat{\mathbf{Y}}')$ given $(\mathbf{X}, \mathbf{Y}, \mathbf{X}', \mathbf{Y}')$, with probability at least $(1 - \delta)$ over choices of $S, S', W \sim P_{S'} P_{S,W}$, it holds that[1]*

$$\hat{\mathcal{L}}(S', W) - \hat{\mathcal{L}}(S, W) \leqslant \sqrt{\frac{4}{2n - 1}\left( D_{KL}\Big( P_{\hat{Y}|X,W}^{\otimes 2n}(\hat{\mathbf{Y}}, \hat{\mathbf{Y}}'|\mathbf{X}, \mathbf{X}', W) \Big\| \mathbf{Q} \Big) + \log(\sqrt{2n}/\delta) \right)}.$$

A couple of remarks are in order. The part i of the result of Theorem 1 can be understood as being some form of the f-CMI of [HRVSG21] in which the function $f$ represents the predicted labels – in fact our result is slightly stronger comparatively, since instead of conditioning on $Z^{2n}$ as in the f-CMI of [HRVSG21] one here conditions only on $Y^{2n}$. Incidentally, the result also unveils an appealing connection between an extension of the compressibility approach of [BL03] (in this extension one needs to consider Type-I symmettric priors) and CMI and f-CMI [SZ20, HRVSG21, HD22]. However, for type-II and type-III symmetries, used in the coming sections, our framework goes beyond the CMI framework. The type-I priors have been also used in [GSZ21] to establish (fast-rate) tail bounds for the CMI framework. Due to the difference in the considered setups, their results are not directly comparable with ours. We also note that the above result (and some of the results in the rest of the paper) is kept for clarity and can be trivially improved, by i) moving $\mathbb{E}_{\mathbf{Y},\mathbf{Y}'}$ outside the square root, ii) by rewriting $\mathbb{E}[\mathrm{gen}(s, w)] = \frac{1}{n} \sum_{i \in [n]} \mathbb{E}[\mathrm{gen}(z_i, w)]$, and applying the bound for each $\mathbb{E}[\mathrm{gen}(z_i, w)]$, similar to [BZV20, HRVSG21, RGBTS21], iii) by extending the results in accordance with e-CMI framework [SZ20, HD22], and iv) establishing tail bound on $\mathrm{gen}(S, W)$, by noting that with probability at least $(1-\delta)$, $\hat{\mathcal{L}}(S', W) \geqslant \mathcal{L}(W) - \sqrt{\log(1/\delta)/(2n)}$ (similar to Theorem 5).

Next, observe that for any $\mathbf{Q} := \mathbb{E}_{\mathbf{X},\mathbf{X}'|\mathbf{Y},\mathbf{Y}'}\big[ \mathbf{Q}_1(\hat{\mathbf{Y}}, \hat{\mathbf{Y}}'|\mathbf{X}, \mathbf{X}', \mathbf{Y}, \mathbf{Y}') \big]$, where $\mathbf{Q}_1(\hat{\mathbf{Y}}, \hat{\mathbf{Y}}'|\mathbf{X}, \mathbf{X}', \mathbf{Y}, \mathbf{Y}')$ is an arbitrary type-I symmetric prior, and by using the Jensen's inequality, we have

$$\text{RHS of (2)} \leqslant \mathbb{E}_{S,S',W \sim P_{S,W} P_{S'}}\Big[ D_{KL}\big( P_{\hat{Y}|X,W}^{\otimes 2n}(\hat{Y}^n, \hat{Y}'^n|X^n, X'^n, W) \| \mathbf{Q}_1 \big) \Big]. \quad (3)$$

**Relation to Mutual Information.** A particular choice of the conditional prior $\mathbf{Q}_1$ in the RHS of (3) is $Q^{\otimes 2n}$ for some prior $Q$ defined over $\mathcal{Y}$. For this special choice, the RHS of (3) is given by

$$n\mathbb{E}_{S,W}\mathbb{E}_{X \sim \hat{\mu}_{X|S}}\big[ D_{KL}\big( P_{\hat{Y}|X,W}(\hat{Y}|X, W) \| Q \big) \big] + n\mathbb{E}_{X',W \sim \mu_X P_W}\big[ D_{KL}\big( P_{\hat{Y}|X,W}(\hat{Y}'|X', W) \| Q \big) \big], \quad (4)$$

where $\hat{\mu}_{X|S}$ is the empirical distribution of $X$ in the dataset $S$. Moreover, by choosing $Q$ as the marginal distribution of $\hat{Y}$ under $P_{\hat{Y}|X,W} P_{S,W}$, it is easy to see that the first term of the sum of the RHS of (4) coincides with $n\hat{I}(X; \hat{Y})$ for that choice, where $\hat{I}(\cdot; \cdot)$ stands for the "empirical mutual information" as computed from the available samples. Thus, the contribution of this term to the bound on generalization error does not necessarily vanish as $n \to \infty$ (unless the empirical mutual information itself is small). In fact, as already observed in [GK19], there exist models which generalize well but have non-small mutual-information $\hat{I}(X; \hat{Y})$. This instantiates that mutual-information type bounds may fall short of explaining true generalization capability, an observation which was already made in [GK19]. The bound on the generalization error of our Theorem 1, which is provably tighter (see (3)), then possibly remedies this issue.

---

[1]The reader may notice the absence of a "disintegrated bound" here, as opposed to the classical single-draw PAC Bayes bound [Cat07]. This is due to the choice of the loss function, which includes an expectation term.

In what follows we further investigate the relationship of the result of our Theorem 1, which is based on KL-divergence, to VC-dimension and mutual information type bounds on the generalization error.

**Relation to VC-dimension** Suppose that the VC-dimension of the hypothesis class is $d$. Then, using the Sauer–Shelah lemma [Sau72, She72] we get that given any $\mathbf{X}$ and $\mathbf{X}'$ one can have at most $(2en/d)^d$ distinct labels. By letting the conditional prior $\mathbf{Q}_1$ be a uniform distribution over all such possible predictions, it is easily seen that the KL divergence term of our Theorem 1 is upper bounded by $d\log(2en/d)$. This means that the result of our Theorem 1 recovers and possibly improves over the VC-dimension bound.

**Structure and "simplicity" of the learning algorithm.** Let us consider a simple example. Suppose that $\mathcal{X} = [0,1] \subset \mathbb{R}$ and let $\tau \in (0,1)$ be a parameter. Then, consider the set of deterministic classifiers $w$ as follows: $P_{\hat{Y}|X,W}(\hat{y}|x,w) = 1$ if ($\hat{y} = 1$ and $x \geq \tau$) *or* ($\hat{y} = 0$ and $x < \tau$); and $P_{\hat{Y}|X,W}(\hat{y}|x,w) = 0$ otherwise. It is well known that the VC-dimension of this learning class is $d = 1$. Then, by recalling the aforementioned relation to the VC dimension, we obtain that our Theorem 1 yields a bound on the generalization error which is $\mathcal{O}(\sqrt{\log(n)/n})$. In particular, it is easy to see that this bound vanishes as $n \to \infty$. This is in sharp contrast with the mutual information term $\hat{I}(X;\hat{Y})$ which does not necessarily vanish for large $n$. For example, if the pair $(X,Y)$ is such that $Y = 1$ iff $X \geq \tau^*$ for some $\tau^* \in (0,1)$, then the ratio between RHS of (4) and $n$ converges (for large $n$) to a value which is at least $H(Y)$ (because $\hat{I}(X;\hat{Y}) \to I(X;Y)$ as $n \to \infty$ and, in this example, $I(X;Y) = H(Y)$).

The above example indicates that using mutual information as a regularizer for learning algorithms (as is the case in IB) may fail to find models that generalize well and have a comparatively simple structure. In fact, using a different approach, it was already observed in [AG19] and [DKSV20] that using the mutual information term as a complexity measure does not reflect the "structure" of the learning algorithm and considering it as a regularizer might not favor "simple" learning algorithms. This suggests that mutual information regularizer in the IB approach may be replaced by the KL divergence term of the RHS of (3) (when the latent variables are considered instead of predictions) which does reflect the complexity of the encoder's structure. As investigated and shown in Section 3, in addition to favoring encoders of simpler structure, our approach provides theoretical guarantees of the generalization error.

### 2.2.1 Lossy compressiblity

The above approach and results can be extended to any bounded loss function and continuous variables $Y$ and $\hat{Y}$ (see the next section where continuous latent variables are studied). In the regime of continuous variables, a standard result of rate-distortion theory stipulates that any *lossless* encoding of $\hat{Y}^n$ and $\hat{Y}'^n$ may require an infinite number of bits, making the generalization bounds vacuous. This is precisely why Shannon's mutual information fails as a regularizer in deterministic learning algorithms with continuous alphabet variables. On the other hand, our approach can be easily extended to include the *lossy compression* of the labels (see Appendix C.1.2; in the same spirit as done in [SGRS22] for the hypothesis compression. The following result states a lossy in-expectation bound and a lossy tail bound is reported in Appendix A.

**Theorem 2.** *Let $\mathcal{Q}_i$ be the set of type-I symmetric conditional priors on $(\hat{\mathbf{Y}}, \hat{\mathbf{Y}}')$ given $(\mathbf{Y}, \mathbf{Y}')$. Then, for any $\epsilon \in \mathbb{R}$, $\mathbb{E}_{S,W}[\mathrm{gen}(S,W)]$ is upper bounded by*

$$\inf_{P_{\hat{W}|S}} \inf_{\mathbf{Q}\in\mathcal{Q}_i} \sqrt{\frac{1}{n}\mathbb{E}_{\mathbf{Y},\mathbf{Y}'}\left[D_{KL}\left(\mathbb{E}_{\mathbf{X}',\mathbf{X},\hat{W}}\left[P_{\hat{Y}|X,\hat{W}}^{\otimes 2n}(\hat{\mathbf{Y}},\hat{\mathbf{Y}}'|\mathbf{X},\mathbf{X}',\hat{W})\right]\Big\|\mathbf{Q}\right)\right]} + \epsilon,$$

*where $\mathbf{Y},\mathbf{Y}' \sim \mu_Y^{\otimes 2n}$, $\mathbf{X}',\mathbf{X},\hat{W} \sim P_{\mathbf{X}'|\mathbf{Y}'}P_{\mathbf{X},\hat{W}|\mathbf{Y}}$, and the first infimum is over all $P_{\hat{W}|S}$ satisfying $\mathbb{E}_{P_{S,W}P_{\hat{W}|S}}\left[\mathrm{gen}(S,W) - \mathrm{gen}(S,\hat{W})\right] \leq \epsilon$.*

A proof of this theorem follows by an easy combination of the distortion criterion with the bound on $\mathbb{E}_{S,\hat{W}}\left[\mathrm{gen}(S,\hat{W})\right]$ obtained by application of Theorem 1 to the compressed model $\hat{W}$ (not $W$). This simple trick, which is related conceptually to *lossy source coding*, prevents the KL-divergence term from taking very large (infinite) values for continuous alphabet variables. Moreover, the lossy compressibility also offers an interpretation of the *geometrical compressibility* concept that was observed to be related to the generalization performance in [GK19].

## 2.3 Generalization bounds using type-II symmetric priors

In this section, we consider another way to rearrange indistinguishably $(S, S')$ as $\mathfrak{Z}^{2n}$, which is inspired by [BL03]. Let $\mathbf{T} = \{T_1,\dots,T_n\}$, be a random set obtained by picking uniformly $n$ indices from $\{1,\dots,2n\}$, without replacement. Note that in contrast to $\mathbf{J}$ which had i.i.d. components, here the components of $\mathbf{T}$

are dependent. let $\mathbf{T}^c = \{1, \ldots, 2n\} \backslash \mathbf{T}$ be the complement of $\mathbf{T}$, having the elements $\mathbf{T}^c = \{T_1^c, \ldots, T_n^c\}$. Now, for each $i \in [n]$, let $(\mathfrak{Z}_{T_i}, \mathfrak{Z}_{T_i}^c) = (Z_i, Z_i')$. We use type-II symmetric prior to *cover* such rearranged vectors. To state the result, first we need to define the function $h_D(x; x') \colon [0,1] \times [0,1] \to [0,2]$ as follows:

$$h_D(x, x') := 2h_b\left(\frac{x + x'}{2}\right) - h_b(x) - h_b(x'), \tag{5}$$

where $h_b(x) := -x \log_2(x) - (1-x) \log_2(1-x)$. Note that $h_D(x, x')/2$ is equal to the Jensen-Shannon divergence between two binary Bernoulli distributions with parameters $x$ and $x'$. The reader is referred to Appendix A for the relation of this function with the combinatorial term appeared in [BL03]. The function $h_D(\cdot, \cdot)$ has the following interesting properties, proved in Appendix E.3.

**Lemma 1.** $\forall (x, x') \in [0,1] \times [0,1]$, *we have: **(i)*** $h_D(x, x') \geqslant (x - x')^2$, ***(ii)*** $h_D(x, 0) \geqslant x$, ***(iii)*** $h_D(x, x')$ *is increasing with respect to $x$ in the range $[x', 1]$, and **(iv)*** $h_D(x, x')$ *is convex with respect to both inputs.*

Now, we state the main result of this section.

**Theorem 3.** *Let $\mathcal{Q}_{ii}$ be the set of type-II symmetric priors on $(\hat{\mathbf{Y}}, \hat{\mathbf{Y}}')$ given $(\mathbf{Y}, \mathbf{Y}')$. Then, for $n \geqslant 10$,*

$$nh_D\Big(\mathbb{E}_W\big[\mathcal{L}(W)\big], \mathbb{E}_{S,W}\big[\hat{\mathcal{L}}(S, W)\big]\Big) \leqslant$$

$$\inf_{\mathbf{Q} \in \mathcal{Q}_{ii}} \mathbb{E}_{\mathbf{Y}, \mathbf{Y}'}\Big[D_{KL}\Big(\mathbb{E}_{\mathbf{X}', \mathbf{X}, W}\big[P_{\hat{Y}|X,W}^{\otimes 2n}(\hat{\mathbf{Y}}, \hat{\mathbf{Y}}'|\mathbf{X}, \mathbf{X}', W)\big]\Big\|\mathbf{Q}\Big)\Big] + \log(n) = I\Big(\mathbf{T}; \hat{Y}^{2n}|Y^{2n}\Big) + \log(n),$$

*where $\mathbf{Y}, \mathbf{Y}' \sim \mu_Y^{\otimes 2n}$ and $\mathbf{X}', \mathbf{X}, W \sim P_{\mathbf{X}'|\mathbf{Y}'} P_{\mathbf{X},W|\mathbf{Y}}$. Also the mutual information is calculated with respect to the joint distribution $P_{\mathbf{T}, \hat{Y}^{2n}, Y^{2n}} = P_{\mathbf{T}} \mu_Y^{\otimes 2n} P_{\hat{Y}_{\mathbf{T}}^{2n}, \hat{Y}_{\mathbf{T}^c}^{2n}|Y_{\mathbf{T}}^{2n} Y_{\mathbf{T}^c}^{2n}}$ and the latter term is defined as $\mathbb{E}_{\mathbf{X}', \mathbf{X}, W}\Big[P_{\hat{Y}|X,W}^{\otimes 2n}(\hat{Y}_{\mathbf{T}}^{2n}, \hat{Y}_{\mathbf{T}^c}^{2n}|\mathbf{X}, \mathbf{X}', W)\Big]$ in which $\mathbf{X}', \mathbf{X}, W \sim P_{\mathbf{X}'|Y_{\mathbf{T}^c}^{2n}} P_{\mathbf{X}, W|Y_{\mathbf{T}}^{2n}}$.*

The proof of Theorem 3 is deferred to Appendix E.4. Also, a similar tail bound is provided in Appendix A.

Using the part (i) of Lemma 1, it can be seen that if the value of the KL divergence term is larger than $\log(n)$ then the bound of Theorem 3 is tighter than that Theorem 1. Also, using part (ii) of Lemma 1 it is seen that if the error on the training set is zero (a setting referred to as "realizable case" in [BL03]), our Theorem 3 yields a bound on the generalization error which is $\mathcal{O}(1/n)$.

## 3 Generalization bounds in terms of latent variable complexity

The generalization bounds of the previous section are particularly useful in the following sense: if the learning algorithm is "simple" enough to produce a low "relative entropy" sequence of labels for training and test sets, then the algorithm generalizes well. However, they have a downside that they cannot be used directly as they are in the optimization, since minimizing those bounds may result in solutions with large empirical risk. In this section, we extend the results of the previous section to settings in which the processing is split into two parts: an encoder part that produces a family of representations that have the property to generalize well (developed according to the guidelines of the previous section) and a decoder part that selects among that family one representation that minimizes the empirical risk. As such the goal of the encoder is to guarantee a small generalization error and that of the decoder is to guarantee a small empirical risk. This procedure, which is similar to the Information Bottleneck method, aims at finding a good balance between generalizing well to unseen data and minimizing the risk of the training data.

Let $W = (W_e, W_d)$, where $W_e$ and $W_d$ are the hypotheses (or models) used by the encoder and the decoder, respectively, as illustrated in Fig. 1b. Also, let $U$ denote the output of the encoder, which will be referred to hereafter interchangeably as "representation" or "latent variable". The encoder produces the representation $U$ according to the conditional $P_{U|X,W_e}$. The decoder produces an estimate $\hat{Y}$ of the true label $Y$ according to the conditional $P_{\hat{Y}|U,W_d}$. We consider a stochastic learning algorithm $\mathcal{A} \colon \mathcal{Z}^n \to \mathcal{W}$ which picks a model $W = (W_e, W_d) \in \mathcal{W} := \mathcal{W}_e \times \mathcal{W}_d$ according to $P_{W|S}$. Let a loss function $\ell \colon \mathcal{Z} \times \mathcal{W} \to [0,1]$ be given. For a model $w = (w_e, w_d)$ the quality of the prediction of $\hat{Y}$ is evaluated as

$$\ell(z, w) := \mathbb{E}_{\hat{Y} \sim P_{\hat{Y}|X,W}}(\hat{Y}|x,w)\big[\mathbb{1}_{\{y \neq \hat{Y}\}}\big] := \mathbb{E}_{U \sim P_{U|X,W_e}}(U|x,w_e) \mathbb{E}_{\hat{Y} \sim P_{\hat{Y}|U,W_d}}(\hat{Y}|U,w_d)\Big[\mathbb{1}_{\{y \neq \hat{Y}\}}\Big].$$

Empirical risk, population risk, and generalization error are defined similarly to the previous section. According to the above motivation, we are interested in establishing a bound on the generalization error of the algorithm $\mathcal{A}$ that depends only on the part $W_e$ of the model $W$. In accordance with our compressibility

framework of Section 2 such bound would then depend on the complexity of the latent variable $U$. However, here, the encoder part $W_e$ and the decoder part $W_d$ are both trained using the dataset $S$. Thus, they may be statistically dependent in general and therefore the "rearrangement" ideas using type-I and type-II symmetries do not work here. To elaborate on this a bit more, recall that for example for type-I symmetry, to rearrange $(\mathbf{Y}, \hat{\mathbf{Y}})$ and $(\mathbf{Y}', \hat{\mathbf{Y}}')$ in an indistinguishable manner, we randomly shuffle the positions of the pairs $(Y_i, \hat{Y}_i)$ and $(Y'_i, \hat{Y}'_i)$. If we now consider using the type-I symmetry for the new setup, then shuffling of the pairs $(Y_i, U_i, \hat{Y}_i)$ and $(Y'_i, U'_i, \hat{Y}'_i)$, changes the dataset $S$ and thus the distributions $P_{W|S}$ and $P_{\hat{Y}|W_d,U}$. Hence, covering the resulting rearranged sequence would unavoidably depend on both encoder and decoder parts. To overcome this issue, we use type-III symmetry, where we leave the train and ghost datasets untouched and randomly "swap" only those latent variables and their corresponding predictions, *i.e.*, $(U_i, \hat{Y}_i)$ and $(U'_j, \hat{Y}'_j)$, that are associated with the train and ghost samples with the same label, *i.e.*, if $Y_i = Y'_j$. The following result is the main theorem of this section and the paper derived using the type-III symmetric priors. A formal proof of this result can be found in Appendix E.5.

**Theorem 4** (Generalization Bound for Representation Learning Algorithms). *Consider a $K$-classification learning task. Let $\mathbf{Q}$ be a type-III symmetric conditional prior over $U^{2n}$ given $X^{2n}, Y^{2n}$ and $\hat{W}_e \in \mathcal{W}_e$, namely, $\mathbf{Q}\Big((u_{\pi(1)}, \ldots, u_{\pi(2n)}) \big| (x_1, \ldots, x_{2n}), (y_1, \ldots, y_{2n}), \hat{w}_e\Big)$ remains the same for all permutations $\pi \colon [2n] \mapsto [2n]$ that preserves the label, i.e., $y_{\pi(i)} = y_i$ for $i \in [2n]$. Then,*

$$\mathbb{E}_{S,W}[\mathrm{gen}(S,W)] \leqslant \inf 2\sqrt{\frac{2\mathbb{E}_{S,S',\hat{W}_e}\bigg[D_{KL}\Big(P_{U|X,\hat{W}_e}^{\otimes 2n}(\mathbf{U}, \mathbf{U}'|\mathbf{X}, \mathbf{X}', \hat{W}_e)\big\|\mathbf{Q}\Big)\bigg] + K + 2}{n}} + \epsilon,$$

*where $S, S', \hat{W}_e \sim P_{S,\hat{W}_e} P_{S'}$ and the infimum is over all Markov kernels $P_{\hat{W}_e|S}$ such that for $\hat{W} = (\hat{W}_e, W_d)$, $\mathbb{E}_{P_{S,W} P_{\hat{W}_e|S}}\big[\mathrm{gen}(S,W) - \mathrm{gen}(S,\hat{W})\big] \leqslant \epsilon$.*

The bound of Theorem 4 on the generalization error of representation learning, and the related IB-encoder, is to the best of our knowledge the first of its kind and significance. In fact, while few works have already investigated the generalization error of IB [SST10, VPV18, KDJH23], for the special case of discrete variables, their bounds, which in part are expressed in terms of the empirical mutual information $\hat{I}(U; X)$, appear to be vacuous for most settings as already observed in [RG19, GK19, AG19, DKSV20] (see introduction and Appendix B.2 for more details). Furthermore, as also extensively discussed in [AG19, GK19, DKSV20], mutual information may not be a good indicator of the generalization error. Such aspects, including how mutual information fails to reflect "simplicity" or "structure" of the encoder, are discussed in more detail in the previous section. An alternate bound on the generalization error of representation learning appeared in [DKSV20]. However, their bound only holds for encoders that find the *optimal* representation in a sense defined therein.

Now, few remarks on the result of Theorem 4 are in order. First, note that the generalization bound of this theorem depends *only* on the encoder and more precisely on the complexity of the latent space $U$; and, so, this bound is valid for *any* choice decoder.[2] Next, similar to the bounds of the Theorems 1-5, the KL-divergence term of Theorem 4 explicitly takes into account the "structure" and "simplicity" of the encoder, and, therefore, it resolves one of the major issues of the IB method [AG19, GK19, DKSV20].

Moreover, the result of Theorem 4 suggests that the considered prior could depend on the data and model. This enables a larger class of choices for the prior. Examples include (i) Symmetric jointly Gaussian priors, (ii) priors that depend on the category (label), *i.e.,* the prior for a category $k \in [K]$ at each optimization iteration could possibly depend on some statistics of the latent variables of all training and test samples having label $k$, and (iii) priors that steer latent variables toward some pre-defined "constellations" in the latent space, depending on their label. On this aspect, note that allowing the prior to depend on labels enables a connection with the "conditional information-bottleneck" of [Fis20].

Another important property of our bound of Theorem 4 is that, unlike the empirical mutual information term of [SST10, VPV18, KDJH23],[3] it does not become infinite for deterministic encoders with continuous input-output. Moreover, our approach which is based on *lossy compression* provides an interpretation of the *geometric compression* of [GK19, GVDBG$^+$19] where latent variables are concentrated around some

---

[2]In some cases, the bound may be loose, however, e.g., with a decoder $w_d$ that produces an estimate $\hat{Y}$ *independently* of the obtained representation $U$.

[3]The lossy compression trick, however, can also be applied for results of [SST10, VPV18, KDJH23].

constellation points.[4] We hasten to mention that "lossy compression" here should not be confused with approaches that add noise after the encoder. The main difference is that in those approaches the noisy representations are passed to the decoder; while here, the "noisy" representations are used only to estimate lossy compressibility. These "noisy" representations can be achieved by either adding small "noise" to the model parameters or the latent variable. Note that by increasing the noise level the $\epsilon$ term in Theorem 4 increases while the KL-divergence term potentially decreases. In practice, a suitable trade-off between the two effects can be found by treating the amount of added noise as a *hyper-parameter* to optimize.

Finally, a similar tail bound on the generalization error has been established in Appendix A.

## 4 Experiments

In this section, we illustrate our results via some experiments. For more detail and other experiments the reader is referred to Appendix D.

Our main Theorems 4 and 7 *suggest* that for the representation learning setup of Fig. 1b the generalization error is controlled essentially by the divergence term $D_{KL}\left(P_{U|X,W_e}^{\otimes 2n}(\mathbf{U}, \mathbf{U}'|\mathbf{X}, \mathbf{X}', \mathbf{Y}, \mathbf{Y}', W_e)\|\mathbf{Q}\right)$ where $\mathbf{Q}$ is a type-III symmetric prior. In a sense, this also means that, with a proper *data-dependent* choice of the prior, the usage of the aforementioned divergence term as a regularizer possibly offers better generalization guarantees (relative, e.g., to the conventional data-independent prior of VIB). In what follows, we propose a new family of data-dependent priors which appear to better capture the "simplicity" or "structure" of the encoder. We also compare the associated accuracy with that offered by the fixed prior of VIB.

More precisely, in VIB the prior $\mathbf{Q}$ factorizes as a product of $2n$ scalar standard Gaussian priors, *i.e.,* $\mathbf{Q} = Q^{\otimes 2n}$, where $Q = \mathcal{N}(\mathbf{0}_m, \mathbf{I}_m)$, $\mathbf{0}_m \in \mathbb{R}^m$ is the zero vector and $\mathbf{I}_m$ is the $m \times m$ identity matrix. In our lossless approach, which we here coin as Lossless Category-Dependent VIB (CDVIB), the prior $\mathbf{Q}$ still factorizes as a product of $2n$ scalar Gaussian priors, *i.e.,* $\mathbf{Q} = \prod_{i\in[2n]} Q_i$, but with three major differences: **i.** Each $Q_i$ can be chosen from a set of $M \times K$ priors – $M$ priors for each label. **ii.** Unlike VIB, each of $M \times K$ priors can depend on some statistics of $(\mathbf{S}, \mathbf{S}')$ and also on the label of each sample, **iii.** Unlike VIB, where the prior is fixed, here $\mathbf{Q}$ is 'learned' during the training phase. To this end, the mean and variance of the scalar priors are updated after each training iteration using a moving average with some small coefficient, allowing the latent space to better adapt to the structure of the encoder and the data. Taking the moving average also has another role, which is to "partially" reproduce the effect of the "ghost data" (which comes from the test dataset that is usually unavailable during training). In lossy CDVIB, similar priors are considered but over "noisy" versions of the latent variables, *i.e.,* $(\hat{\mathbf{U}}, \hat{\mathbf{U}}')$. Note that while the noisy versions are considered for the regularizer, the decoder receives as input $(\mathbf{U}, \mathbf{U}')$.

We consider CIFAR10 [KH+09] image classification using a small CNN-based encoder and a linear decoder. The results shown in Fig. 2 indicate that the model trained using our priors achieves better ($\sim 2.5\%$) performance in terms of both generalization error and population risk. This *suggests* that our priors help to find a better *representation* than the standard VIB prior.

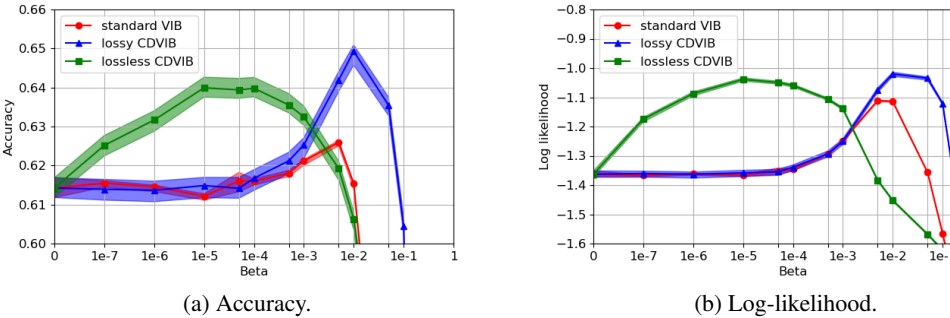

(a) Accuracy.  (b) Log-likelihood.

Figure 2: Accuracy during test phase of our two-step prediction model trained using the standard VIB prior and our "lossless" CDVIB and "lossy" CDVIB priors computed for $M = 5$. The values are averaged over 5 runs. The graphs are displayed together with 95% bootstrap confidence intervals.

---

[4]The reader is referred to Appendix C.3 for a simple example of geometric compression.

## 5 Acknowledgement

The authors would like to thank the anonymous reviewers for their many insightful comments and suggestions. In particular, by pointing out the connection between our results and the f-CMI literature.

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

# Appendices

The appendices are organized as follows.

- Appendix A contains further theoretical results. Specifically,
  - in Appendix A.1 two more tail bounds on the generalization error in terms of the predicted label complexity are presented.
  - in Appendix A.2 a tail bound on the generalization error in terms of the latent variable complexity is presented.
- We discuss some of the related works in more detail in Appendix B. In particular,
  - in Appendix B.1, we recall the compressibility framework of Blum and Langford [BL03].
  - in Appendix B.2, we discuss the results of [KDJH23].
- In Appendix C, we discuss various topics that could not be sufficiently addressed in the paper, due to lack of space. More precisely,
  - we define various notions of compressibility, including lossless, lossy, and variable-size compressibility in Appendix C.1,
  - we present an intuition on the function $h_D(\cdot, \cdot)$, used in Section 2, in Appendix C.2,
  - we give a simple example of geometrical compression, and its relation to lossy compressibility, in Appendix C.3,
  - and finally we present the conclusion of our work together with some interesting future directions in Appendix C.4.
- Appendix D contains the details of the experiments presented in Section 4.
  - Appendix D.1 recalls the standard VIB objective function,
  - Appendix D.2 defines a family of lossless objective functions using data-dependent priors,
  - Appendix D.3 defines a family of lossy objective functions using data-dependent priors,
  - Appendix D.4 details the training and testing datasets used in experiments,
  - Appendix D.5 details the model architecture used in experiments,
  - Appendix D.6 provides training details,
  - Appendix D.7 presents and discusses the numerical results.
- Appendix E contains the deferred proofs of all our theoretical results. More precisely,
  - the intuition behind our proof techniques is presented in Appendix E.1,
  - Appendix E.2 contains the proof of Theorem 1,
  - Appendix E.3 contains the proof of Lemma 1,
  - Appendix E.4 contains the proof of Theorem 3,
  - Appendix E.5 contains the proof of Theorem 4,
  - Appendix E.6 contains the proof of Theorem 5,
  - Appendix E.7 contains the proof of Theorem 6,
  - Appendix E.8 contains the proof of Theorem 7,
  - Appendix E.9 contains the proof of Lemma 3.

## A  Additional theoretical results

In this section, we present several tail bounds on the generalization error in terms of the complexity of the predicted labels or the latent variables.

### A.1  Tail bounds on the generalization error in terms of predicted label complexity

First, we start by stating a *lossy* tail bound on the generalization error in terms of the complexity of the tail bound.

**Theorem 5.** *Consider the learning framework defined above. Let* $\mathbf{Q}$ *be any fixed type-I symmetric prior on* $(\hat{\mathbf{Y}}, \hat{\mathbf{Y}}')$ *that could depend on* $(\mathbf{X}, \mathbf{Y}, \mathbf{X}', \mathbf{Y}')$*. Then for any* $\epsilon \in \mathbb{R}$ *and* $\delta \in \mathbb{R}^+$ *with probability at least* $(1 - \delta)$ *over choice of* $S$ *and* $S'$*, we have that* $\mathbb{E}_{W \sim P_{W|S}}[\mathrm{gen}(S, W)]$ *is upper bounded by*

$$\sqrt{\frac{\log(2/\delta)}{2n}} + \inf \sqrt{\frac{4}{2n-1}\left(\mathbb{E}_{\hat{W} \sim P_{\hat{W}|S}}\left[D_{KL}\left(P_{\hat{Y}|X,\hat{W}}^{\otimes 2n}(\hat{\mathbf{Y}}, \hat{\mathbf{Y}}'|\mathbf{X}, \mathbf{X}', \hat{W})\middle\|\mathbf{Q}\right)\right] + \log(\sqrt{8n}/\delta)\right)} + \epsilon,$$

*where the infimum is over all* $P_{\hat{W}|S}$ *that satisfy*

$$\mathbb{E}_{P_{W|S}P_{\hat{W}|S}}\left[\left|\left(\hat{\mathcal{L}}(S', W) - \hat{\mathcal{L}}(S, W)\right) - \left(\hat{\mathcal{L}}(S', \hat{W}) - \hat{\mathcal{L}}(S, \hat{W})\right)\right|\right] \leqslant \epsilon/2. \tag{6}$$

The theorem is proved in Appendix E.6. In the above result, it is easy to replace the expectation with respect to $P_{W|S}$ with any arbitrary expectations, as it is common in PAC-Bayes bounds and used for example in [NBS18].

Next, we state a tail bound in terms of the function $h_D(\cdot, \cdot)$, defined in (5).

**Theorem 6.** *Consider the learning framework of Section 2. Let* $\mathbf{Q}$ *be any fixed type-II symmetric prior on* $(\hat{\mathbf{Y}}, \hat{\mathbf{Y}}')$ *that could depend on* $(\mathbf{X}, \mathbf{Y}, \mathbf{X}', \mathbf{Y}')$*. Then, for any* $\delta \in \mathbb{R}^+$*, with probability at least* $(1 - \delta)$ *over choices of* $(S, S', W) \sim P_{S'}P_{S,W}$*, it holds that*

$$nh_D\left(\hat{\mathcal{L}}(S', W), \hat{\mathcal{L}}(S, W)\right) \leqslant D_{KL}\left(P_{\hat{Y}|X,W}^{\otimes 2n}(\hat{\mathbf{Y}}, \hat{\mathbf{Y}}'|\mathbf{X}, \mathbf{X}', W)\middle\|\mathbf{Q}\right) + \log(n/\delta).$$

This theorem is proved in Appendix E.7. In particular, when $\hat{\mathcal{L}}(S, W) = 0$, using Lemma 1 conclude that, this result yields that with probability at least $(1 - \delta)$,

$$\hat{\mathcal{L}}(S', W) \leqslant \frac{D_{KL}\left(P_{\hat{Y}|X,W}^{\otimes 2n}(\hat{\mathbf{Y}}, \hat{\mathbf{Y}}'|\mathbf{X}, \mathbf{X}', W)\middle\|\mathbf{Q}\right) + \log(n/\delta)}{n}.$$

### A.2 Tail bounds on the generalization error in terms of latent variable complexity

In this section, we present a tail generalization bound for the two-step learning setup of Section 3 that highlights the relevance of latent variable compressibility to generalization error.

**Theorem 7.** *Consider a* $K$*-classification learning task with the above defined framework. Let* $\mathbf{Q}$ *be a type-III symmetric conditional prior over* $U^{2n}$ *given* $X^{2n}, Y^{2n}$ *and* $W_e \in \mathcal{W}_e$*, namely,* $\mathbf{Q}\left((u_{\pi(1)}, \ldots, u_{\pi(2n)})\middle|(x_1, \ldots, x_{2n}), (y_1, \ldots, y_{2n}), w\right)$ *remains the same for all permutations* $\pi\colon [2n] \mapsto [2n]$ *that preserves the label,* i.e., $y_{\pi(i)} = y_i$ *for* $i \in [n]$*. Then, for any* $\lambda \in \mathbb{R}^+$*,*

**i.** *with probability at least* $(1 - \delta)$ *over choices of* $(S, S', W) \sim P_{S'}P_{S,W}$*,*

$$\hat{\mathcal{L}}(S', W) - \hat{\mathcal{L}}(S, W) \leqslant \frac{D_{KL}\left(P_{U|X,W_e}^{\otimes 2n}(\mathbf{U}, \mathbf{U}'|\mathbf{X}, \mathbf{X}', W_e)\middle\|\mathbf{Q}\right) + (K+2)/2 + \log(1/\delta)}{\lambda} + \frac{2\lambda}{n},$$

**ii.** *with probability at least* $(1 - \delta)$ *over choices of* $(S, S', W) \sim P_{S'}P_{S,W}$*,*

$$\mathrm{gen}(S, W) \leqslant \frac{D_{KL}\left(P_{U|X,W_e}^{\otimes 2n}(\mathbf{U}, \mathbf{U}'|\mathbf{X}, \mathbf{X}', W_e)\middle\|\mathbf{Q}\right) + (K+2)/2 + \log(2/\delta)}{\lambda} + \frac{2\lambda}{n} + \sqrt{\frac{\log(2/\delta)}{n}}.$$

Note that the second part of the theorem can be easily derived from the first part using Hoeffding's inequality. We provide the proof of the first part in Appendix E.8.

## B    Related works

In this section, we discuss some of the related works in more detail.

## B.1 PAC-MDL framework of Blum and Langford

Here, we recall the PAC-MDL framework of [BL03] which, therein, was introduced in the form of a (compression) game between two agents, Alice and Bob. Alice has access to both a labeled training set $S = (X^n, Y^n)$, consisting of $n$ labeled samples, and a test set $S' = X'^n$, consisting of $n$ unlabeled samples, all drawn independently from $\mu$.[5] Bob has available just the test set and the unlabeled version of the training set, *i.e.*, $(X^n, X'^n)$, ordered in some predefined (e.g., lexicographic) order known to both agents. Hereafter we denote the ordered vector of labeled and unlabeled samples as $\pi(X^n, X'^n)$. It is assumed that by observing the vector $\pi(X^n, X'^n)$ Bob cannot know which samples of it are from the training set and which are from the test set. The goal of Alice is to communicate the labels $\pi(Y^n, Y'^n)$ to Bob using as few bits as possible. [6] To this end, let $\mathcal{E} \colon \mathcal{Z}^n \times \mathcal{X}^n \to \{0,1\}^*$ be a mapping (encoder) used by Alice and $\sigma := \mathcal{E}(S, X'^n)$ the string transmitted to Bob. The goal of Bob is to guess the labels of both sets $X^n$ and $X'^n$; and does so by running a decoder mapping $\mathcal{D} \colon \mathcal{X}^{2n} \times \{0,1\}^* \to \mathcal{Y}^{2n}$. That is, Bob forms an estimate of the true labels as $\mathcal{D}(\pi(X^n, X'^n), \sigma)$. In this context, the *empirical risk* is measured as $\hat{\mathcal{L}}(\sigma, s, s') := \frac{1}{n} \sum_{i \in [n]} \mathbb{1}_{\{y_i \neq \hat{y}_i\}}$ and a *proxy test risk* is measured as $\mathcal{L}(\sigma, s, s') := \frac{1}{n} \sum_{i \in [n]} \mathbb{1}_{\{y'_i \neq \hat{y}'_i\}}$. Essentially, the PAC-MDL bound of [BL03] can be seen as a generalized version of Occam's-Razor theorem [LW86, BEHW87] which states that if one can explain (or encode) the labels of a set of $n$ training samples by a hypothesis that can be described using only $|\sigma| \ll n$ bits then this guarantees that this hypothesis generalizes well for unseen samples.

**Theorem 8** ([BL03, Theorem 6]). *For any prior distribution* $\mathbf{Q}(\sigma)$ *of* $\sigma$ *and any* $\delta > 0$, *with probability at least* $1 - \delta$ *over the choices of* $S, S' \sim P_S P_{S'}$, *we have:*

$$\forall \sigma \colon n\mathcal{L}(\sigma, S, S') \leqslant b_{\max}\Big(n, \hat{\mathcal{L}}(\sigma, S, S'), \mathbf{Q}(\sigma)\delta\Big), \tag{7}$$

*where*

$$b_{\max}\Big(n, \frac{a}{n}, \delta\Big) := \max\{b \colon \textit{Bucket}(n, a, b) \geqslant \delta\}, \tag{8}$$

*and*

$$\textit{Bucket}(n, a, b) := \sum_{c \in [b, a+b]} \frac{\binom{n}{c}\binom{n}{a+b-c}}{\binom{2n}{a+b}}. \tag{9}$$

The result is made more explicit for the so-called realizable case, *i.e.*, when $\hat{\mathcal{L}}(\sigma, S, S') = 0$. For this case, we have [BL03, Corollary 3]

$$\mathbb{P}\Big(\forall \sigma \colon \hat{\mathcal{L}}(\sigma, S, S') > 0 \text{ or } \mathcal{L}(\sigma, S, S') \leqslant (|\sigma| + \log_2(1/\delta))/n\Big) > 1 - \delta,$$

where $|\sigma|$ denotes the size in bits of the string $\sigma$ using some a priori fixed codewords. The result clearly shows that if the string $\sigma$ can be sent using few bits then the algorithm generalizes well. Also, the approach can be used in order to establish similar results for the VC-dimension and PAC-Bayes bounds.

We emphasize that our construction is in sharp contrast with an adaptation of the above-described approach of Blum and Langford [BL03] in which one would reveal both labeled and unlabeled samples $(S^m, X'^{mn})$ to Alice and only unlabeled samples $(X^{mn}, X'^{mn})$ to Bob (both in a predefined order). We, however, in our proposed approach in Section 2.1, reveal the vector $\mathfrak{Z}^{2mn}$, containing $(S^m, S'^m)$ in a predefined order to both. Furthermore, we emphasize that the goal in our approach is not to recover the labels $\mathfrak{Y}^{2mn}$, but the predictions $\hat{\mathfrak{Y}}^{2mn}$ as produced by the picked models or hypotheses $W^m$.

On another note, we remark that the interpretation of $R$, which appeared in the size of the codebook in our approach, *i.e.*, $|\hat{\mathcal{Y}}_m| \leqslant e^{mR}$, corresponds to the average number of bits (per dataset $S$) that is needed to send a compressed version of the string $\sigma$ of [BL03]. As such, for the fixed-size codebook explained in Section 2.1 and when $|\sigma|$ is constant, $R \leqslant |\sigma|$. Similar relations hold in general, by considering the *variable-size* codebook.

---

[5]For simplicity, we assume here that the training and test sets $S$ and $S'$ have identical sizes, i.e., $|S| = |S'|$; but all the results that will follow extend easily to the case of $|S| \neq |S'|$.

[6]Note that Alice does not know the true labels $Y'^n$ of the test samples $X'^n$ and can only estimate them as $\hat{Y}'^n$.

## B.2 On the generalization bounds of [KDJH23]

After the initial submission of our work, another work [KDJH23] appeared on arXiv, accepted at ICML 2023, that provides an upper bound on the generalization error of representation learning algorithms. As claimed, this result justifies the benefits of the information bottleneck principle, by relating IB to the generalization error. The results are provided for the multi-layer neural networks and for different choices of latent variables corresponding to the output of different layers. Here, to adapt the results to the setup of this paper, we only consider the output of the encoder layer as the latent variable and adapt correspondingly the notations of the results in [KDJH23] to our notations.

With the adapted notations, [KDJH23] claims to bound the generalization error of a representation learning algorithm "roughly by

$$\tilde{\mathcal{O}}\left(\sqrt{\frac{I(X;U|Y)+1}{n}}\right).$$"

However, firstly, since this bound is in terms of the mutual information function, the critics on the relation of mutual information and generalization error, discussed in [KTVK18, RG19, AG19, DKSV20] and provided also in the "Critics to IB" section of the introduction (Section 1), are valid for the results of [KDJH23], as well. More importantly, we could not conclude the reported order-wise behavior from [KDJH23, Theorem 2]. Using the notations of our work, their result states that with probability at least $1 - \delta$ over training data $S$, the generalization error is bounded by

$$G_3\sqrt{\frac{(I(X;U|Y)+I(W_e;S))\ln(2)+\hat{\mathcal{G}}_2}{n}} + \frac{G_1}{\sqrt{n}},$$

where the constants $G_1$, $\hat{\mathcal{G}}_2$, and $G_3$ are defined in [KDJH23, Appendix E.1] and claimed that $\hat{\mathcal{G}}_2 \sim \tilde{\mathcal{O}}(1)$ and $G_3 \sim \tilde{\mathcal{O}}(1)$, as $n \to \infty$. However, we were unable to resolve the following concerns regarding this bound.

   i. Firstly, it is not clear how $I(W_e;S)$ is considered to behave as $\tilde{\mathcal{O}}(1)$, when $n \to \infty$. Indeed, the size of dataset $S$ and the learning algorithm $P_{W_e|S}$ change as $n \to \infty$.

   ii. Secondly, by referring to the definitions of the constants in [KDJH23, Appendix E.1], it can be easily verified that $\hat{\mathcal{G}}_2 = C_1 + (H(U|X,Y) + H(W_e|S))\ln(2)$, for some non-negative constant $C_1$. Hence, the bound can be re-written as

$$G_3\sqrt{\frac{(H(U|Y)+H(W_e))\ln(2)+C_1}{n}} + \frac{G_1}{\sqrt{n}}.$$

   Thus, the bound is in terms of $H(U|Y) + H(W_e)$, and not $I(X;U|Y) + I(W_e;S)$.

   iii. Lastly, the term $G_3$, defined in [KDJH23, Appendix E.1], is composed of $T_Y \in \mathbb{N}$ elements. By [KDJH23, Lemma 2], $T_Y$ behaves roughly as $\tilde{\mathcal{O}}(2^{H(U|Y)})$. Using [KDJH23, Lemma 2], it is shown that $G_3$ is bounded. However, it *seems* that this is *only* shown by assuming that the $T_Y$ terms of $G_3$ satisfy certain conditions related to the decreasing rate of their ordered values. This assumption however, may not hold in general, and hence $G_3$ may behave as $\tilde{\mathcal{O}}(T_y) \approx \tilde{\mathcal{O}}(2^{H(U|Y)})$, which then becomes the dominant term in the bound.

# C  Further clarifications and discussions

In this section, we explain our compressibility framework in more detail. Moreover, we present some intuitions on the function $h_D(\cdot,\cdot)$ and give an example of how lossy compression is related to geometrical compression. Finally, we discuss some potential future works.

## C.1  Compressibility framework

In this section, we propose various notions of the compressibility used in Section 2.

### C.1.1  Lossless fixed-size compressibility

We start by recalling the needed elements for joint compression of a *block* of the predicted labels. Consider $m$ i.i.d. pairs of train and test datasets $S_j := (Z_{j,1}, \ldots, Z_{j,n})$ and $S'_j := (Z'_{j,1}, \ldots, Z'_{j,n})$,

where $Z_{j,i} = (X_{j,i}, Y_{j,i})$ and $Z'_{j,i} = (X'_{j,i}, Y'_{j,i})$. Let $S^m = (S_1, \ldots, S_m)$, $S'^m = (S'_1, \ldots, S'_m)$ and $W^m := (W_1, \ldots, W_m)$, where $W_j \sim P_{W_j|S_j}$. It is important to note that the model or hypothesis $W_j$ is chosen based on the dataset $S_j$ only and the introduction of the (*ghost*) dataset $S'_j$ is here only for the sake of the analysis, similarly as it was done in the derivation of the PAC-MDL bound of [BL03] or in Rademacher sample complexity or the conditional mutual information of [SZ20]. Denote the predicted labels using model $W_i$ for inputs $X_{i,j}$ and $X'_{j,i}$ as $\hat{Y}_{j,i}$ and $\hat{Y}'_{j,i}$, for $j \in [m]$ and $i \in [n]$.

Moreover, let $\mathfrak{Z}^{2mn} \in \mathcal{Z}^{2mn}$ denote a rearrangement of the elements of $(S^m, S'^m)$ such that while the entire sets $S^m$ and $S'^m$ are known by having the vector $\mathfrak{Z}^{2mn}$, but they are arranged in an order that one cannot distinguish whether a given sample $z$ is from $S^m$ or $S'^m$. Accordingly, denote the rearranged versions of $(Y^{mn}, Y'^{mn})$ and $(\hat{Y}^{mn}, \hat{Y}'^{mn})$ respectively by $\mathfrak{Y}^{2mn}$ and $\hat{\mathfrak{Y}}^{2mn}$. In Sections 2.2 and 2.3, we presented two methods for such rearrangement. As mentioned before, in both methods, the rearrangement of $(S^m, S'^m)$ as $\mathfrak{Z}^{2mn} = \{\mathfrak{Z}_{j,i}\}_{j \in [m], i \in [n]}$ is done in two steps: First, for each $j \in [m]$, we rearrange indistinguishably $(S_j, S'_j)$ as $\mathfrak{Z})j^{2n}$, and then we concatenated the $m$ resulting $\{\mathfrak{Z}_j^{2n}\}_{j \in [m]}$ to achieve $\mathfrak{Z}^{2mn}$. Now, we explain how to rearrange indistinguishably a given $(S, S')$ as $\mathfrak{Z}^{2n}$ using two methods:

- Section 2.2: Let $\mathbf{J} = (J_1, \ldots, J_n)$ be a vector of $n$ i.i.d. Bernoulli($\frac{1}{2}$) random variables $J_i \in \{i, i+n\}$, $i \in [n]$. Denote by $\mathbf{J} = (J_1^c, \ldots, J_n^c)$ the vector of the complementary choices in $\mathbf{J}$, *i.e.*, $J_i \cup J_i^c = \{i, i+n\}$. Now, for each $i \in [n]$, let $(\mathfrak{Z}_{J_i}, \mathfrak{Z}_{J_i^c})$ to be equal to $(Z_i, Z'_i)$.

- Section 2.3: Let $\mathbf{T} = \{T_1, \ldots, T_n\}$, be a random set obtained by picking uniformly $n$ indices from $\{1, \ldots, 2n\}$, without replacement. Note that in contrast to $\mathbf{J}$ which had i.i.d. components, here the components of $\mathbf{T}$ are dependent. let $\mathbf{T}^c = \{1, \ldots, 2n\} \backslash \mathbf{T}$ be the complement of $\mathbf{T}$, having the elements $\mathbf{T}^c = \{T_1^c, \ldots, T_n^c\}$. Now, for each $i \in [n]$, let $(\mathfrak{Z}_{T_i}, \mathfrak{Z}_{T_i^c}) = (Z_i, Z'_i)$.

Based on the above, hereafter we study the *compressibility* of the sorted model-predicted labels $\hat{\mathfrak{Y}}^{2mn}$, from an information-theoretic point of view. The rationale is that, in accordance with "Occam's Razor" theorem [LW86, BEHW87], since the (rearranged) predicted-labels vector $\hat{\mathfrak{Y}}^{2mn}$ agrees mostly with the true labels $\mathfrak{Y}^{2mn}$ on the dataset $S^m$, if $\hat{\mathfrak{Y}}^{2mn}$ can be described using only a few bits, this guarantees that the model $W$ generalizes well. As it will be shown from the result that will follow, instead of the size of the message needed to be sent in the Blum-Langford approach (see Appendix B.1), a new quantity emerges in our work as a measure of the compressibility of $\hat{\mathfrak{Y}}^{2mn}$: the relative entropy of the joint conditional $P(\hat{Y}^n, \hat{Y}'^n | Y^n, Y'^n)$ and a (symmetric) conditional prior $\mathbf{Q}$ over $\hat{\mathcal{Y}}^{2n}$ given $Y^{2n}$. Depending on the way we rearrange $(S^m, S'^m)$, the type-I or type-II symmetric priors are needed to be applied.

Let $R \in \mathbb{R}^+$. In our block-coding rate-distortion theoretic framework, $R$ is said to be *achievable* if there exists a compression codebook of size $\approx e^{mR}$, fixed a priori, which *covers* the space spanned by the model-predicted labels $\hat{\mathfrak{Y}}^{2mn}$ with high probability. Specifically, if there exists a sequence of label books $\{\hat{\mathcal{Y}}_m\}_{m \in \mathbb{N}}$, where $\hat{\mathcal{Y}}_m := \{\hat{\mathbf{y}}[r], r \in [l_m]\} \subseteq \mathcal{Y}^{2mn}$, $l_m \in \mathbb{N}$, $\hat{\mathbf{y}}[r] = (\hat{\mathbf{y}}_1[r], \ldots, \hat{\mathbf{y}}_m[r])$ and $\hat{\mathbf{y}}_j[r] = (\hat{y}_{j,1}[r], \ldots, \hat{y}_{j,2n}[r]) \in \mathcal{Y}^{2n}$ such that:

(i) $l_m \leqslant e^{mR}$,

(ii) there exist a sequence $\{\delta_m\}_{m \in \mathbb{N}}$ for which $\lim_{m \to \infty} \delta_m = 0$ and with probability at least $(1 - \delta_m)$ over the choices of $S^m$, $S'^m$, and $\mathbf{J}$ or $\mathbf{T}$, one can find at least one index $r \in [l_m]$ whose associated $\hat{\mathbf{y}}[r]$ exactly equals $\hat{\mathfrak{Y}}^{2mn}$.

The above framework, explained in Section 2.1, is called *fixed-size compressibility*, as the size of the codebook does not depend on a given dataset. This type of compressibility is useful for establishing "data-independent" bounds,[7] *e.g.,* bounds on the expectation of the generalization error. This is also called *lossless*, since we look for a codeword that is exactly equal to $\hat{\mathfrak{Y}}^{2mn}$.

### C.1.2 Lossy fixed-size compressibility

As already stated, the above approach and results can be extended to any bounded loss function and continuous variables $Y$ and $\hat{Y}$ (see for example Section 3 where continuous latent variables are studied). In this case, a standard result of rate-distortion theory states that to cover *losslessly* and reliably $\hat{\mathfrak{Y}}^{2mn}$, $R$ should be infinity, which makes the framework and resulting bounds vacuous. However, the approach can

---

[7]Here, we emphasize that by data-dependent bounds we refer to bounds that depend on the particular sample of the input data at hand, rather than for example just on the distribution of the data which is unknown.

be easily extended to include the *lossy compression* of the labels; in the same spirit as done in [SGRS22] for the hypothesis compression. More precisely, for a given distortion threshold $\epsilon \in \mathbb{R}$, one can consider the same way of codebook generation, but with the following conditions:

(i) $l_m \leqslant e^{mR}$,

(ii) there exist a sequence $\{\delta_m\}_{m \in \mathbb{N}}$ for which $\lim_{m \to \infty} \delta_m = 0$ and with probability at least $(1 - \delta_m)$ over the choices of $S^m$, $S'^m$, and $\mathbf{J}$ or $\mathbf{T}$, one can find at least one index $r \in [l_m]$ whose associated $\hat{\mathbf{y}}[r]$ satisfies:

$$\frac{1}{mn} \sum_{j \in [m], i \in [n]} \left( \left( \mathbb{1}_{\{\mathfrak{Y}_{j,t_i^c} \neq \hat{\mathfrak{Y}}_{j,t_i^c}\}} - \mathbb{1}_{\{\mathfrak{Y}_{j,t_i} \neq \hat{\mathfrak{Y}}_{j,t_i}\}} \right) - \left( \mathbb{1}_{\{\mathfrak{Y}_{j,t_i^c} \neq \hat{y}_{j,t_i^c}[r]\}} - \mathbb{1}_{\{\mathfrak{Y}_{j,t_i} \neq \hat{y}_{j,t_i}[r]\}} \right) \right) < \epsilon,$$

where it is assumed that the set of indices $\{(j, t_i)\}_{j \in [m], i \in [n]}$ and $\{(j, t_i^c)\}_{j \in [m], i \in [n]}$ are the sets of indices in the rearranged sequence that belong to the training and ghost datasets, respectively. Here, $\bigcup_{i \in [n]} (t_i \cup t_i^c) = \{1, \ldots, 2n\}$.

In the case of lossy compressibility, the block-coding technique brings another advantage in addition to the advantages discussed for the lossless case: it allows to consider the *average* distortion criterion, instead of *worst-case* distortion criterion. Please refer to [SGRS22] for further discussion on this.

### C.1.3 Variable-size compressibility

In the frameworks, explained in previous sections, for a given $m$, the size of the codebook $e^{mR}$ is fixed. Hence, the bounds derived using this framework, cannot depend on a particular training dataset at hand. In particular, while above mentioned frameworks are useful to establish bounds on the expectation of the generalization error, they are not appropriate for establishing *data-dependent* tail bounds. To overcome this issue, in [SZ23], a "variable-size" compressibility is proposed, in which, one is allowed to search for a suitable $\hat{\mathbf{y}}[r]$ only in the *first* $e^{mR_{(S,S',W)}}$ elements of this codebook, where $R_{(S,S',W)}$ is the data-dependent term that will consequently appear in the resulting bound using this approach. For instance, the term $R_{(S,S',W)}$ in Part.ii of Theorem 1 is exactly the KL-divergence term appearing in the bound.

### C.2 Intuition about the function $h_D$

In Section 2.3, we have provided the bound on the generalization error in terms of the function $h_D(x; x') \colon [0, 1] \times [0, 1] \to [0, 2]$, defined as:

$$h_D(x, x') := 2h_b\left(\frac{x + x'}{2}\right) - h_b(x) - h_b(x'),$$

where $h_b(x) := -x \log_2(x) - (1 - x) \log_2(1 - x)$. As mentioned, $\frac{1}{2} h_D(x, x')$ is equal to the Jensen-Shannon divergence between two binary Bernoulli distributions with parameters $x$ and $x'$.

In this section, we provide an intuition about this function, by showing its relation with the combinatorial term that appeared in [BL03]. Recall that the main result of [BL03], *i.e.,* Theorem 6 therein (re-stated in the Appendix B.1), is established in terms of $b_{\max}(n, a/n, \delta)$ defined as follows:

$$b_{\max}\left(n, \frac{a}{n}, \delta\right) := \max\{b \colon \text{Bucket}(n, a, b) \geqslant \delta\},$$

where

$$\text{Bucket}(n, a, b) := \sum_{c \in [b, a+b]} \frac{\binom{n}{c}\binom{n}{a+b-c}}{\binom{2n}{a+b}},$$

where $[b, a + b] \subset \mathbb{N}$ denotes the integer interval and $c \in \mathbb{N}$.

Now, the intuition about the function $h_D(x, x')$ is as follows: by Stirling's formula, we have that

$$\frac{\binom{mn}{mt}\binom{mn}{ma+mb-mt}}{\binom{2mn}{ma+mb}} \longrightarrow e^{-mnh_D\left(\frac{t}{n}, \frac{a+b-t}{n}\right)} \tag{10}$$

as $m \to \infty$. Hence, for large (infinite) values of $m$ we have that the function $\text{Bucket}(mn, ma, mb)$ of [BL03] is dominated by

$$\max_{t \in [\![b, a+b]\!]} e^{-mnh_D\left(\frac{t}{n}, \frac{a+b-t}{n}\right)}, \tag{11}$$

where $[\![b, a+b]\!] \subset \mathbb{R}$ denotes the real interval and $t \in \mathbb{R}$. Furthermore and as a consequence

$$\frac{1}{m} b_{\max}\left(mn, \frac{a}{n}, \delta^m\right) \longrightarrow \max\left\{b \colon \min_{t \in [\![b, a+b]\!]} n h_D\left(\frac{t}{n}, \frac{a+b-t}{n}\right) \leqslant \log(1/\delta)\right\}, \qquad (12)$$

as $m \to \infty$.

It can be observed that by considering *block-coding* and letting $m \to \infty$, the intractable combinatorial terms appeared in [BL03] can be expressed in terms of the function $h_D(\cdot, \cdot)$.

## C.3 On relation between lossy compressibility and geometric compression

The experimental studies suggest the existence of a relation between generalization performance and *geometrical compression* [GK19]. Geometrical compression occurs when the latent variables are concentrated around a limited number of clusters. Please refer to [GK19, Fig. 2] for a visual representation. As mentioned both in Section 3, *lossy compression* provides an interpretation of the *geometric compression* of [GK19, GVDBG$^+$19] where latent variables are concentrated around some constellation points. In this section, we provide a simple example of that.

In geometrical compression, the latent variables $U \sim P_{U|X}$ of $X$ are distinct but concentrated around a few "centers". Hence, while in this case, the "lossless compression" captured by $I(U; X)$ becomes very large or infinite, the "lossy compression" captured by rate-distortion may be very small, as these scattered latent variables around the centers can be seen as a lossy version of the mappings from $X$ to one of these centers, with some small distortion. To give a simple example, suppose $X \in [0, 1]$, and if $X < 0.5$, then $U = -1 + X/5$, and if $X > 0.5$, then $U = 1 + X/5$. In this case while $I(U; X) = \infty$, a simple lossy mapping $P_{\hat{U}|X}$ with average distortion less than $0.05$ can be found such that $I(\hat{U}; X) = 1$. This explains the geometrical compression observation.

## C.4 Conclusion and future directions

In this paper, inspired by [BL03], we developed a compressibility framework that we used to establish various bounds on the generalization error of the stochastic learning algorithms. The bounds are expressed in terms of the newly defined notion of "minimum description length"(MDL) of the predicted labels or the latent variables. The new notion is expressed in terms of a "symmetric" prior, where the "symmetry" is defined and used in three different ways. The type-I and type-II symmetries are useful to establish the bounds in terms of MDL of the predicted labels. The former one, in particular, shows a clear connection between the seemingly different approaches of [BL03] and CMI [SZ20, HRVSG21]. The type-III symmetry is used to derive the main result of this paper which is a bound in terms of MDL of latent variables. The results of the last section suggest that the generalization error in representation learning is related to the newly defined MDL of latent variables. Unlike mutual information, which captures the "information leakage", the new notion of MDL captures the simplicity and structure of the encoder. These insights are then partly exploited to propose new regularizers in terms of new "data-dependent" priors. The performed simulations show the advantage of these priors over the classical ones used in VIB.

Our proposed framework and obtained results also open up several future research directions. In the following, we discuss some of those directions.

- In Part i. of Theorem 1, we proposed a tail bound, which unlike the classical PAC-Bayes bound with a single draw from the posterior [Cat07], does not contain a "disintegrated" term. As explained, this is due to the choice of the loss function, which inherently contains an expectation term. It would be interesting to consider a loss function that captures the "one-shot" prediction of the performance and to develop a disintegrated bound for such a loss function.

- One of the contributions of this work has been to establish "rate-distortion theoretic" bounds by using a lossy compressibility framework. For example, the established bound in Theorem 4, contains an infimum over all "compressed algorithms" $P_{\hat{W}_e|S}$ that satisfy some distortion criterion. Note that this implies that this bound holds for "any" choice of eligible $P_{\hat{W}_e|S}$, and thus, taking the infimum to obtain a "valid bound" is not necessarily required. Thus, any simple technique that adds noise or applies parameter quantization can be considered. However, besides these general approaches, an interesting direction is how to find an "optimal" $P_{\hat{W}_e|S}$. Perhaps, the approach taken in [THP23], which uses a combination of the MINE estimator [BBR$^+$18] and the "SFRL" [LEG18] could be adapted to our setup.

- Another potentially valuable direction would be to compute the bounds *analytically* in simple setups, such as two-layer neural networks with Gaussian data, and compare the results with the existing results such as [VPV18]. Also, it is instructive to study *numerically* the geometry and MDL of the latent variables for different network architectures, *e.g.,* FCN versus CNN. This may lead to a new understanding of how and why some architectures produce "better" representations.

- Inspired by our results, we have proposed some simple "data-dependent" priors. While the proposed priors show improvements over the classical priors, they are limited to priors that can be factorized as products of some Gaussian distributions. Thus, they only "partially" capture the joint compressibility of the latent variables (and hence the encoder structure). We imagine that more "appropriate" priors can be proposed that capture better the structure of the encoder.

- In this paper, we partially discuss the intuition behind the importance of the "structure" of the encoder. However, the explicit effect of the "structure" on the geometry of latent variables and predictions needs to be investigated, as partially shown in [BM06].

- Finally, we emphasize that in representation learning, one is interested in extracting good representations that are suitable in terms of generalization error for *multiple* learning tasks, simultaneously. It would be interesting to extend our results to such more realistic settings.

# D   Details of the experiments

In this section, we present the details of the experiments presented in Section 4.

## D.1   VIB objective function

The traditional information bottleneck (IB) approach [TPB00, SST10] for training representation learning models, and particularly its variational implementation (VIB) by [AFDM17], considers a fixed data-independent prior $\mathbf{Q} = Q^{\otimes 2n}$. Then, for some Lagrange multiplier $\beta > 0$, the VIB approach minimizes

$$\frac{1}{n} \sum_{i=1}^{n} \Big\{ \beta D_{KL}\Big[ P_{U|X,W_e}(U_i|x_i, W_e) \| Q \Big] - \mathbb{E}_{U_i \sim P_{U|X,W_e}(U_i|x_i, W_e)}\Big[ \log P_{\hat{Y}|U,W_d}(y_i|U_i, W_d) \Big] \Big\},$$

using the reparametrization trick of [KW14]. As can be noticed, the first term, which acts as a *regularizer*, only takes into account the training dataset part of $D_{KL}\Big( P_{U|X,X',W_e}^{\otimes 2n}(\mathbf{U}, \mathbf{U}'|\mathbf{X}, \mathbf{X}', W_e) \big\| \mathbf{Q} \Big)$, as the test set is not available. The second term attempts to maximize the *relevance* of $U$ for prediction. Intuitively, it seeks to find the best decoder among the possible choices, i.e., the one that minimizes the empirical risk.

A popular choice for $Q$ is the multi-dimensional standard Gaussian distribution $Q = \mathcal{N}(\mathbf{0}_m, \mathrm{I}_m)$, where $\mathrm{I}_m$ is the identity matrix, $m$ is the dimension of the latent variable $U$, and $\mathbf{0}_m \in \mathbb{R}^m$ is the all zero vector. In the original implementation of [AFDM17], for each sample $x$, the encoder generates the mean $\mu_x = (\mu_{x,1}, \ldots, \mu_{x,m}) \in \mathbb{R}^m$ and variance $\sigma_x^2 = (\sigma_{x,1}^2, \ldots, \sigma_{x,m}^2) \in \mathbb{R}^m$ of the latent variable $U$. Then, we let $P_{U|X,W_e}(U|x, W_e) = \mathcal{N}(\mu_x, \mathrm{diag}(\sigma_x^2))$, where $\mathrm{diag}(\sigma_x^2) \in \mathbb{R}^{m \times m}$ denotes a diagonal matrix whose diagonal elements are denoted by the vector $\sigma_x^2$. This means that the latent variable for the input $x$ is generated according to $U \sim \mathcal{N}(\mu_x, \mathrm{diag}(\sigma_x^2))$. Hence, the objective function to minimize becomes

$$\frac{1}{b} \sum_{i=1}^{b} \Big\{ \beta \, D_{KL}\Big[ \mathcal{N}(\mu_{x_i}, \mathrm{diag}(\sigma_{x_i}^2)) \| \mathcal{N}(\mathbf{0}_m, \mathrm{I}_m) \Big] - \mathbb{E}_{U_i \sim P_{U|X,W_e}(U_i|x_i, W_e)}\Big[ \log P_{\hat{Y}|U,W_d}(y_i|U_i, W_d) \Big] \Big\},$$

$$(13)$$

where $b$ is the size of a mini-batch of training samples $z_1, \ldots, z_b$, $z_i = (x_i, y_i)$. Moreover, (13) is repeated iteratively over multiple mini-batches until the convergence of the representation learning model.

## D.2   Lossless CDVIB objective function

We describe here another learning approach which, unlike the VIB, uses a data-dependent prior. This prior, coined Category-Dependent VIB (CDVIB) is again factorized as a product of $2n$ scalar Gaussian priors, *i.e.,* $\mathbf{Q} = \prod_{i \in [2n]} Q_i$. Each of these scalar priors $Q_i$ is chosen among one of the $K \times M$ Gaussian priors (centers) – $M$ priors per each label. More precisely, for each label $k \in [K]$, we consider $M \in \mathbb{N}$ priors

$$Q_{k,r}^{(t)} = \mathcal{N}\left( \bar{\mu}_{k,r}^{(t)}, \mathrm{diag}\left( \bar{\sigma}_{k,r}^{(t)\,2} \right) \right), \quad r \in [M], k \in [K],$$

defined over $\mathbb{R}^m$, where the superscript $t \in \mathbb{N}$ represents the optimization iteration.

Unlike in the VIB approach, the mean and variance of the scalar priors are updated during the training process. Firstly, the vectors $\bar{\mu}_{k,r}^{(t)}$ and $\bar{\sigma}_{k,r}^{(t)\,2}$ are initialized respectively as $m$-dimensional vectors of zeros and ones. Next, these initialized vectors are updated at each iteration using a procedure defined below.

For any sample $z = (x, y)$ with label $y = k$ and for any iteration $t \geqslant 1$, let $r \in [M]$ denote the index of the category-dependent prior (center) which is the closest to $\mathcal{N}(\mu_x, \mathrm{diag}(\sigma_x^2))$ in terms of the KL-divergence

$$r_z^{(t)} = \arg\min_{r \in [M]} D_{KL}\left[\mathcal{N}(\mu_x, \mathrm{diag}(\sigma_x^2)) \| \mathcal{N}\left(\bar{\mu}_{k,r}^{(t)}, \bar{\sigma}_{k,r}^{(t)\,2}\right)\right].$$

Suppose that the picked mini-batch at iteration $t$ is $\mathcal{B}_t = \{z_1, \ldots, z_b\}$. For each $k \in [K]$ and $r \in [M]$, let

$$\mathcal{I}_{k,r}^{(t)} = \left\{z_i : i \in [b], y_i = k, r_{z_i}^{(t)} = r\right\},$$

and let $b_{k,r} = |\mathcal{I}_{k,r}^{(t)}|$. Now, if $b_{k,r} \neq 0$, update $\bar{\mu}_{k,r}^{(t)}$ and $\bar{\sigma}_{k,r}^{(t)}$ as

$$\bar{\mu}_{k,r}^{(t)} := (1 - \alpha b_{k,r}) \bar{\mu}_{k,r}^{(t-1)} + \alpha \sum_{z_i \in \mathcal{I}_{k,r}} \mu_{x_i},$$

$$\bar{\sigma}_{k,r,j}^{(t)} := \sqrt{(1 - \alpha b_{k,r})\, \bar{\sigma}_{k,r,j}^{(t-1)\,2} + \alpha \sum_{z_i \in \mathcal{I}_{k,r}} \sigma_{x_i,j}^2}, \quad j = 1, \ldots, m,$$

where $\alpha \in [0, 1]$ is a coefficient that smoothens the evolution of the mean and variance, and also implicitly takes into account the effect of the ghost dataset $S'$ appearing in the bounds of Theorems 4 and 7. The optimal value of $\alpha$, among others, depends on the mini-batch size $b$ and the number of centers $M$.[8]

Finally, the considered objective function at iteration $t$ in the Lossless CDVIB approach is

$$\frac{1}{b} \sum_{i=1}^{b} \left\{\beta\, D_{KL}\left[\mathcal{N}\left(\mu_{x_i}, \mathrm{diag}(\sigma_{x_i}^2)\right) \| \mathcal{N}\left(\bar{\mu}_{y_i, r_{z_i}^{(t)}}^{(t)}, \mathrm{diag}(\bar{\sigma}_{y_i, r_{z_i}^{(t)}}^{(t)\,2})\right)\right]\right.$$

$$\left. - \mathbb{E}_{U_i \sim P_{U|X, W_e}(U_i|x_i, W_e)}\left[\log P_{\hat{Y}|U, W_d}(y_i|U_i, W_d)\right]\right\}. \tag{14}$$

### D.3 Lossy CDVIB objective function

Inspired by the lossy compression and bounds introduced in our work, we consider the MDL of the "perturbed" latent variable, while passing the un-perturbed latent variable to the decoder. More precisely, as before, we consider the log loss for evaluation of the relevance of $U$ in the decoder, *i.e.,*

$$\mathbb{E}_{U_i \sim P_{U|X, W_e}(U_i|x_i, W_e)}\left[\log P_{\hat{Y}|U, W_d}(y_i|U_i, W_d)\right].$$

For the regularizer, we first consider the perturbed $U$ as

$$\hat{U} = U + Z_2 = \mu_X + Z_2 + \sigma_X Z_1 = \hat{U}_1 + \hat{U}_2, \tag{15}$$

where $Z_1$ and $Z_2$ are independently drawn from the same distribution $\mathcal{N}(\mathbf{0}_m, \mathrm{I}_m)$. Note that we chose

$$\hat{U}_1 := \mu_X + Z_2, \qquad \hat{U}_2 := \sigma_X Z_1. \tag{16}$$

Hence, given $(X, W_e)$, $\hat{U}_1 \sim \mathcal{N}(\mu_X, \mathrm{I}_m)$ is independent from $\hat{U}_2 \sim \mathcal{N}(\mathbf{0}_m, \mathrm{diag}(\sigma_X^2))$. Let us define two sets of priors $\mathcal{Q}_1 := \{Q_{1,k,r}\}_{k \in [K], r \in [M]}$ over $\hat{U}_1$ and $\mathcal{Q}_2 := \{Q_{2,k,r}\}_{k \in [K], r \in [M]}$ over $\hat{U}_2$. Next, for each $i \in [b]$, we select two priors $Q_{1,i} \in \mathcal{Q}_1$ and $Q_{2,i} \in \mathcal{Q}_2$, in a manner that will become clear in the following. Denote the induced prior for $\hat{U} = \hat{U}_1 + \hat{U}_2$, where $\hat{U}_1 \sim Q_{1,i}$ and $\hat{U}_2 \sim Q_{2,i}$ by $Q_i$. Then, the KL divergence of $P_{\hat{U}|X, W_e}(\hat{U}|X, W_e)$ and $Q_i$, can be upper bounded by

$$D_{KL}\left[P_{\hat{U}|X, W_e}(\hat{U}|X, W_e) \| Q_i\right]$$

$$\leqslant D_{KL}\left[P_{\hat{U}_1|X, W_e}(\hat{U}_1|X, W_e) \| Q_{1,i}\right] + D_{KL}\left[P_{\hat{U}_2|X, W_e}(\hat{U}_2|X, W_e) \| Q_{2,i}\right]$$

$$= D_{KL}\left[\mathcal{N}(\mu_X, \mathrm{I}_m) \| Q_{1,i}\right] + D_{KL}\left[\mathcal{N}(\mathbf{0}_m, \mathrm{diag}(\sigma_X^2)) \| Q_{2,i}\right]. \tag{17}$$

---

[8]It can be shown that this choice of prior for $M = 1$ satisfies the type-III symmetry property and for $M > 1$ is an approximation of a prior that satisfies such a symmetry.

We use this upper bound for our regularizer. To make things more formal, for each label $k \in [K]$, we consider $2M$ priors, $M \in \mathbb{N}$,

$$Q_{1,k,r}^{(t)} = \mathcal{N}\left(\bar{\mu}_{k,r}^{(t)}, \mathrm{I}_m\right), \quad r \in [M], k \in [K],$$

$$Q_{2,k,r}^{(t)} = \mathcal{N}\left(\mathbf{0}_m, \mathrm{diag}\left(\bar{\sigma}_{k,r}^{(t)\,2}\right)\right), \quad r \in [M], k \in [K],$$

defined over $\mathbb{R}^m$, where the superscript $t \in \mathbb{N}$ represents the optimization iteration.

Similar to lossless CDVIB, firstly, the vectors $\bar{\mu}_{k,r}^{(t)}$ and $\bar{\sigma}_{k,r}^{(t)\,2}$ are initialized respectively as $m$-dimensional vectors of zeros and ones. Next, these initialized vectors are updated at each iteration using a procedure defined below.

For any sample $z = (x, y)$ with label $y = k$ and for any iteration $t \geqslant 1$, let $r \in [M]$ denote the index of the category-dependent prior (center) which has the smallest distance to $(Q_{1,k,r}^{(t)}, Q_{2,k,r}^{(t)})$ in a sense that minimizes the RHS of (17). In other words, $r_z^{(t)}$ is equal to

$$\underset{r \in [M]}{\arg\min}\left\{ D_{KL}\left[\mathcal{N}(\mu_x, \mathrm{I}_m)\|\mathcal{N}\left(\bar{\mu}_{k,r}^{(t)}, \mathrm{I}_m\right)\right] + D_{KL}\left[\mathcal{N}(\mathbf{0}_m, \mathrm{diag}(\sigma_x^2))\|\mathcal{N}\left(\mathbf{0}_m, \mathrm{diag}\left(\bar{\sigma}_{k,r}^{(t)\,2}\right)\right)\right]\right\}.$$

Then, the mean and variances of the priors are updated exactly similarly to Lossless CDVIB. For completeness, we repeat this procedure here. Suppose that the picked mini-batch at iteration $t$ is $\mathcal{B}_t = \{z_1, \ldots, z_b\}$. For each $k \in [K]$ and $r \in [M]$, let

$$\mathcal{I}_{k,r}^{(t)} = \left\{z_i : i \in [b], y_i = k, r_{z_i}^{(t)} = r\right\},$$

and let $b_{k,r} = |\mathcal{I}_{k,r}|$. Now, if $b_{k,r} \neq 0$, update $\bar{\mu}_{k,r}^{(t)}$ and $\bar{\sigma}_{k,r}^{(t)}$ as

$$\bar{\mu}_{k,r}^{(t)} := (1 - \alpha b_{k,r})\bar{\mu}_{k,r}^{(t-1)} + \alpha \sum_{z_i \in \mathcal{I}_{k,r}} \mu_{x_i},$$

$$\bar{\sigma}_{k,r,j}^{(t)} := \sqrt{(1 - \alpha b_{k,r})\bar{\sigma}_{k,r,j}^{(t-1)\,2} + \alpha \sum_{z_i \in \mathcal{I}_{k,r}} \sigma_{x_i,j}^2}, \quad j = 1, \ldots, m,$$

where $\alpha \in [0, 1]$ is a coefficient that smoothens the evolution of the mean and variance, and also implicitly takes into account the effect of the ghost dataset $S'$ appearing in the bounds of Theorems 4 and 7. The optimal value of $\alpha$, among others, depends on the mini-batch size $b$ and the number of centers $M$.

Finally, the considered objective function at iteration $t$ in the Lossy CDVIB approach is

$$\frac{1}{b}\sum_{i=1}^{b}\left\{\beta D_{KL}\left[\mathcal{N}(\mu_{x_i}, \mathrm{I}_m)\|\mathcal{N}\left(\bar{\mu}_{y_i,r_{z_i}^{(t)}}^{(t)}, \mathrm{I}_m\right)\right] + \beta D_{KL}\left[\mathcal{N}(\mathbf{0}_m, \mathrm{diag}(\sigma_{x_i}^2))\|\mathcal{N}\left(\mathbf{0}_m, \mathrm{diag}\left(\bar{\sigma}_{y_i,r_{z_i}^{(t)}}^{(t)\,2}\right)\right)\right]\right.$$

$$\left. - \mathbb{E}_{U_i \sim P_{U|X,W_e}(U_i|x_i,W_e)}\left[\log P_{\hat{Y}|U,W_d}(y_i|U_i, W_d)\right]\right\}. \tag{18}$$

### D.4 Datasets

In the experiments, we used CIFAR10 [KH$^+$09]. The full dataset was split into a training set with 50,000 labeled images and a validation set with 10,000 labeled images, all of them of size $32 \times 32 \times 3$. The input images were scaled to have most values between 0 and 1 before being fed to the network.

### D.5 Architecture details

The model architecture considered in our experiments is detailed in Table 1. The encoder part of our two-step prediction model is a convolutional network consisting of four convolutional layers followed by two linear layers. We use max-pooling and a LeakyReLU activation function with a negative slope coefficient equal to 0.1. The encoder takes as input re-scaled images and produces parameters $\mu_x$ and variance $\sigma_x^2$ of the latent variable of dimension $m = 64$. The latent samples are generated using the reparameterization trick of [KW14]. Next, the produced latent samples are processed by a decoder consisting of one linear layer with a softmax activation function. The decoder outputs a soft class prediction.

Similarly, as in [DKSV20], our evaluated encoder is complex enough in order to make it close to "a universal function approximator". On the other hand, we use a simple decoder as in [AFDM17] in order to reduce spurious regularization introduced by the high decoder's complexity and hence to highlight the benefits of our regularizer in terms of generalization performance.

Table 1: The model architecture used in experiments. The convolutional layers are parametrized respectively by the number of input channels, the number of output channels, and the filter size. The linear layers are defined by their input and output sizes.

| Encoder | | Encoder cont'd | | Encoder cont'd | |
|---|---|---|---|---|---|
| Number | Layer | Number | Layer | Number | Layer |
| 1 | Conv2D(3,8,5) | 6 | Conv2D(16,16,3) | 11 | LeakyReLU(0.1) |
| 2 | Conv2D(3,8,5) | 7 | LeakyReLU(0.1) | 12 | Linear(256,128) |
| 3 | LeakyReLU(0.1) | 8 | MaxPool(2,2) | Decoder | |
| 4 | MaxPool(2,2) | 9 | Flatten | 1 | Linear(64,10) |
| 5 | Conv2D(8,16,3) | 10 | Linear(1024,256) | 2 | Softmax |

## D.6   Implementation and training details

Our prediction model was trained using PyTorch [PGM+19] and a GPU Tesla P100 with CUDA 11.0. All weights were initialized using the default PyTorch Xavier initialization scheme [GB10] with all biases initialized to zero. The Adam optimizer [KB15] ($\beta_1 = 0.5$, $\beta_2 = 0.999$) was used with an initial learning rate of $10^{-4}$ and an exponential decay of 0.97. The batch size was equal to 128 throughout the whole experiment. The code used in the experiments is available at `https://github.com/PiotrKrasnowski/MDL_and_Generalization_Guarantees_for_Representation_Learning`.

During the training phase, we jointly trained the encoder and the decoder parts for 200 epochs using either the standard Gaussian prior of the traditional VIB objective function or our CDVIB objective functions. As in [AFDM17], we generated one latent sample per image during training and 12 samples during testing.

## D.7   Numerical findings

Figure 3 displays the training and test performance of our model for a wide range of parameters $\beta$. It could be noticed that the best test accuracy with our Lossy CDVIB objective function is 65% (for $\beta = 0.01$), which is about 2.5% better than the best test accuracy for VIB (62.5% for $\beta = 0.005$). In the case of Lossless CDVIB, the best achieved test accuracy is 64% for $\beta = 1e-5$.

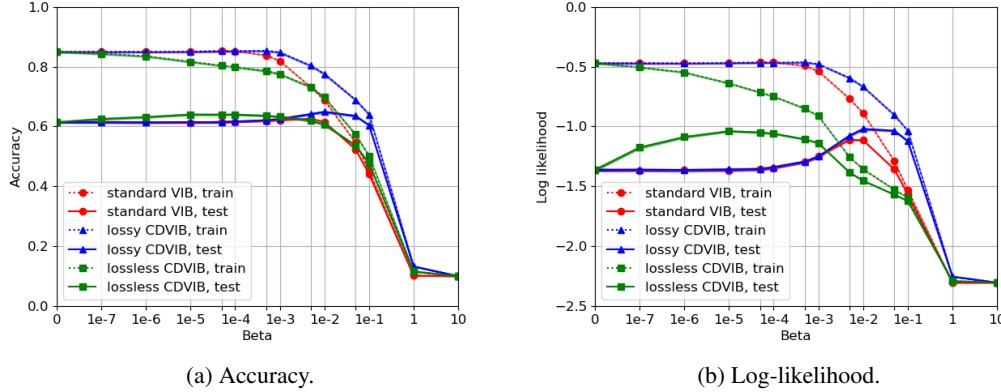

    (a) Accuracy.                (b) Log-likelihood.

Figure 3: Test and train performances of our two-step prediction model trained using the standard VIB prior, the "lossless" CDVIB prior, and the "lossy" CDVIB prior, both with $M = 5$. The plots show the average over 5 runs and 95% bootstrap confidence intervals.

# E  Proofs

## E.1  General proof techniques

Most of our proofs contain two main steps.

Technically speaking, the first step uses Donsker-Varadhan's variational representation lemma, to change the measure. As an example, in proof of Theorem 1, the first step gives

$$\lambda \mathbb{E}_{S,W}[\text{gen}(S,W)]$$

$$\leqslant D_{KL}\left(\mu_Y^{\otimes 2n}\mathbb{E}_{S',S,W}\left[P_{\hat{Y}|X,W}^{\otimes 2n}(\hat{Y}^n,\hat{Y}'^n|X^n,X'^n,W)\right]\middle\|\mu_Y^{\otimes 2n}\mathbf{Q}(\hat{Y}^n,\hat{Y}_i'^n|Y^n,Y'^n)\right)$$

$$+ \log \mathbb{E}_{Y^n,Y'^n,\hat{Y}^n,\hat{Y}'^n \sim \mu_Y^{\otimes 2n}\mathbf{Q}(\hat{Y}^n,\hat{Y}_i'^n|Y^n,Y'^n)}\left[e^{\frac{\lambda}{n}\sum_{i\in[n]}\left(\mathbb{1}_{\{Y_i'\neq\hat{Y}_i'\}}-\mathbb{1}_{\{Y_i\neq\hat{Y}_i\}}\right)}\right],$$

that changes the measure from

$$\mu_Y^{\otimes 2n}\mathbb{E}_{S',S,W}\left[P_{\hat{Y}|X,W}^{\otimes 2n}(\hat{Y}^n,\hat{Y}'^n|X^n,X'^n,W)\right],$$

to

$$\mu_Y^{\otimes 2n}\mathbf{Q}(\hat{Y}^n,\hat{Y}'^n|Y^n,Y'^n).$$

This has a particular meaning and intuition in our *fixed-size* compressibility approach. Indeed, one can show that there exists a proper sequence of *compression books*, with size $|\hat{\mathcal{Y}}_m| \leqslant e^{mR}$, where

$$R = D_{KL}\left(\mu_Y^{\otimes 2n}P_{\hat{Y}^n,\hat{Y}'^n|Y^n,Y'^n}\middle\|\mu_Y^{\otimes 2n}\mathbf{Q}(\hat{Y}^n,\hat{Y}_i'^n|Y^n,Y'^n)\right).$$

This means that by this step, we *fix* a suitable sequence of compression books such that with high probability (that goes to 1 as $m \to \infty$), one can find the sequence of predicted labels in this codebook. Using Donsker-Varadhan's change of measure is a shortcut to this, as previously explained in [SGRS22, Appendix B.1.] in the context of hypothesis compression. Then, we consider the union bound over all elements of this codebook in the next step. For tail bounds, similar interpretations hold, but this time with *variable-size* compressibility notion, as shown in [SZ23], in the context of hypothesis compression.

The second step can be seen as bounding the generalization error for *every* element of such codebook. This step is achieved in quite different manners for the proofs of different results of the paper, as follows in the coming sections. Note that in this step, the predicted labels have distributions according to the prior $\mathbf{Q}(\hat{Y}^n,\hat{Y}_i'^n|Y^n,Y'^n)$, rather than the one induced by the learning algorithm, *i.e.*, $P_{\hat{Y}|X,W}^{\otimes 2n}(\hat{Y}^n,\hat{Y}'^n|X^n,X'^n,W)$.

## E.2  Proof of Theorem 1

### E.2.1  Part i.

*Proof.* We first show that

$$\inf_{\mathbf{Q}\in\mathcal{Q}_i}\mathbb{E}_{\mathbf{Y},\mathbf{Y}'}\left[D_{KL}\left(\mathbb{E}_{\mathbf{X}',\mathbf{X},W}\left[P_{\hat{Y}|X,W}^{\otimes 2n}(\hat{\mathbf{Y}},\hat{\mathbf{Y}}'|\mathbf{X},\mathbf{X}',W)\right]\middle\|\mathbf{Q}\right)\right] = I\left(\mathbf{J};\hat{Y}^{2n}|Y^{2n}\right). \tag{19}$$

Let $\mathcal{Q}_i'$, be the set of conditional priors $\mathbf{Q}'$ that can be written as

$$\mathbf{Q}'\left(\hat{\mathbf{Y}},\hat{\mathbf{Y}}'|\mathbf{Y},\mathbf{Y}'\right) = \mathbb{E}_{\mathbf{J}}\left[\mathbf{Q}_1'\left(\hat{Y}_{\mathbf{J}}^{2n},\hat{Y}_{\mathbf{J}^c}^{2n}|Y_{\mathbf{J}}^{2n},Y_{\mathbf{J}^c}^{2n}\right)\right], \tag{20}$$

for some arbitrary distribution $\mathbf{Q}_1'$. Here $Y^{2n}$ and $\hat{Y}^{2n}$ are the concatenations of the vectors $(\mathbf{Y},\mathbf{Y}')$ and $(\hat{\mathbf{Y}},\hat{\mathbf{Y}}')$, respectively. It is easy to verify that $\mathcal{Q}_i = \mathcal{Q}_i'$. Hence, by denoting $P(\hat{\mathbf{Y}},\hat{\mathbf{Y}}'|\mathbf{Y},\mathbf{Y}') \coloneqq \mathbb{E}_{\mathbf{X}',\mathbf{X},W}\left[P_{\hat{Y}|X,W}^{\otimes 2n}(\hat{\mathbf{Y}},\hat{\mathbf{Y}}'|\mathbf{X},\mathbf{X}',W)\right]$ we can write

$$\text{LHS} = \inf_{\mathbf{Q}\in\mathcal{Q}_i}\mathbb{E}_{\mathbf{Y},\mathbf{Y}'}\left[D_{KL}\left(P(\hat{\mathbf{Y}},\hat{\mathbf{Y}}'|\mathbf{Y},\mathbf{Y}')\middle\|\mathbf{Q}\right)\right]$$

$$= \inf_{\mathbf{Q}'\in\mathcal{Q}_i'}\mathbb{E}_{\mathbf{Y},\mathbf{Y}'}\left[D_{KL}\left(P(\hat{\mathbf{Y}},\hat{\mathbf{Y}}'|\mathbf{Y},\mathbf{Y}')\middle\|\mathbf{Q}'\right)\right]$$

$$= \inf_{\mathbf{Q}' \in \mathcal{Q}'_i} \mathbb{E}_{Y^{2n}} \mathbb{E}_{\mathbf{J}} \Big[ D_{KL}\Big( P\big(\hat{Y}_{\mathbf{J}}^{2n}, \hat{Y}_{\mathbf{J}^c}^{2n} | Y_{\mathbf{J}}^{2n}, Y_{\mathbf{J}^c}^{2n}\big) \Big\| \mathbf{Q}' \Big) \Big]$$

$$= \inf_{\mathbf{Q}'_1} \mathbb{E}_{Y^{2n}} \mathbb{E}_{\mathbf{J}} \Big[ D_{KL}\Big( P\big(\hat{Y}_{\mathbf{J}}^{2n}, \hat{Y}_{\mathbf{J}^c}^{2n} | Y_{\mathbf{J}}^{2n}, Y_{\mathbf{J}^c}^{2n}\big) \Big\| \mathbb{E}_{\mathbf{J}}\big[ \mathbf{Q}'_1\big(\hat{Y}_{\mathbf{J}}^{2n}, \hat{Y}_{\mathbf{J}^c}^{2n} | Y_{\mathbf{J}}^{2n}, Y_{\mathbf{J}^c}^{2n}\big) \big] \Big) \Big]$$

$$= I\Big( \mathbf{J} ; \hat{Y}^{2n} | Y^{2n} \Big).$$

This completes the proof of showing the equality in (2).

Now, we proceed to prove the upper bound. Let $\mathbf{Q}$ be any fixed type-I symmetric conditional prior on $(\hat{Y}^n, \hat{Y}'^n)$ given $(Y^n, Y'^n)$. We show that

$$\mathbb{E}_{S,W}[\mathrm{gen}(S,W)] \leqslant \sqrt{ \frac{2 \mathbb{E}_{Y^n, Y'^n} \Big[ D_{KL}\Big( \mathbb{E}_{X'^n, X^n, W} \Big[ P_{\hat{Y}|X,W}^{\otimes 2n}(\hat{Y}^n, \hat{Y}'^n | X^n, X'^n, W) \Big] \Big\| \mathbf{Q} \Big) \Big] }{n} }, \quad (21)$$

where $Y^n, Y'^n \sim \mu_Y^{\otimes 2n}$ and $X'^n, X^n, W \sim P_{X'^n | Y'^n} P_{X^n, W | Y^n}$. For ease of notation, denote

$$P_{\hat{Y}^n, \hat{Y}'^n | Y^n, Y'^n} := \mathbb{E}_{X'^n, X^n, W} \Big[ P_{\hat{Y}|X,W}^{\otimes 2n}(\hat{Y}^n, \hat{Y}'^n | X^n, X'^n, W) \Big],$$

where again $X'^n, X^n, W \sim P_{X'^n | Y'^n} P_{X^n, W | Y^n}$. We start the proof by applying the change of measure using Donsker-Varadhan's variational representation (step $(a)$ below):

$$\lambda \mathbb{E}_{S,W}[\mathrm{gen}(S,W)]$$

$$= \mathbb{E}_{S, S', W, \hat{Y}^n, \hat{Y}'^n \sim P_{S'} P_{S,W} P_{\hat{Y}|X,W}^{\otimes 2n}(\hat{Y}^n, \hat{Y}'^n | X^n, X'^n, W)} \left[ \frac{\lambda}{n} \sum_{i \in [n]} \Big( \mathbb{1}_{\{Y'_i \neq \hat{Y}'_i\}} - \mathbb{1}_{\{Y_i \neq \hat{Y}_i\}} \Big) \right]$$

$$= \mathbb{E}_{Y^n, Y'^n, \hat{Y}^n, \hat{Y}'^n \sim \mu_Y^{\otimes 2n} P_{\hat{Y}^n, \hat{Y}'^n | Y^n, Y'^n}} \left[ \frac{\lambda}{n} \sum_{i \in [n]} \Big( \mathbb{1}_{\{Y'_i \neq \hat{Y}'_i\}} - \mathbb{1}_{\{Y_i \neq \hat{Y}_i\}} \Big) \right]$$

$$\overset{(a)}{\leqslant} D_{KL}\Big( \mu_Y^{\otimes 2n} P_{\hat{Y}^n, \hat{Y}'^n | Y^n, Y'^n} \Big\| \mu_Y^{\otimes 2n} \mathbf{Q}(\hat{Y}^n, \hat{Y}_i'^n | Y^n, Y'^n) \Big)$$

$$+ \log \mathbb{E}_{Y^n, Y'^n, \hat{Y}^n, \hat{Y}'^n \sim \mu_Y^{\otimes 2n} \mathbf{Q}(\hat{Y}^n, \hat{Y}_i'^n | Y^n, Y'^n)} \left[ e^{\frac{\lambda}{n} \sum_{i \in [n]} \Big( \mathbb{1}_{\{Y'_i \neq \hat{Y}'_i\}} - \mathbb{1}_{\{Y_i \neq \hat{Y}_i\}} \Big)} \right]. \quad (22)$$

Now, we bound the second term. Consider the notation $\mathfrak{Y}_{i,j} \in \mathcal{Y}^{2n}$ and $\hat{\mathfrak{Y}}_{i,j} \in \mathcal{Y}^{2n}$ for $i \in [n]$ and $j \in \{1, 2\}$. Denote $j^c := \mathbb{1}_{\{j=1\}} + 1$. Furthermore, for every $i \in [n]$, let $K_i$ be a random variable that takes values uniformly over $\{1, 2\}$ — The variables $\{K_i\}$ are assumed to be mutually independent. Let the complement variable $K_i^c$ be equal to 2 if $K_{i,j} = 1$ and 1 otherwise.

$$\log \mathbb{E}_{Y^n, Y'^n, \hat{Y}^n, \hat{Y}'^n \sim \mu_Y^{\otimes 2n} \mathbf{Q}(\hat{Y}^n, \hat{Y}_i'^n | Y^n, Y'^n)} \left[ e^{\frac{\lambda}{n} \sum_{i \in [n]} \Big( \mathbb{1}_{\{Y'_i \neq \hat{Y}'_i\}} - \mathbb{1}_{\{Y_i \neq \hat{Y}_i\}} \Big)} \right]$$

$$\overset{(a)}{=} \log \mathbb{E}_{\mathfrak{Y}^{n \times 2}, \hat{\mathfrak{Y}}^{n \times 2}, K^n \sim P_Y^{\otimes 2n} \mathbf{Q}(\hat{\mathfrak{Y}}^{n \times 2} | \mathfrak{Y}^{n \times 2}) \, \mathrm{Unif}(1,2)^{\otimes n}} \left[ e^{\frac{\lambda}{n} \sum_{i \in [n]} \Big( \mathbb{1}_{\{\mathfrak{Y}_{K_i, i} \neq \hat{\mathfrak{Y}}_{K_i, i}\}} - \mathbb{1}_{\{\mathfrak{Y}_{K_i^c, i} \neq \hat{\mathfrak{Y}}_{K_i^c, i}\}} \Big)} \right]$$

$$= \log \mathbb{E}_{\mathfrak{Y}^{n \times 2}, \hat{\mathfrak{Y}}^{n \times 2} \sim P_Y^{\otimes 2n} \mathbf{Q}(\hat{\mathfrak{Y}}^{n \times 2} | \mathfrak{Y}^{n \times 2})} \mathbb{E}_{K^n \sim \mathrm{Unif}(1,2)^{\otimes n}} \left[ e^{\frac{\lambda}{n} \sum_{i \in [n]} \Big( \mathbb{1}_{\{\mathfrak{Y}_{K_i, i} \neq \hat{\mathfrak{Y}}_{K_i, i}\}} - \mathbb{1}_{\{\mathfrak{Y}_{K_i^c, i} \neq \hat{\mathfrak{Y}}_{K_i^c, i}\}} \Big)} \right]$$

$$\overset{(b)}{\leqslant} \log \left( \frac{e^{\lambda/n} + e^{-\lambda/n}}{2} \right)^n$$

$$\leqslant \frac{\lambda^2}{2n}, \quad (23)$$

where $(a)$ is concluded by symmetry of $\mathbf{Q}$ and $(b)$ by inequality $\frac{e^x + e^{-x}}{2} \leqslant e^{x^2/2}$. Combining this with (22) yield

$$\mathbb{E}_{S,W}[\mathrm{gen}(S,W)] \leqslant \frac{1}{\lambda} D_{KL}\Big( \mu_Y^{\otimes 2n} P_{\hat{Y}^n, \hat{Y}'^n | Y^n, Y'^n} \Big\| \mu_Y^{\otimes 2n} \mathbf{Q}(\hat{Y}^n, \hat{Y}_i'^n | Y^n, Y'^n) \Big) + \frac{\lambda}{2n}.$$

Letting

$$\lambda := \sqrt{2nD_{KL}\left(\mu_Y^{\otimes 2n}P_{\hat{Y}^n,\hat{Y}'^n|Y^n,Y'^n}\middle\|\mu_Y^{\otimes 2n}\mathbf{Q}(\hat{Y}^n,\hat{Y}_i'^n|Y^n,Y'^n)\right)},$$

completes the proof. $\square$

### E.2.2 Part ii.

*Proof.* The proof of this proposition follows similarly as proof of Theorem 5 (with $\epsilon = 0$), and avoided for brevity. Note that similar to Theorem 5, by noting that with probability at least $(1 - \delta)$, $\hat{\mathcal{L}}(S', W) \geqslant \mathcal{L}(W) - \sqrt{\log(1/\delta)/(2n)}$, one can also establish a tail bound on $\mathrm{gen}(S, W)$. $\square$

### E.3 Proof of Lemma 1

*Proof.* Parts i. and ii. can be easily verified numerically. To show Part iii., take the derivative with respect to $x$. This derivative, *i.e.*, $\log\left(\frac{2-x-x'}{x+x'}\right) - \log\left(\frac{1-x}{x}\right)$, is always non-negative for $1 > x > x'$. For Part iv. the second partial derivative of $h_D(x, x')$ with respect to $x$ is equal to $\frac{1}{x(1-x)} - \frac{2}{(x+x')(2-x-x')}$ which is always positive for $0 \leqslant x, x' \leqslant 1$. Finally, we show the convexity with respect to both variables $x$ and $x'$ simultaneously. The Hessian of the function $h_D(x, x')$ equals

$$H = \begin{bmatrix} \frac{1}{x(1-x)} - \frac{2}{(x+x')(2-x-x')} & -\frac{2}{(x+x')(2-x-x')} \\ -\frac{2}{(x+x')(2-x-x')} & \frac{1}{x'(1-x')} - \frac{2}{(x+x')(2-x-x')} \end{bmatrix}. \tag{24}$$

We show the eigenvalues of this symmetric matrix is always non-negative, and hence $H$ is positive semi-definite. This completes the proof.

Solving

$$|\lambda\mathrm{I}_2 - H| = \left|\begin{bmatrix} \lambda - \frac{1}{x(1-x)} + \frac{2}{(x+x')(2-x-x')} & \frac{2}{(x+x')(2-x-x')} \\ \frac{2}{(x+x')(2-x-x')} & \lambda - \frac{1}{x'(1-x')} + \frac{2}{(x+x')(2-x-x')} \end{bmatrix}\right| = 0, \tag{25}$$

reduces to solving

$$(\lambda + a)^2 - \left(\frac{1}{x(1-x)} + \frac{1}{x'(1-x')}\right)(\lambda + a) + \frac{1}{x(1-x)x'(1-x')} - a^2 = 0, \tag{26}$$

where $a := \frac{2}{(x+x')(2-x-x')}$. The roots of this equation are

$$\lambda = \frac{\left(\frac{1}{x(1-x)} + \frac{1}{x'(1-x')}\right) \pm \sqrt{\left(\frac{1}{x(1-x)} - \frac{1}{x'(1-x')}\right)^2 + 4a^2}}{2} - a. \tag{27}$$

It is straightforward to verify that both roots are always non-negative, which completes the proof. $\square$

### E.4 Proof of Theorem 3

*Proof.* We first show that

$$\inf_{\mathbf{Q}\in\mathcal{Q}_{ii}} \mathbb{E}_{\mathbf{Y},\mathbf{Y}'}\left[D_{KL}\left(\mathbb{E}_{\mathbf{X}',\mathbf{X},W}\left[P_{\hat{Y}|X,W}^{\otimes 2n}(\hat{\mathbf{Y}}, \hat{\mathbf{Y}}'|\mathbf{X}, \mathbf{X}', W)\right]\middle\|\mathbf{Q}\right)\right] = I\left(\mathbf{T}; \hat{Y}^{2n}|Y^{2n}\right). \tag{28}$$

Let $\mathcal{Q}'_{ii}$ be the set of conditional priors $\mathbf{Q}'$ that can be written as

$$\mathbf{Q}'\left(\hat{\mathbf{Y}}, \hat{\mathbf{Y}}'|\mathbf{Y}, \mathbf{Y}'\right) = \mathbb{E}_{\mathbf{T}}\left[\mathbf{Q}'_1\left(\hat{Y}_{\mathbf{T}}^{2n}, \hat{Y}_{\mathbf{T}^c}^{2n}|Y_{\mathbf{T}}^{2n}, Y_{\mathbf{T}^c}^{2n}\right)\right], \tag{29}$$

for some arbitrary (and not necessarily symmetric distribution $\mathbf{Q}'_1$. Here $Y^{2n}$ and $\hat{Y}^{2n}$ are the concatenations of the vectors $(\mathbf{Y}, \mathbf{Y}')$ and $(\hat{\mathbf{Y}}, \hat{\mathbf{Y}}')$, respectively. It is easy to verify that $\mathcal{Q}_{ii} = \mathcal{Q}'_{ii}$. Hence, by denoting

$P(\hat{\mathbf{Y}}, \hat{\mathbf{Y}}'|\mathbf{Y}, \mathbf{Y}') := \mathbb{E}_{\mathbf{X}', \mathbf{X}, W}\left[P_{\hat{Y}|X,W}^{\otimes 2n}(\hat{\mathbf{Y}}, \hat{\mathbf{Y}}'|\mathbf{X}, \mathbf{X}', W)\right]$ we can write

$$
\begin{aligned}
\text{LHS} &= \inf_{\mathbf{Q} \in \mathcal{Q}_{ii}} \mathbb{E}_{\mathbf{Y}, \mathbf{Y}'}\left[D_{KL}\left(P(\hat{\mathbf{Y}}, \hat{\mathbf{Y}}'|\mathbf{Y}, \mathbf{Y}')\middle\|\mathbf{Q}\right)\right] \\
&= \inf_{\mathbf{Q}' \in \mathcal{Q}'_{ii}} \mathbb{E}_{\mathbf{Y}, \mathbf{Y}'}\left[D_{KL}\left(P(\hat{\mathbf{Y}}, \hat{\mathbf{Y}}'|\mathbf{Y}, \mathbf{Y}')\middle\|\mathbf{Q}'\right)\right] \\
&= \inf_{\mathbf{Q}' \in \mathcal{Q}'_{ii}} \mathbb{E}_{Y^{2n}}\mathbb{E}_{\mathbf{T}}\left[D_{KL}\left(P\left(\hat{Y}_{\mathbf{T}}^{2n}, \hat{Y}_{\mathbf{T}^c}^{2n}|Y_{\mathbf{T}}^{2n}, Y_{\mathbf{T}^c}^{2n}\right)\middle\|\mathbf{Q}'\right)\right] \\
&= \inf_{\mathbf{Q}'_1} \mathbb{E}_{Y^{2n}}\mathbb{E}_{\mathbf{T}}\left[D_{KL}\left(P\left(\hat{Y}_{\mathbf{T}}^{2n}, \hat{Y}_{\mathbf{T}^c}^{2n}|Y_{\mathbf{T}}^{2n}, Y_{\mathbf{T}^c}^{2n}\right)\middle\|\mathbb{E}_{\mathbf{T}}\left[\mathbf{Q}'_1\left(\hat{Y}_{\mathbf{T}}^{2n}, \hat{Y}_{\mathbf{T}^c}^{2n}|Y_{\mathbf{T}}^{2n}, Y_{\mathbf{T}^c}^{2n}\right)\right]\right)\right] \\
&= I\left(\mathbf{T}; \hat{Y}^{2n}|Y^{2n}\right).
\end{aligned}
$$

This completes the proof of showing the equality in (28).

Now we proceed to show the upper bound on $nh_D\left(\mathbb{E}_W[\mathcal{L}(W)], \mathbb{E}_{S,W}\left[\hat{\mathcal{L}}(S, W)\right]\right)$. Let $\mathbf{Q}$ be any fixed type-II symmetric conditional prior on $(\hat{Y}^n, \hat{Y}'^n)$ given $(Y^n, Y'^n)$. We show that for $n \geqslant 10$,

$$
\begin{aligned}
nh_D\left(\mathbb{E}_W[\mathcal{L}(W)], \mathbb{E}_{S,W}\left[\hat{\mathcal{L}}(S, W)\right]\right) & \\
&\leqslant \mathbb{E}_{Y^n, Y'^n}\left[D_{KL}\left(\mathbb{E}_{S',S,W}\left[P_{\hat{Y}|X,W}^{\otimes 2n}(\hat{Y}^n, \hat{Y}'^n|X^n, X'^n, W)\right]\middle\|\mathbf{Q}\right)\right] + \log(n),
\end{aligned}
$$

where $Y^n, Y'^n \sim \mu_Y^{\otimes 2n}$ and $S', S, W \sim P_{S'|Y'^n}P_{S,W|Y^n}$.

For ease of notation, denote

$$
P_{\hat{Y}^n, \hat{Y}'^n|Y^n, Y'^n} := \mathbb{E}_{S',S,W}\left[P_{\hat{Y}|X,W}^{\otimes 2n}(\hat{Y}^n, \hat{Y}'^n|X^n, X'^n, W)\right],
$$

where $S', S, W \sim P_{S'|Y'^n}P_{S,W|Y^n}$. We have

$$
\begin{aligned}
&nh_D\left(\mathbb{E}_W[\mathcal{L}(W)], \mathbb{E}_{S,W}\left[\hat{\mathcal{L}}(S, W)\right]\right) \\
&\leqslant n\mathbb{E}_{P_{S'}P_{S,W}}\left[h_D\left(\hat{\mathcal{L}}(S', W), \hat{\mathcal{L}}(S, W)\right)\right] \\
&= n\mathbb{E}_{S',S,W}\left[h_D\left(\mathbb{E}_{P_{\hat{Y}|X,W}^{\otimes n}(\hat{Y}'^n|X'^n, W)}\left[\frac{1}{n}\sum_{i\in[n]}\mathbb{1}_{\{Y'_i \neq \hat{Y}'_i\}}\right], \mathbb{E}_{P_{\hat{Y}|X,W}^{\otimes n}(\hat{Y}^n|X^n, W)}\left[\frac{1}{n}\sum_{i\in[n]}\mathbb{1}_{\{Y_i \neq \hat{Y}_i\}}\right]\right)\right] \\
&\leqslant n\mathbb{E}_{P_{S'}P_{S,W}P_{\hat{Y}|X,W}^{\otimes 2n}(\hat{Y}^n, \hat{Y}'^n|X^n, X'^n, W)}\left[h_D\left(\frac{1}{n}\sum_{i\in[n]}\mathbb{1}_{\{Y'_i \neq \hat{Y}'_i\}}, \frac{1}{n}\sum_{i\in[n]}\mathbb{1}_{\{Y_i \neq \hat{Y}_i\}}\right)\right] \\
&\leqslant D_{KL}\left(\mu_Y^{\otimes 2n}P_{\hat{Y}^n, \hat{Y}'^n|Y^n, Y'^n}\middle\|\mu_Y^{\otimes 2n}\mathbf{Q}(\hat{Y}^n, \hat{Y}'^n|Y^n, Y'^n))\right) \\
&\quad + \log \mathbb{E}_{Y^n, Y'^n, \hat{Y}^n, \hat{Y}'^n \sim \mu_Y^{\otimes 2n}\mathbf{Q}(\hat{Y}^n, \hat{Y}'^n_i|Y^n, Y'^n)}\left[e^{nh_D\left(\frac{1}{n}\sum_{i\in[n]}\mathbb{1}_{\{Y'_i \neq \hat{Y}'_i\}}, \frac{1}{n}\sum_{i\in[n]}\mathbb{1}_{\{Y_i \neq \hat{Y}_i\}}\right)}\right]. \quad (30)
\end{aligned}
$$

Now, we compute the last term, which does not depend on $W$ anymore. Suppose that $\text{Unif}(2n)$ is a distribution that picks uniformly $n$ indices among indices $2n$ indices, i.e., the probability of each one is $\frac{1}{\binom{2n}{n}}$. We denote such indices by $\mathbf{T} = (T_1, \ldots, T_n)$, and the corresponding distribution by $\text{Unif}(2n)$. For a vector $Y^{2n}$ of length $2n$, we denote the elements corresponding to $n$ indices picked by $\mathbf{T}$ as $Y_{\mathbf{T}}^{2n} = (Y_{T_1}^{2n}, \ldots, Y_{T_n}^{2n})$. We denote by $\mathbf{T}^c = (T_1^c, \ldots, T_n^c)$ the other remaining $n$ elements in $1, \ldots, 2n$ that are not picked by $\mathbf{T}$. We denote $\mathfrak{Y}^{T^{n,c}} = (\mathfrak{Y}_{T_1^c}, \ldots, \mathfrak{Y}_{T_n^c})$. Then,

$$
\begin{aligned}
&\log \mathbb{E}_{Y^n, Y'^n, \hat{Y}^n, \hat{Y}'^n \sim \mu_Y^{\otimes 2n}\mathbf{Q}(\hat{Y}^n, \hat{Y}'^n_i|Y^n, Y'^n)}\left[e^{nh_D\left(\frac{1}{n}\sum_{i\in[n]}\mathbb{1}_{\{Y'_i \neq \hat{Y}'_i\}}, \frac{1}{n}\sum_{i\in[n]}\mathbb{1}_{\{Y_i \neq \hat{Y}_i\}}\right)}\right] \\
&= \log \mathbb{E}_{\mathfrak{Y}^{2n}, \hat{\mathfrak{Y}}^{2n}, \mathbf{T}, \sim \mu^{\otimes 2n}\mathbf{Q}(\hat{\mathfrak{Y}}^{2n}|\mathfrak{Y}^{2n})\text{Unif}(2n)}\left[e^{nh_D\left(\frac{1}{n}\sum_{i\in[n]}\mathbb{1}_{\{\mathfrak{Y}_{T_i^c} \neq \hat{\mathfrak{Y}}_{T_i^c}\}}, \frac{1}{n}\sum_{i\in[n]}\mathbb{1}_{\{\mathfrak{Y}_{T_i} \neq \hat{\mathfrak{Y}}_{T_i}\}}\right)}\right]. \\
&\hspace{14cm} (31)
\end{aligned}
$$

Let $V$ be a random variable indicating $V := \sum_{i \in [2n]} \mathbb{1}_{\{\mathfrak{Y}_i \neq \hat{\mathfrak{Y}}_i\}}$ in the sequence $\mathfrak{Y}^{2n}$. Then, we consider different cases for $V$ and show that

$$\mathbb{E}_{\mathbf{T} \sim \mathrm{Unif}(2n)}\left[ e^{nh_D\left(\frac{1}{n}\sum_{i \in [n]} \mathbb{1}_{\{\mathfrak{Y}_{T_i^c} \neq \hat{\mathfrak{Y}}_{T_i^c}\}}, \frac{1}{n}\sum_{i \in [n]} \mathbb{1}_{\{\mathfrak{Y}_{T_i} \neq \hat{\mathfrak{Y}}_{T_i}\}}\right)} \right] \leqslant n, \tag{32}$$

for $n \geqslant 10$. This completes the proof.

We use the following lemma repeatedly in the rest of the proof.

**Lemma 2** ([Gal68, Exercise 5.8.a]). *For $j \geqslant 1$ and $n - j \geqslant 1$, where $j, n \in \mathbb{N}$,*

$$\sqrt{\frac{n}{8j(n-j)}} \leqslant \binom{n}{j} e^{-nh_b(j/n)} \leqslant \sqrt{\frac{n}{2\pi j(n-j)}}. \tag{33}$$

Now,

**i. If $V \in [1, n]$:**

$$\sum_{j=0}^{V} e^{nh_D(j/n,(V-j)/n)} \frac{\binom{n}{j}\binom{n}{V-j}}{\binom{2n}{V}} = \frac{2}{\binom{2n}{V}} + \sum_{j=1}^{V-1} e^{nh_D(j/n,(V-j)/n)} \frac{\binom{n}{j}\binom{n}{V-j}}{\binom{2n}{V}}$$

$$\overset{(a)}{\leqslant} \frac{2}{\binom{2n}{V}} + \sum_{j=1}^{V-1} \frac{\sqrt{nV(2n-V)}}{\pi\sqrt{j(n-j)(V-j)(n-V+j)}}$$

$$\overset{(b)}{\leqslant} \frac{2}{\binom{2n}{V}} + \frac{1}{2\pi}\sqrt{nV(2n-V)} \sum_{j=1}^{V-1}\left(\frac{1}{j(n-j)} + \frac{1}{(V-j)(n-V+j)}\right)$$

$$= \frac{2}{\binom{2n}{V}} + \frac{1}{\pi}\sqrt{nV(2n-V)} \sum_{j=1}^{V-1} \frac{1}{j(n-j)}$$

$$= \frac{2}{\binom{2n}{V}} + \frac{1}{n\pi}\sqrt{nV(2n-V)} \sum_{j=1}^{V-1}\left(\frac{1}{j} + \frac{1}{n-j}\right)$$

$$\overset{(c)}{\leqslant} \frac{2}{\binom{2n}{V}} + \frac{2}{n\pi}\sqrt{n^3} \sum_{j=1}^{n-1} \frac{1}{j}$$

$$\leqslant \frac{2}{2n} + \frac{2\sqrt{n}}{\pi}(\log(n-1) + 0.58 + 1/(2n-2))$$

$$\overset{(d)}{\leqslant} n,$$

where $(a)$ is deduced using Lemma 2, $(b)$ is due to inequality $\frac{1}{\sqrt{xy}} \leqslant \frac{1}{2x} + \frac{1}{2y}$ for any $x, y > 0$, $(c)$ by the upper bound on the Harmonic series, and $(d)$ holds for $n \geqslant 2$.

**ii. If $V \in [n+1, 2n-1]$:**

$$\sum_{j=V-n}^{n} e^{nh_D(j/n,(V-j)/n)} \frac{\binom{n}{j}\binom{n}{V-j}}{\binom{2n}{V}}$$

$$= 2e^{nh_D(n/n,(V-n)/n)} \frac{\binom{n}{V-n}}{\binom{2n}{V}} + \sum_{j=V-n+1}^{n-1} e^{nh_D(j/n,(V-j)/n)} \frac{\binom{n}{j}\binom{n}{V-j}}{\binom{2n}{V}}$$

$$\overset{(a)}{\leqslant} 2\max\left(\max_{V \in [n+2, 2n-1]} \frac{\sqrt{2V}}{\sqrt{\pi(V-n)}}, e^{nh_D(1,1/n)} \frac{n}{\binom{2n}{n+1}}\right)$$

$$+ \sum_{j=1}^{V-1} \frac{\sqrt{nV(2n-V)}}{\pi\sqrt{j(n-j)(V-j)(n-V+j)}}$$

$$\overset{(b)}{\leqslant} 2\sqrt{\frac{(n+2)}{\pi}} + \frac{1}{2\pi}\sqrt{nV(2n-V)} \sum_{j=V-n+1}^{n-1}\left(\frac{1}{j(n-j)} + \frac{1}{(V-j)(n-V+j)}\right)$$

$$=2\sqrt{\frac{(n+2)}{\pi}}+\frac{1}{\pi}\sqrt{nV(2n-V)}\sum_{j=V-n+1}^{n-1}\frac{1}{j(n-j)}$$

$$=2\sqrt{\frac{(n+2)}{\pi}}+\frac{1}{n\pi}\sqrt{nV(2n-V)}\sum_{j=V-n+1}^{n-1}\left(\frac{1}{j}+\frac{1}{n-j}\right)$$

$$\overset{(c)}{\leqslant}2\sqrt{\frac{(n+2)}{\pi}}+\frac{2}{n\pi}\sqrt{n^3}\sum_{j=1}^{n-1}\frac{1}{j}$$

$$\leqslant2\sqrt{\frac{(n+2)}{\pi}}+\frac{2\sqrt{n}}{\pi}\left(\log(n-1)+0.58+1/(2n-2)\right)$$

$$\overset{(d)}{\leqslant}n,$$

where $(a)$ is deduced using Lemma 2, $(b)$ is due to inequality $\frac{1}{\sqrt{xy}}\leqslant\frac{1}{2x}+\frac{1}{2y}$ for any $x,y>0$ and by verifying the first term for $V=n+1$ numerically, $(c)$ by the upper bound on the Harmonic series, and $(d)$ holds for $n\geqslant10$.

**iii. If $V=2n$:** In this case $\sum_{j=V-n}^{n}e^{nh_D(j/n,(V-j)/n)}\frac{\binom{n}{j}\binom{n}{V-j}}{\binom{2n}{V}}=1$. $\qquad\square$

### E.5 Proof of Theorem 4

Let $\mathbf{Q}$ be a conditional prior over $U^{2n}$ given $X^{2n},Y^{2n}$ and $\hat{W}_e\in\mathcal{W}_e$, which is symmetric in the following sense: $\mathbf{Q}\Big((u_{\pi(1)},\ldots,u_{\pi(2n)})\big|(x_1,\ldots,x_{2n}),(y_1,\ldots,y_{2n}),\hat{w}_e\Big)$ remains the same for all permutations $\pi\colon[2n]\mapsto[2n]$ that preserves the label, *i.e.*, $y_{\pi(i)}=y_i$. Then, we show that for the $K$-classification learning task,

$$\mathbb{E}_{S,W}[\text{gen}(S,W)]\leqslant2\sqrt{\frac{2B+K+2}{n}}+\epsilon,$$

where

$$B:=\inf\mathbb{E}_{S,S',\hat{W}_e\sim P_{S,\hat{W}_e}P_{S'}}\Big[D_{KL}\Big(P_{U|X,\hat{W}_e}^{\otimes2n}(U^n,U'^n|X^n,X'^n,\hat{W}_e)\Big\|\mathbf{Q}\Big)\Big],$$

and the infimum is over all $P_{\hat{W}_e|S}$ such that for $\hat{W}=(\hat{W}_e,W_d)$,

$$\mathbb{E}_{P_{S,W}P_{\hat{W}_e|S}}\Big[\text{gen}(S,W)-\text{gen}(S,\hat{W})\Big]\leqslant\epsilon.$$

*Proof.* We prove the theorem for $\hat{W}=W$ and $\epsilon=0$. The general result follows by the distortion criterion and applying the theorem on $\text{gen}(S,\hat{W})$. We start the proof similar to the proof of Theorem 1; but with a different expansion of the probability distribution.

$$\lambda\mathbb{E}_{S,W}[\text{gen}(S,W)]$$

$$=\mathbb{E}_{S,S',W,U^n,U'^n,\hat{Y}^n,\hat{Y}'^n\sim P_{S'}P_{S,W}\nu_1}\left[\frac{\lambda}{n}\sum_{i\in[n]}\left(\mathbb{1}_{\{Y'_i\neq\hat{Y}'_i\}}-\mathbb{1}_{\{Y_i\neq\hat{Y}_i\}}\right)\right]$$

$$\leqslant D_{KL}\left(P_{S'}P_{S,W}P_{U|X,W_e}^{\otimes2n}(U^n,U'^n|X^n,X'^n,W_e)\Big\|P_{S'}P_{S,W}\mathbf{Q}(U^n,U'^n|S,S',W)\right)$$

$$+\log\mathbb{E}_{S,S',W,U^n,U'^n,\hat{Y}^n,\hat{Y}'^n\sim P_{S'}P_{S,W}\nu_2}\left[e^{\frac{\lambda}{n}\sum_{i\in[n]}\left(\mathbb{1}_{\{Y'_i\neq\hat{Y}'_i\}}-\mathbb{1}_{\{Y_i\neq\hat{Y}_i\}}\right)}\right],\tag{34}$$

where

$$\nu_1:=P_{U|X,W_e}^{\otimes2n}(U^n,U'^n|X^n,X'^n,W_e)P_{\hat{Y}|U,W_d}^{\otimes2n}(\hat{Y}^n,\hat{Y}'^n|U^n,U'^n,W_d),$$

$$\nu_2:=\mathbf{Q}(U^n,U'^n|S,S',W)P_{\hat{Y}|U,W_d}^{\otimes2n}(\hat{Y}^n,\hat{Y}'^n|U^n,U'^n,W_d).$$

Now, we bound the last term. Let

$$P_{\mathbf{Q}}\Big(\hat{Y}^n,\hat{Y}'^n|S,S',W\Big):=\mathbb{E}_{(U^n,U'^n)\sim\mathbf{Q}(U^n,U'^n|S,S',W)}\Big[P_{\hat{Y}|U,W_d}^{\otimes2n}(\hat{Y}^n,\hat{Y}'^n|U^n,U'^n,W_d)\Big].\tag{35}$$

Then, the last term in (34) can be written as

$$\log \mathbb{E}_{S,S',W,\hat{Y}^n,\hat{Y}'^n \sim P_{S'} P_{S,W} P_{\mathbf{Q}}\left(\hat{Y}^n,\hat{Y}'^n|S,S',W\right)} \left[ e^{\frac{\lambda}{n} \sum_{i \in [n]} \left( \mathbb{1}_{\{Y'_i \neq \hat{Y}_{1,i}\}} - \mathbb{1}_{\{Y_i \neq \hat{Y}_{2,i}\}} \right)} \right].$$

Note that since we have $P_{\hat{Y}|U,W_d,X,Y} = P_{\hat{Y}|U,W_d}$, it can be easily verified that the distribution $P_{\mathbf{Q}}\left(\hat{Y}^n, \hat{Y}'^n|S,S',W\right)$ is symmetric with respect to all permutations $\pi$ that preserve the labels of $Y$.

For each $s, s'$, let $f \colon [n] \to [n], i \in [n]$ and $k \colon [n] \to [n], i \in [n]$ be permutations of indices of $s$ and $s'$, respectively, where $Y_{f_i} = Y'_{k_i}$ for $i \leqslant T$ and $Y_{f_i} \neq Y'_{k_i}$ for $i > T$, and where $T$ equals $n - \frac{n}{2} \|\hat{p}_s - \hat{p}_{s'}\|_1$. Here, $\hat{p}_s$ and $\hat{p}_s$ are empirical distributions of $Y$ in $s$ and $s'$, respectively. We denote $(\hat{y}^n, \hat{y}'^n)$ by $\hat{y}^{2 \times n}$, where $\forall i \in [n], \hat{y}_{1,i} = \hat{y}'_i$ and $\hat{y}_{2,i} = \hat{y}_i$.

Consider the binary random variable $K_i$ taking value as $(2, f_i)$ or $(1, k_i)$ with probability $1/2$ (independent of other $j \neq i$). Denote the complementary choice as $K_i^c$, *i.e.*, $K_i^c = (2, f_i)$ iff $K_i = (1, k_i)$.

Then, due to the particular symmetry of $\mathbf{Q}$, and by using shorthand notation

$$\mathsf{P} := P_{S'} P_{S,W} P_{\mathbf{Q}}\left(\hat{Y}^n, \hat{Y}'^n|S,S',W\right),$$

we have

$$\mathbb{E}_{S,S',W,\hat{Y}^{2 \times n} \sim \mathsf{P}} \left[ e^{\frac{\lambda}{n} \sum_{i \in [n]} \left( \mathbb{1}_{\{Y'_{k_i} \neq \hat{Y}_{1,k_i}\}} - \mathbb{1}_{\{Y_{f_i} \neq \hat{Y}_{2,f_i}\}} \right)} \right]$$

$$= \mathbb{E}_{S,S',W,\hat{Y}^{2 \times n} \sim \mathsf{P}} \left[ \left\{ \prod_{i=1}^{T} e^{\frac{\lambda}{n} \left( \mathbb{1}_{\{Y'_{k_i} \neq \hat{Y}_{1,k_i}\}} - \mathbb{1}_{\{Y_{f_i} \neq \hat{Y}_{2,f_i}\}} \right)} \right\} \left\{ \prod_{i=T+1}^{n} e^{\frac{\lambda}{n} \left( \mathbb{1}_{\{Y'_{k_i} \neq \hat{Y}_{1,k_i}\}} - \mathbb{1}_{\{Y_{f_i} \neq \hat{Y}_{2,f_i}\}} \right)} \right\} \right]$$

$$\leqslant \mathbb{E}_{S,S',W,\hat{Y}^{2 \times n} \sim \mathsf{P}} \left[ \left\{ \prod_{i=1}^{T} e^{\frac{\lambda}{n} \left( \mathbb{1}_{\{Y_i \neq \hat{Y}_{1,k_i}\}} - \mathbb{1}_{\{Y_{f_i} \neq \hat{Y}_{2,f_i}\}} \right)} \right\} e^{\frac{\lambda(n-T)}{n}} \right]$$

$$= \mathbb{E}_{S,S',W,\hat{Y}^{2 \times n},K^n \sim \mathsf{P} \, \mathrm{Unif}((2,f_i),(1,k_i))^{\otimes n}} \left[ \left\{ \prod_{i=1}^{T} e^{\frac{\lambda}{n} \left( \mathbb{1}_{\{Y_i \neq \hat{Y}_{K_i}\}} - \mathbb{1}_{\{Y_{f_i} \neq \hat{Y}_{K_i^c}\}} \right)} \right\} e^{\frac{\lambda(n-T)}{n}} \right]$$

$$\leqslant \mathbb{E}_{S,S',W \sim P_{S'} P_{S,W}} \left[ \left( \frac{e^{\lambda/n} + e^{-\lambda/n}}{2} \right)^T e^{\frac{\lambda(n-T)}{n}} \right]$$

$$\leqslant \mathbb{E}_{S,S',W \sim P_{S'} P_{S,W}} \left[ e^{\frac{\lambda^2 T}{2n^2}} e^{\frac{\lambda(n-T)}{n}} \right]$$

$$\overset{(a)}{\leqslant} e^{\frac{\lambda^2}{2n}} \times \mathbb{E}_{S,S', \sim P_{S'} P_{S,W}} \left[ e^{\frac{\lambda}{2} \|\hat{p}_S - \hat{p}_{S'}\|_1} \right]$$

$$\leqslant \exp\left( \frac{\lambda^2}{2n} + \frac{K+2}{2} + \frac{3\lambda^2}{2n} \right)$$

$$\leqslant \exp\left( \frac{2\lambda^2}{n} + \frac{K+2}{2} \right),$$

where $(a)$ is deduced from Lemma 3, conditioned that $\lambda/n < 1.36$.

Now, combining this with (34), and letting $\lambda = \lambda^* = \sqrt{n(2B + K + 2)/4}$, the expectation of generalization error is upper bounded by

$$\mathbb{E}_{S,W}[\mathrm{gen}(S,W)] = \frac{2B + K + 2}{2\lambda} + \frac{2\lambda}{n}$$

$$\leqslant 2\sqrt{\frac{(2B + K + 2)}{n}},$$

if $\lambda^*/n < 1.36$. Note that if $1.36 < \lambda^*/n = \sqrt{\frac{(2B+K+2)}{4n}}$, then

$$2\sqrt{\frac{(2B + K + 2)}{n}} = 4(\lambda^*/n) > 1.$$

Since generalization error is always bounded by 1, hence, this bound always holds.

**Lemma 3.** *Suppose that $Y^n$ and $Y'^n$ are $2n$ i.i.d. instances of $Y \sim p \in [K]$. Then, if $\lambda/n < 0.68$, then,*

$$\log \mathbb{E}_{Y^n, Y'^n}\left[ e^{\lambda \|\hat{p}_{Y^n} - \hat{p}_{Y'^n}\|_1} \right] \leq \frac{K+2}{2} + \frac{6\lambda^2}{n}.$$

The lemma is proved in Appendix E.9 using results of [Dev83]. $\qquad\square$

## E.6 Proof of Theorem 5

Let $\mathbf{Q}$ be any fixed symmetric prior on $(\hat{Y}^n, \hat{Y}'^n)$ that could depend on $(X^n, Y^n, X'^n, Y'^n)$. We show that for any $\epsilon \in \mathbb{R}$ and $\delta \in \mathbb{R}^+$ with probability at least $(1 - \delta)$ over choices of $S$ and $S'$, we have that $\mathbb{E}_{W \sim Q}[\mathrm{gen}(S, W)]$ is upper bounded by

$$\sqrt{\frac{\log(2/\delta)}{2n}} + \inf \sqrt{\frac{\mathbb{E}_{\hat{W} \sim P_{\hat{W}|S}}\left[ D_{KL}\left( P_{\hat{Y}|X,\hat{W}}^{\otimes 2n}(\hat{Y}^n, \hat{Y}'^n | X^n, X'^n, \hat{W}) \middle\| \mathbf{Q} \right) \right] + \log(\sqrt{8n}/\delta)}{(2n-1)/4}} + \epsilon, \quad (36)$$

where the infimum is over all $P_{\hat{W}|S}$ that satisfy

$$\mathbb{E}_{P_{W|S} P_{\hat{W}|S}}\left[ \left| \left( \hat{\mathcal{L}}(S', W) - \hat{\mathcal{L}}(S, W) \right) - \left( \hat{\mathcal{L}}(S', \hat{W}) - \hat{\mathcal{L}}(S, \hat{W}) \right) \right| \right] \leq \epsilon/2. \quad (37)$$

*Proof.* Consider a distribution $P_{\hat{W}|S}$ that satisfies (37). Denote $\lambda^* = \frac{2n-1}{4}$ and

$$\Delta(S, \mathbf{Q}) := \sqrt{\frac{\mathbb{E}_{\hat{W} \sim P_{\hat{W}|S}}\left[ D_{KL}\left( P_{\hat{Y}|X,\hat{W}}^{\otimes 2n}(\hat{Y}^n, \hat{Y}'^n | X^n, X'^n, \hat{W}) \middle\| \mathbf{Q} \right) \right] + \log(\sqrt{8n}/\delta)}{\lambda^*}} + \epsilon.$$

Furthermore, we use the shorthand notations $\hat{p} := P_{\hat{Y}|X,\hat{W}}^{\otimes 2n}(\hat{Y}^n, \hat{Y}'^n | X^n, X'^n, \hat{W})$. Then,

$$\mathbb{P}_{S \sim \mu^{\otimes n}}\left( \mathbb{E}_{W \sim P_{W|S}}[\mathrm{gen}(S, W)] > \sqrt{\frac{\log(2/\delta)}{2n}} + \Delta(S, \mathbf{Q}) \right)$$

$$\overset{(a)}{\leq} \mathbb{P}_{(S,S') \sim \mu^{\otimes 2n}}\left( \mathbb{E}_{W \sim P_{W|S}}\left[ \hat{\mathcal{L}}(S', W) - \hat{\mathcal{L}}(S, W) \right] > \Delta(S, \mathbf{Q}) \right) + \delta/2$$

$$\overset{(b)}{\leq} \mathbb{P}_{(S,S') \sim \mu^{\otimes 2n}}\left( \mathbb{E}_{W \sim P_{W|S}}\left[ \lambda^* \left( \hat{\mathcal{L}}(S', W) - \hat{\mathcal{L}}(S, W) \right)^2 \right] > \lambda^* \Delta(S, \mathbf{Q})^2 \right) + \delta/2$$

$$\overset{(c)}{\leq} \mathbb{P}_{(S,S') \sim \mu^{\otimes 2n}}\left( \mathbb{E}_{\hat{W} \sim P_{\hat{W}|S}}\left[ \lambda^* \left( \hat{\mathcal{L}}(S', \hat{W}) - \hat{\mathcal{L}}(S, \hat{W}) \right)^2 \right] > \lambda^* \Delta(S, \mathbf{Q})^2 \right) + \delta/2$$

$$= \mathbb{P}_{(S,S') \sim \mu^{\otimes 2n}}\left( \lambda^* \mathbb{E}_{\hat{W}, \hat{Y}^n, \hat{Y}'^n \sim P_{\hat{W}|S}\hat{p}}\left[ \left( \frac{1}{n} \sum_{i \in [n]} \left( \mathbb{1}_{\{Y'_i \neq \hat{Y}'_i\}} - \mathbb{1}_{\{Y_i \neq \hat{Y}_i\}} \right) \right)^2 \right] \geq \lambda^* \Delta(S, \mathbf{Q})^2 \right)$$
$$+ \delta/2$$

$$\overset{(d)}{\leq} \mathbb{P}_{(S,S') \sim \mu^{\otimes 2n}}\left( D_{KL}\left( P_{\hat{W}|S}\hat{p} \middle\| P_{\hat{W}|S}\mathbf{Q} \right) \right. \quad (38)$$
$$\left. + \log \mathbb{E}_{\hat{Y}^n, \hat{Y}'^n \sim \mathbf{Q}}\left[ \exp\left( \lambda^* \left( \hat{\mathcal{L}}(S', Y'^n) - \hat{\mathcal{L}}(S, Y^n) \right) \right)^2 \right] \geq \lambda^* \Delta(S, \mathbf{Q})^2 \right)$$
$$+ \delta/2$$

$$\overset{(e)}{\leq} \mathbb{P}_{(S,S') \sim \mu^{\otimes 2n}}\left( \log \mathbb{E}_{\hat{Y}^n, \hat{Y}'^n \sim \mathbf{Q}}\left[ e^{\lambda^* \left( \frac{1}{n} \sum_{i \in [n]} \left( \mathbb{1}_{\{Y'_i \neq \hat{Y}'_i\}} - \mathbb{1}_{\{Y_i \neq \hat{Y}_i\}} \right) \right)^2} \right] \geq \right.$$
$$\left. \log \mathbb{E}_{S, S', \hat{Y}^n, \hat{Y}'^n \sim P_S P_{S'} \mathbf{Q}}\left[ e^{\lambda^* \left( \frac{1}{n} \sum_{i \in [n]} \left( \mathbb{1}_{\{Y'_i \neq \hat{Y}'_i\}} - \mathbb{1}_{\{Y_i \neq \hat{Y}_i\}} \right) \right)^2} \right] + \log(2/\delta) \right)$$
$$+ \delta/2$$

$$\overset{(f)}{\leq} \delta, \quad (39)$$

where $(a)$ holds by Hoeffding inequality and using the fact that for arbitrary random variables $U, V$ and constants $a, b \in \mathbb{R}$, $\mathbb{P}(U + V > a + b) \leqslant \mathbb{P}(U > a) + \mathbb{P}(V > b)$, $(b)$ by applying the Jensen inequality on the convex function $f(x) = x^2$, $(c)$ by using the distortion function (37) and since the loss is bounded by one, $(d)$ by using the Donsker-Varadhan's variational representation lemma, $(e)$ is shown in the following, and $(f)$ by Markov inequality.

Hence, it remains to show the step $(e)$. To this end, it is sufficient upper bound

$$\log \mathbb{E}_{S, S', \hat{Y}^n, \hat{Y}'^n \sim P_S P_{S'} \mathbf{Q}} \left[ e^{\lambda^* \left( \frac{1}{n} \sum_{i \in [n]} \left( \mathbb{1}_{\{Y'_i \neq \hat{Y}'_i\}} - \mathbb{1}_{\{Y_i \neq \hat{Y}_i\}} \right) \right)^2} \right]$$

by $\log(\sqrt{2n})$. Note that this terms equals

$$\log \mathbb{E}_{Y^n, Y'^n, \hat{Y}^n, \hat{Y}'^n \sim P_Y^{\otimes 2n} \mathbf{Q}_1} \left[ e^{\lambda^* \left( \frac{1}{n} \sum_{i \in [n]} \left( \mathbb{1}_{\{Y'_i \neq \hat{Y}'_i\}} - \mathbb{1}_{\{Y_i \neq \hat{Y}_i\}} \right) \right)^2} \right],$$

where $\mathbf{Q}_1$ is equal to $\mathbb{E}_{P_{X^n | Y^n} P_{X'^n | Y'^n}} \left[ \mathbf{Q}(\hat{Y}^n, \hat{Y}'^n_i | X^n, Y^n, X'^n, Y'^n) \right]$. Now,

$$\mathbb{E}_{Y^n, Y'^n, \hat{Y}^n, \hat{Y}'^n \sim \mu_Y^{\otimes 2n} \mathbf{Q}_1(\hat{Y}^n, \hat{Y}'^n_i | Y^n, Y'^n)} \left[ e^{\lambda^* \left( \frac{1}{n} \sum_{i \in [n]} \left( \mathbb{1}_{\{Y'_i \neq \hat{Y}'_i\}} - \mathbb{1}_{\{Y_i \neq \hat{Y}_i\}} \right) \right)^2} \right]$$

$$\overset{(a)}{=} \mathbb{E}_{\mathfrak{Y}^{n \times 2}, \hat{\mathfrak{Y}}^{n \times 2}, K^n \sim P_{\hat{Y}}^{\otimes 2n} \mathbf{Q}_1(\hat{\mathfrak{Y}}^{n \times 2} | \mathfrak{Y}^{n \times 2}) \, \text{Unif}(1,2)^{\otimes n}} \left[ e^{\lambda^* \left( \frac{1}{n} \sum_{i \in [n]} \left( \mathbb{1}_{\{\mathfrak{Y}_{K_i, i} \neq \hat{\mathfrak{Y}}_{K_i, i}\}} - \mathbb{1}_{\{\mathfrak{Y}_{K^c_i, i} \neq \hat{\mathfrak{Y}}_{K^c_i, i}\}} \right) \right)^2} \right]$$

$$= \mathbb{E}_{\mathfrak{Y}^{n \times 2}, \hat{\mathfrak{Y}}^{n \times 2} \sim P_{\hat{Y}}^{\otimes 2n} \mathbf{Q}_1(\hat{\mathfrak{Y}}^{n \times 2} | \mathfrak{Y}^{n \times 2})} \mathbb{E}_{K^n \sim \text{Unif}(1,2)^{\otimes n}} \left[ e^{\lambda^* \left( \frac{1}{n} \sum_{i \in [n]} \left( \mathbb{1}_{\{\mathfrak{Y}_{K_i, i} \neq \hat{\mathfrak{Y}}_{K_i, i}\}} - \mathbb{1}_{\{\mathfrak{Y}_{K^c_i, i} \neq \hat{\mathfrak{Y}}_{K^c_i, i}\}} \right) \right)^2} \right]$$

$$\overset{(b)}{\leqslant} \sqrt{2n}, \tag{40}$$

where $(a)$ is concluded by symmetry of $\mathbf{Q}_1$ and $(b)$ is deduced since

$$\frac{1}{n} \sum_{i \in [n]} \left( \mathbb{1}_{\{\mathfrak{Y}_{K_i, i} \neq \hat{\mathfrak{Y}}_{K_i, i}\}} - \mathbb{1}_{\{\mathfrak{Y}_{K^c_i, i} \neq \hat{\mathfrak{Y}}_{K^c_i, i}\}} \right),$$

is $1/\sqrt{n}$-subgaussian process and hence

$$\mathbb{E}_{K^n \sim \text{Unif}(1,2)^{\otimes n}} \left[ e^{\left( \frac{1}{n} \sum_{i \in [n]} \left( \mathbb{1}_{\{\mathfrak{Y}_{K_i, i} \neq \hat{\mathfrak{Y}}_{K_i, i}\}} - \mathbb{1}_{\{\mathfrak{Y}_{K^c_i, i} \neq \hat{\mathfrak{Y}}_{K^c_i, i}\}} \right) \right)^2 / (4/(2n-1))} \right] \leqslant \sqrt{2n},$$

due to [Wai19, Theorem 2.6.IV.]. This completes the proof. $\qquad \square$

## E.7 Proof of Theorem 6

Let $\mathbf{Q}$ be any fixed symmetric prior on $(\hat{Y}^n, \hat{Y}'^n)$ that could depend on $(X^n, Y^n, X'^n, Y'^n)$. Then, for any $\delta \in \mathbb{R}^+$, we show that with probability at least $(1 - \delta)$ over choice of $(S, S', W) \sim P_{S'} P_{S,W}$,

$$n h_D \left( \hat{\mathcal{L}}(S', W), \hat{\mathcal{L}}(S, W) \right) \leqslant D_{KL} \left( P_{\hat{Y} | X, W}^{\otimes 2n} (\hat{Y}^n, \hat{Y}'^n | X^n, X'^n, W) \middle\| \mathbf{Q} \right) + \log(n/\delta).$$

*Proof.* The proof is a combination of proofs of Theorems 3 and 5. Denote the RHS of the bound in the theorem as $\Delta(S, S', W)$.

First, note that

$$nh_D\Big(\hat{\mathcal{L}}(S',W),\hat{\mathcal{L}}(S,W)\Big)$$

$$=nh_D\left(\mathbb{E}_{\hat{Y}'^n\sim P_{\hat{Y}|X,W}^{\otimes n}(\hat{Y}'^n|X'^n,W)}\left[\frac{1}{n}\sum_{i\in[n]}\mathbb{1}_{\{Y_i'\neq\hat{Y}_i'\}}\right],\mathbb{E}_{\hat{Y}^n\sim P_{\hat{Y}|X,W}^{\otimes n}(\hat{Y}^n|X^n,W)}\left[\frac{1}{n}\sum_{i\in[n]}\mathbb{1}_{\{Y_i\neq\hat{Y}_i\}}\right]\right)$$

$$\leqslant n\mathbb{E}_{\hat{Y}^n,\hat{Y}'^n\sim P_{\hat{Y}|X,W}^{\otimes 2n}(\hat{Y}^n,\hat{Y}'^n|X^n,X'^n,W)}\left[h_D\left(\frac{1}{n}\sum_{i\in[n]}\mathbb{1}_{\{Y_i'\neq\hat{Y}_i'\}},\frac{1}{n}\sum_{i\in[n]}\mathbb{1}_{\{Y_i\neq\hat{Y}_i\}}\right)\right]$$

$$\leqslant D_{KL}\left(P_{\hat{Y}|X,W}^{\otimes 2n}(\hat{Y}^n,\hat{Y}'^n|X^n,X'^n,W)\Big\|\mathbf{Q}(\hat{Y}^n,\hat{Y}'^n|X^n,Y^n,X'^n,Y'^n)\right)$$

$$+\log\mathbb{E}_{\hat{Y}^n,\hat{Y}'^n\sim\mathbf{Q}(\hat{Y}^n,\hat{Y}_i'^n|X^n,Y^n,X'^n,Y'^n)}\left[e^{nh_D\left(\frac{1}{n}\sum_{i\in[n]}\mathbb{1}_{\{Y_i'\neq\hat{Y}_i'\}},\frac{1}{n}\sum_{i\in[n]}\mathbb{1}_{\{Y_i\neq\hat{Y}_i\}}\right)}\right]. \tag{41}$$

Hence,

$$\mathbb{P}_{S,S',W}\Big(nh_D\Big(\hat{\mathcal{L}}(S',W),\hat{\mathcal{L}}(S,W)\Big)>\Delta(S,S',W)\Big)$$

$$\overset{(a)}{\leqslant}\mathbb{P}_{Y^n,Y'^n\sim\mu^{\otimes 2n}}\left(\log\mathbb{E}_{\hat{Y}^n,\hat{Y}'^n\sim\mathbf{Q}}\left[e^{nh_D\left(\frac{1}{n}\sum_{i\in[n]}\mathbb{1}_{\{Y_i'\neq\hat{Y}_i'\}},\frac{1}{n}\sum_{i\in[n]}\mathbb{1}_{\{Y_i\neq\hat{Y}_i\}}\right)}\right]>\log(n/\delta)\right)$$

$$\overset{(b)}{\leqslant}\mathbb{P}_{Y^n,Y'^n\sim\mu^{\otimes 2n}}\left(\log\mathbb{E}_{\hat{Y}^n,\hat{Y}'^n\sim\mathbf{Q}}\left[e^{nh_D\left(\frac{1}{n}\sum_{i\in[n]}\mathbb{1}_{\{Y_i'\neq\hat{Y}_i'\}},\frac{1}{n}\sum_{i\in[n]}\mathbb{1}_{\{Y_i\neq\hat{Y}_i\}}\right)}\right]>\right.$$

$$\left.\log\mathbb{E}_{Y^n,Y'^n,\hat{Y}^n,\hat{Y}'^n\sim\mu_Y^{\otimes 2n}\mathbf{Q}}\left[e^{nh_D\left(\frac{1}{n}\sum_{i\in[n]}\mathbb{1}_{\{Y_i'\neq\hat{Y}_i'\}},\frac{1}{n}\sum_{i\in[n]}\mathbb{1}_{\{Y_i\neq\hat{Y}_i\}}\right)}\right]+\log(1/\delta)\right)$$

$$\overset{(c)}{\leqslant}\delta,$$

where $(a)$ follows by (41), $(b)$ holds since

$$\log\mathbb{E}_{Y^n,Y'^n,\hat{Y}^n,\hat{Y}'^n\sim\mu_Y^{\otimes 2n}\mathbf{Q}}\left[e^{nh_D\left(\frac{1}{n}\sum_{i\in[n]}\mathbb{1}_{\{Y_i'\neq\hat{Y}_i'\}},\frac{1}{n}\sum_{i\in[n]}\mathbb{1}_{\{Y_i\neq\hat{Y}_i\}}\right)}\right]\leqslant\log(n),$$

by the proof of Theorem 3, and $(c)$ is derived using Markov inequality. This completes the proof. $\square$

### E.8 Proof of Theorem 7

Let $\mathbf{Q}$ be a type-III symmetric conditional prior over $U^{2n}$ given $X^{2n},Y^{2n}$ and $W_e\in\mathcal{W}_e$, namely, $\mathbf{Q}\Big((u_{\pi(1)},\ldots,u_{\pi(2n)})\big|(x_1,\ldots,x_{2n}),(y_1,\ldots,y_{2n}),w\Big)$ remains the same for all permutations $\pi:[2n]\mapsto[2n]$ that preserves the label, *i.e.,* $y_{\pi(i)}=y_i$ for $i\in[n]$. Then, for any $\lambda\in\mathbb{R}^+$, we show that for $K$-classification learning task, with probability at least $(1-\delta)$ over $(S,S',W)\sim P_{S'}P_{S,W}$,

$$\hat{\mathcal{L}}(S',W)-\hat{\mathcal{L}}(S,W)\leqslant\frac{D_{KL}\Big(P_{U|X,W_e}^{\otimes 2n}(U^n,U'^n|X^n,X'^n,W_e)\Big\|\mathbf{Q}\Big)+(K+2)/2+\log(1/\delta)}{\lambda}+\frac{2\lambda}{n}.$$

The poof of the second part follows trivially using Hoeffding's inequality.

*Proof.* Note that whenever $\lambda/n\geqslant 1.36$, the bound trivially holds. Hence, assume that $\lambda/n<1.36$. Denote the RHS of the bound of Part i. as $\Delta(S,S',W)$. We use the shorthand notation $p_1:=P_{U|X,W_e}^{\otimes 2n}(U^n,U'^n|X^n,X'^n,W_e)$ and $p_2:=P_{\hat{Y}|U,W_d}^{\otimes 2n}(\hat{Y}^n,\hat{Y}'^n|U^n,U'^n,W_d)$.

Note that $(S,S',W)\sim P_{S'}P_{S,W}$. Now,

$$\mathbb{P}_{S,S',W}\Big(\hat{\mathcal{L}}(S',W)-\hat{\mathcal{L}}(S,W)>\Delta(S,S',W)\Big)$$

$$=\mathbb{P}_{S,S',W}\left(\lambda\mathbb{E}_{p_1p_2}\left[\frac{1}{n}\sum_{i\in[n]}\Big(\mathbb{1}_{\{Y_i'\neq\hat{Y}_i'\}}-\mathbb{1}_{\{Y_i\neq\hat{Y}_i\}}\Big)\right]\geqslant\lambda\Delta(S,S',W)\right)$$

$$\overset{(a)}{\leqslant}\mathbb{P}_{S,S',W}\left(D_{KL}\Big(p_1p_2\Big\|\mathbf{Q}p_2\Big)\right) \tag{42}$$

$$+ \log \mathbb{E}_{\mathbf{Q}p_2}\Big[\exp\Big(\lambda\Big(\hat{\mathcal{L}}(S', Y'^n) - \hat{\mathcal{L}}(S, Y^n)\Big)\Big)\Big] \geqslant \lambda\Delta(S, S', W)\Big)$$

$$\overset{(c)}{\leqslant} \mathbb{P}_{S,S',W}\left(\log \mathbb{E}_{\mathbf{Q}p_2}\left[e^{\frac{\lambda}{n}\sum_{i\in[n]}\left(\mathbb{1}_{\{Y'_i \neq \hat{Y}'_i\}} - \mathbb{1}_{\{Y_i \neq \hat{Y}_i\}}\right)}\right] \geqslant\right.$$

$$\left.\log \mathbb{E}_{P_{S'}P_{S,W}\mathbf{Q}p_2}\left[e^{\frac{\lambda}{n}\sum_{i\in[n]}\left(\mathbb{1}_{\{Y'_i \neq \hat{Y}'_i\}} - \mathbb{1}_{\{Y_i \neq \hat{Y}_i\}}\right)}\right] + \log(1/\delta)\right)$$

$$\overset{(d)}{\leqslant} \delta, \tag{43}$$

where $(a)$ by using the Donsker-Varadhan's variational representation lemma, $(b)$ since due to proof of Theorem 4 (in Appendix E.5),

$$\log \mathbb{E}_{P_{S'}P_{S,W}\mathbf{Q}p_2}\left[e^{\frac{\lambda}{n}\sum_{i\in[n]}\left(\mathbb{1}_{\{Y'_i \neq \hat{Y}'_i\}} - \mathbb{1}_{\{Y_i \neq \hat{Y}_i\}}\right)}\right] \leqslant \frac{2\lambda^2}{n} + \frac{K+2}{2},$$

when $\lambda/n < 1.36$, and $(d)$ by Markov inequality. This completes the proof. $\qquad\square$

### E.9   Proof of Lemma 3

*Proof.* The proof takes its main elements from [Dev83]. Let $Y_1, Y_2, \ldots$ and $Y'_1, Y'_2, \ldots$ be i.i.d. realizations of $Y \sim p \in [K]$. Denote $P_Y(k) = p_k$ for $k \in [K]$. Let $N$ and $N'$ be two independent Poisson random variables with mean $n$. Let $U_i$ and $U'_i$, $i \in [K]$, be number of occurrences of value $k$ among $Y^{[n]}$ and $Y'^{[n]}$, respectively. Moreover, let $V_i$ and $V'_i$, $i \in [K]$, be number of occurrences of value $k$ among $Y^{[N]}$ and $Y'^{[N']}$, respectively. As shown in [Dev83], $V_1, \ldots, V_K$ and $V'_1, \ldots, V'_K$ are independent Poisson random variables with means $\mathbb{E}[V_k] = \mathbb{E}[V'_k] = np_k$, for $k \in [K]$. In addition, $(U_1, \ldots, U_K)$ and $(U'_1, \ldots, U'_K)$ are two independent multinomial $(n, p_1, \ldots, p_K)$ random vectors. Then,

$$\mathbb{E}\left[e^{\lambda\|\hat{p}_{Y^n} - \hat{p}_{Y'^n}\|_1}\right] \leqslant \mathbb{E}\left[e^{\frac{\lambda}{n}\left(\sum_{i=1}^K |U_i - V_i - (U'_i - V'_i)|\right)} e^{\frac{\lambda}{n}\left(\sum_{i=1}^K |V_i - V'_i|\right)}\right]$$

$$\overset{(a)}{\leqslant} \sqrt{\mathbb{E}\left[e^{\frac{2\lambda}{n}\left(\sum_{i=1}^K |U_i - V_i - (U'_i - V'_i)|\right)}\right] \mathbb{E}\left[e^{\frac{2\lambda}{n}\left(\sum_{i=1}^K |V_i - V'_i|\right)}\right]}$$

$$\leqslant \sqrt{\mathbb{E}\left[e^{\frac{2\lambda}{n}\left(\sum_{i=1}^K |U_i - V_i| + |U'_i - V'_i|\right)}\right] \mathbb{E}\left[e^{\frac{2\lambda}{n}\left(\sum_{i=1}^K |V_i - V'_i|\right)}\right]}$$

$$\leqslant \sqrt{\mathbb{E}\left[e^{\frac{2\lambda}{n}\left(|N - n| + |N' - n|\right)}\right] \mathbb{E}\left[e^{\frac{2\lambda}{n}\left(\sum_{i=1}^K |V_i - V'_i|\right)}\right]}, \tag{44}$$

where $(a)$ is due to Cauchy-Schwarz inequality.

Recall that for random variable $X$ having the Poisson distribution with mean $\eta$, we have $\mathbb{E}\left[e^{tX}\right] = e^{\eta(e^t - 1)}$. Now,

$$\mathbb{E}\left[e^{\frac{2\lambda}{n}\left(|N-n| + |N'-n|\right)}\right] = \mathbb{E}\left[e^{\frac{2\lambda}{n}\left(|N-n|\right)}\right]\mathbb{E}\left[e^{\frac{2\lambda}{n}\left(|N'-n|\right)}\right]$$

$$\leqslant \mathbb{E}\left[e^{\frac{2\lambda}{n}\left(N-n\right)} + e^{\frac{2\lambda}{n}\left(n-N\right)}\right]\mathbb{E}\left[e^{\frac{2\lambda}{n}\left(N'-n\right)} + e^{\frac{2\lambda}{n}\left(n-N'\right)}\right]$$

$$\leqslant 4e^{2n(e^{\frac{2\lambda}{n}} - 1 - \frac{2\lambda}{n})}, \tag{45}$$

and

$$\mathbb{E}\left[e^{\frac{2\lambda}{n}\left(\sum_{i=1}^K |V_i - V'_i|\right)}\right] \leqslant \prod_{i=1}^K \left(\mathbb{E}\left[e^{\frac{2\lambda}{n}\left(V'_i - V_i\right)} + e^{\frac{2\lambda}{n}\left(\hat{V}'_i - V'_i\right)}\right]\right)$$

$$= 2^K \prod_{i=1}^K e^{np_i(e^{\frac{2\lambda}{n}} - 1 + e^{-\frac{2\lambda}{n}} - 1)}$$

$$= 2^K e^{n(e^{\frac{2\lambda}{n}} - 1 + e^{-\frac{2\lambda}{n}} - 1)}, \tag{46}$$

Combining (44),(45), and (45) yield

$$\log \mathbb{E}\left[e^{\lambda \|\hat{p}_S - \hat{p}_{S'}\|_1}\right] \overset{(a)}{\leqslant} \frac{K+2}{2} + n\left(e^{\frac{2\lambda}{n}} - 1 - \frac{2\lambda}{n} + \frac{1}{2}e^{\frac{2\lambda}{n}} + \frac{1}{2}e^{-\frac{2\lambda}{n}} - 1\right)$$

$$= \frac{K+2}{2} + n\left(\frac{1}{1.2}\left(\frac{2\lambda}{n}\right)^2 + \frac{7}{12}\left(\frac{2\lambda}{n}\right)^2\right)$$

$$\leqslant \frac{K+2}{2} + \frac{6\lambda^2}{n},$$

where $(a)$ holds due to two inequalities that can be verified numerically: i) for $x < 1.36$, $e^x < 1 + x + \frac{x^2}{1.2}$ and ii) for $x < 1.3$, $\frac{e^x + e^{-x}}{2} < 1 + \frac{7}{12}x^2$.

$\square$

