# OpenReview forum: "Minimum Description Length and Generalization Guarantees for Representation Learning"
_NeurIPS.cc/2023/Conference — NeurIPS 2023 poster_

### Official Review · Reviewer_Lcin · 2023-07-03

**Soundness:** 3 good
**Presentation:** 3 good
**Contribution:** 3 good
**Rating:** 7
**Confidence:** 4

**Summary:**

This paper obtains new generalization bounds of the conditional mutual information-flavor (Steinke and Zakynthinou, 2020), where the bounds depend only on the predictions that the algorithm induces on the supersample. These bounds are phrased more generally in terms of a KL divergence with symmetric priors, analogous to the almost exchangeable priors of Audibert (2004), and are placed in a rate-distortion theoretic perspective. This is further extended to lossy compression and tail bounds. Tighter bounds in terms of a function of the binary entropy are derived, which appear similar to classical PAC-Bayes bounds with the binary KL divergence in the style of Maurer (2004). Throughout, this is related to the representation in the style of the information bottleneck, leading to the main result of the paper: a generalization bound for representation learning algorithms which depends only on the encoder part, allowing for the decoder part to be chosen freely to minimize the training loss without affecting the bound. This allows for priors that are in some senses data-dependent, and the potential benefit of this is demonstrated numerically.

**Post-Rebuttal edit**: I read the author rebuttal, which addressed my questions, in particular regarding related literature. I consequently updated the rating (as stated below).

**Strengths:**

— The new results and their relevance for representation learning are well-established.

— The theoretical analysis is thorough, covering several possible variants and tightenings of the results.

— The presentation is overall very clear and pedagogical, including a very thorough discussion of some related prior work.

— The numerical evaluation nicely illustrates the potential benefit of the proposed methods.

**Weaknesses:**

— There are some missing connections to previous work (see questions).

— A few of the results appear to be stated incorrectly, and some notation seems undefined (see questions).

**Questions:**

— There appears to be a clear connection to the frameworks of functional and evaluated CMI (f-CMI, e-CMI) of, e.g., Harutyunyan et al (2021) and Hellström and Durisi (2022). Is there a clear distinction between these bounds and your Theorem 1, other than the use of an auxiliary prior (the possibility of which is also noted in the mentioned works)?

— The concept of symmetric priors seems identical to the almost exchangeable priors discussed by Audibert (2004). Is there any difference with these?

— The bounds in terms of the function $h_D$ appear closely related to bounds with the binary KL divergence on the left-hand side, e.g. Maurer (2004) and Seeger (2002) — the properties of the function appear to be very similar. Is there a difference or are the two approaches equivalent?

— Proposition 1: The left-hand side here is a single-draw result, which typically does not lead to bounds in terms of the KL divergence, but instead some form of information density or log-likelihood ratio. This is also possible to derive from your results, but the derivations have to be slightly altered. In the proof of Theorem 4, you instead consider a “PAC-Bayesian” bound which is averaged over the posterior, which naturally leads to bounds in terms of the KL divergence. I assume this is what you meant to write in Proposition 1 as well. As noted by Grunwald et al (2023), such PAC-Bayesian bounds follow immediately from the same derivation as average generalization bounds if they are based on “exponential inequalities” (as yours are). This can be extended to single-draw bounds as well, under more stringent absolute continuity assumptions.

Minor:

Abstract: IB undefined

Intro: $\text{gen}(W,S)$ and the corresponding notation for population and training losses (from e.g. Proposition 1) appear to be undefined

Quotation mark lines 34, 234, 33

Line 50: A sentence with only citations.

Line 131: KL divergence notation is flipped in LHS of definition

Fig. 1 is not on the same page as it is discussed

Margin overflow in Eq. (3), lines 579-581, more in appendix

Line 242: $\tau*$ instead of $\tau^*$

Lines 266-272: inconsistent conventions for fractions. Can some expressions be moved to equation environments? Maybe this intuition can be expressed more informally and the math can be moved to the appendix where there is space

I assume $[a,b]=\{a,a+1,\dots,b\}$ for integers in some expressions, but I do not think this is defined and it conflicts with the standard use for closed intervals

Theorem 3: The prior is called $P$ in the theorem statement but $Q$ in the equation.

Line 329, 686: Incomplete sentence

References:

Audibert, “A better variance control for PAC-Bayesian classification”, 2004

Maurer, “A Note on the PAC Bayesian Theorem”, 2004

Harutyunyan et al, "Information-theoretic generalization bounds for black-box learning algorithms”, 2021

Hellstrom and Durisi, “A New Family of Generalization Bounds Using Samplewise Evaluated CMI”, 2022

Seeger, “PAC-Bayesian Generalisation Error Bounds for Gaussian Process Classification”, 2002

Grunwald et al, “Exponential Stochastic Inequality”, 2023

**Limitations:**

Perhaps the discussion of limitations can be expanded, but it is not crucial.

---

> ### Author Rebuttal · Authors · 2023-08-09
>
> We thank the reviewer for their careful reading of the paper, their interest in our work, and their constructive comments.
>
> **Regarding CMI:** We thank the reviewer for bringing our attention to CMI-based works and in particular the f-CMI-based follow-ups. We missed this connection, as the original CMI work is mainly focused on the complexity of the hypothesis space. However, thanks to the feedbacks, we understood that in particular the f-CMI framework has a close connection to Section 2 of our work, and therefore it deserved to be mentioned and discussed.
>
> More precisely, Theorem 1 (with proper choice of the prior) gives a slightly stronger result than the f-CMI framework with $f$ being the predicted variables and $u=[n]$. The reason is that in Theorem 1, the expectation with respect to $Y^n,Y^{'n}$ of KL-divergence (of conditional distributions given $Y^n,Y^{'n}$) is considered, while f-CMI considers the expectation with respect to $X^n,Y^{n},X^{'n},Y^{'n}$ and KL-divergence is a convex function. There is also another small difference: in Theorem 1, we take the expectation w.r.t. $W$ (somehow equivalent to $R$ in the f-CMI setup) inside the KL-divergence term. We will highlight this connection after Theorem 1 by properly referencing the mentioned works and by mentioning that for the in-expectation result, the bounds can be potentially improved by considering $u=[m]$, where $1\leq m <n$. In fact, interestingly, our approach, which is based on the framework of [BL03], shows the connection between this approach and CMI also.
>
> However, other results (including those based on the function $h_D$ and tail bounds) seemingly cannot be directly obtained using the existing f-CMI framework. Besides, in the f-CMI framework, we need the loss to be computable using $f$ and the label $Y$. Hence, this framework cannot be applied to the setup of Section 3 and thus, cannot obtain the main result of the paper (Theorem 5).
>
> **Regarding almost exchangeable priors:** We thank the reviewer for pointing out this valuable work, which we have not been aware of. Indeed, Definition 1.1 therein is applicable in our Theorem 1 and, in fact, we used a nearly identical definition in our preliminary version. However while this more relaxed condition is sufficient for Theorem 1, it is not the case for Theorem 2. In Theorem 1, similarly to the f-CMI approach of [Harutyunyan et al.] (and CMI-approaches in general), we only consider shuffling independently samples in $n$ pairs $(Z_{i,1},Z_{i,2})$ (and their corresponding predictions) of a "super-sample" $\{Z_{1,1},Z_{1,2},\ldots,Z_{n,1},Z_{n,2}\}$ in order to assign randomly one sample to the training set and the other one to the test set. In Theorem 2, which is indeed inspired by [BL03], we consider shuffling the whole super-sample to assign $n$ instances to the training set and the others to the test set. This requires a different definition for the symmetry of the priors. This is why we opted for a unique definition of symmetry for both theorems. We will add these discussions. We note that the symmetry definition in Section 3 is also a bit different as it requires the invariance only under the permutations of latent variables that preserve the associated labeling.
>
> **Regarding $h_{D}$:** The function $h_{D}$ indeed resembles the pointed binary KL divergence function. As the reviewer is definitely aware, such bounds are established in terms of KL-divergence of priors and posteriors on the hypothesis space $D_{KL}(P_{W|S}\\|Q_W)$ that captures the complexity of the hypothesis space. To use this binary KL-divergence function for relating the generalization error to the complexity of the labels/latent variables, we need to consider KL-divergences over the space of labels/latent variables. This turns out to be difficult to achieve, among others, since the cumulant-generating function (equation (23)) appearing in this setup is different than classical PAC-Bayes bounds, and hence the proof does not follow straightforwardly. Thus, we cannot say these functions and approaches are the same, despite the similarity of their properties. We refer the reviewer to line 268 for the intuition behind $h_D$, inspired by [BL03].
>
> **Regarding Proposition 1:** We understand the point of the reviewer. This is the case for the classical PAC-Bayes bounds, where the posterior $P_{W|S}$ and prior $Q_W$ are defined over the hypothesis space. Hence, by considering an instance $(S,W) \sim \mu^{\otimes n} P_{W|S}$, the bound contain the information density $\log\frac{\mathrm{d} P_{W|S}}{\mathrm{d} Q_{W}}$. In our work, however, the prior and posterior, for example in the case of Proposition 1, are defined over the predicted sample space, and hence for any given $(S,S',W) \sim \mu^{\otimes 2n}P_{W|S}$, the KL-divergence term $D_{KL}( P_{\hat{Y}|X,W}^{\otimes 2n}(\hat{Y}^n, \hat{Y}^{\prime n}|X^n,X^{\prime n},W) \\| \mathbf{Q} )$ appears. This is because in our definition of the loss function (after line 156) we already considered the expectation with respect to the prediction, and hence LHS of the bound of Proposition 1 already contains an expectation term.
>
> In Theorem 4, there exists an extra expectation w.r.t $\hat{W}$ for the exactly same reason that the reviewer suggested: because we considered the average over the posterior. We will discuss this difference in our PAC-Bayes bounds w.r.t. the classical PAC-Bayes bounds in terms of the complexity of the hypothesis space.
>
> Moreover, using the provided feedback, we discuss the potential of considering further a "disintegrated PAC-Bayes" bound that not only consider one draw of $(S,S',W) \sim \mu^{\otimes 2n}P_{W|S}$, but also the "one-shot" prediction of the label. For this case, one should remove the expectation term in the loss function.
>
> **Minor Comments:** Due to space limitations, we will apply the minor comments directly. Thanks for such a careful reading. We just precise that by the interval $[a,b]$ in line 272, we meant the closed interval.

---

> > ### Comment · Reviewer_Lcin · 2023-08-11
> >
> > Thank you for your clarifications and detailed response. With the added discussion of related work, I believe that the paper is improved. I have increased my rating.

---

### Official Review · Reviewer_D6jh · 2023-07-05

**Soundness:** 4 excellent
**Presentation:** 3 good
**Contribution:** 4 excellent
**Rating:** 7
**Confidence:** 2

**Summary:**

The paper advances state-of-the-art on information theoretic generalization guarantees for a class of representational learning problems. The contribution is positioned in the framework of information bottleneck approach and builds on the classical work of Blum and Langford for PAC-MDL bounds. The main result is on non-trivial upper bounds on generalization error of a representational learning algorithm in terms of so-called minimum description length of the label. The result is expressed in terms of KL-divergence of a vector of labels and any arbitrary symmetric prior, and resolves the issues related to use of mutual information measure. The aspects related to block coding and lossy compression are new to this type of results.

**Strengths:**

The writing is adequately clear both in terms of technical precision and intuitive interpretations, where appropriate. The problem is important and the proposed results advance state-of-the-art.

**Weaknesses:**

Perhaps a weakness of the paper is that it relies on a solid understanding of information theoretic background; for e.g., [GK19]. More importantly, Theorem 5 which is main result on generalization bound for representation learning needs a bit more explanation, both in terms of intuitive/conceptual ideas and technical contribution based on work in the previous section. The proposed result – as the authors claim – resolves issues with the information bottleneck method, but the reader gets limited understanding of how the complexity of latent space and validity of any choice of decoder plays out in practical settings. On the other hand, the discussion in Section 2 could be more pointed – with focus on key novelties (which might help save space and leave room for more extensive discussion in Section 3). It also appears that a tabular comparison of generalization bounds proposed in this paper, relative to the ones based on mutual information would clarify some of the issues regarding positioning of the results.
Unless I am mistaken, it seems the authors do not formally introduce the key object $\mathbb{E}[gen(S,W)]$.


**Questions:**

1. Better explain the implications of Theorem 5 relative to generalization based on mutual information. Why *should* the encoder structure matter more than information in latent variable.

2. The connection/interpretation to/of geometric compressibility and how the lossy compressibility plays a role here is not very clear.

3. Is possible to have a concrete example where proposed bounds are shown to be superior to classical bounds based on mutual information?

**Limitations:**

not applicable.

---

> ### Author Rebuttal · Authors · 2023-08-09
>
> We thank the reviewer for their interest and their thoughtful comments that help to ameliorate the quality of the work.
>
> We will improve the readability of the paper by gently introducing the used information-theoretic concepts. Among others, by explaining how by adding the "block-coding" technique to the combinatorical approach of [BL03], we ended-up in having bounds containing the KL-divergence term. To this end, we will explain better the "Covering Lemma" in information theory, which will highlight the implication of KL-divergence terms, which are key elements in our results. This will also help to better explain the intuition behind IB (lines 45-49). We will also explain better the lossy compression and its implications (please see the answer to Question 2).
>
> We believe that the step-by-step structure of Section 2 helps the reader to understand the needed fundamental concepts and implications of our main result (Theorem 5). We will, however, try to make this section more pointed. We will also provide more high-level intuition about our proof techniques to highlight the key elements and implications of our results in Sections 2 \& 3; as sketched in "Regarding the presentation" part of our response to Reviewer FoJr.
>
> We apologize for missing the definition of $gen(S,W)$, that will be added. This definition existed in the earlier version and had been removed by mistake while reorganizing the paper.
>
> **Questions**
>
> 1.The generalization bound of [VPV18] has several shortcomings: it loses its relevance in a realistic setup, it cannot explain the geometrical compression, it does not capture the encoder's "simplicity"/"structure", and it also depends on the decoder's structure. Moreover, mutual information (MI) appears to be a bad indicator of generalization error [AG19, GK19, DKSV20]. Our results address these issues. As suggested by the reviewer, we will summarize the benefits of our bound in comparison to [VPV18]. In our answers to Questions 1-3, we briefly discuss some aspects of such benefits.
>
> The importance of the encoder's simplicity/structure can be well explained when the decoder is the identity mapping (no decoder) and hence the latent variables are the final predictions. Then, a common way to describe the simplicity is by studying the corresponding VC-dimension, which is known to be related to the generalization error. For VC-dimension, in contrast to MI, what matters is the structure of the whole vector $(X^n,\hat{Y}^n)$, rather than coordinate-wise relations that matter in MI. An example is given in the paragraph 234-259. When a decoder exists, a similar intuition holds. Loosely speaking, simple encoders generate more "structured" latent variables that consequently make more "structured" predictions. A closely related question is studied before. e.g.. by Bshouty and Mazzaw in "Exact Learning Composed Classes with a Small Number of Mistakes". Making the connection more specific is another interesting future direction to be mentioned.
>
> In principle, the encoder should guarantee good generalization behavior on its own and the role of the decoder is instead to minimize the risk. Our bound reflects this principle, by depending merely only on the encoder and being valid for any decoder.
>
> 2.Borrowed from the rate-distortion theory, lossy compression is a "natural solution" to measure the compressibility of continuous data. Indeed, to describe even a bounded continuous variable $X$ in a "lossless" way, one needs an infinite number of bits. However, if we allow for some "average distortions" in compression, the needed number of bits becomes finite and the i.i.d. instances of compressed versions of $X$ will be "concentrated" around $X$.
>
> Now, consider the geometric compression [GK19]: in this experimentally observed behavior, the latent variables $U\sim P_{U|X}$ of $X$ are distinct but concentrated around a few "centers". Thus, $I(U;X)$ becomes very large or infinite, causing the IB to lose its relevance. This happens for a similar reason as for the impossibility of the lossless compression of continuous sources. Hence, these scattered latent variables around the centers can be seen as a lossy version of the mappings from $X$ to one of these centers, with some small distortion. It should be noted that here there exists a subtle difference with the classical lossy compression, as we consider "lossy mapping" or "lossy algorithm compressibility" [SGRS22], instead of lossy compression. To give a simple example of "lossy mapping", suppose $X \in [0,1]$, and if $X<0.5$, then $U=-1+X/5$, and if $X>0.5$, then $U=1+X/5$. In this case while $I(U;X)=\infty$, a simple lossy mapping $P_{\hat{U}|X}$ with average distortion less than $0.05$ can be found such that $I(\hat{U};X)=1$. This explains the geometrical compression observation.
>
> We note this connection was a partial intuition behind the new algorithm MMCDVIB, as described in the responses to all reviewers.
>
> 3.Thank you for the suggestion. The MI-based bound of [VPV18] becomes infinite for continuous $(X,U)$. Even for discrete spaces, in a reasonable setup, the term containing MI in their bound is not dominant anymore, so the bound loses its relevance [RG19, LLS23]. In any of such a wide range of setups, our bound gives a better result. Moreover, the new experiments show a clear advantage of our KL-divergence regularization over MI. One reason is the data-dependent form of our priors in contrast to fixed priors in classical IB (as explained in our general response).
>
> That said, we agree it would be interesting to compare analytically our bounds with [VPV18], in a simple neural network (NN). However, this evaluation turns out to be a difficult problem even for the classical bound, i.e. to study analytically MI in NN. We will mention this direction, and also studying the geometries of the latent variables in different architectures (e.g., FCN vs CNN), as interesting future directions.

---

> ### Comment · Reviewer_D6jh · 2023-08-11
>
>
> Thank you for your response to my and other reviewers' comments and suggestions.

---

### Official Review · Reviewer_FoJr · 2023-07-06

**Soundness:** 4 excellent
**Presentation:** 3 good
**Contribution:** 3 good
**Rating:** 7
**Confidence:** 3

**Summary:**

**Post-rebuttal**

I thank the authors for their response and have raised my score to an accept.


Understanding the generalization properties of modern machine learning algorithms is one of the key challenges in statistical learning theory. Focusing on the classification setup, this paper first studies the generalization error in terms of the compressibility of the predicted labels using tools from rate distortion theory. The proposed bounds feature a new KL divergence term that takes into account the structure and "simplicity" of the algorithm. The authors then extend the framework to give a generalization bound for information bottleneck-type representation learning in terms of the compressibility of the latent variable. The authors also empirically demonstrate a practical benefit of their theoretical results for the information bottleneck.

**Strengths:**

In terms of originality, the generalization guarantee in terms of the compressibility of the representation that takes into account the complexity and structure of the encoder class appear novel. Also unlike existing mutual information-based bounds, the new bounds are not vacuous for deterministic algorithms. The work builds on prior work on PAC-MDL bounds from Blum and Langford (2003). The latter used a bits-back coding argument, while the tools developed in this paper rely on rate distortion theory and block coding arguments building on prior work [SGRS22] and [SZ23].

The authors also demonstrate a practical benefit of their theoretical results by comparing the performance of the vanilla VIB and a `"category-dependent" version, with the latter performing better in terms of generalization error and population risk.

Overall, the writing is clear though the presentation can be improved in certain parts.

**Weaknesses:**

The presentation in the main text can be improved in certain parts, e.g., giving more intuition for the proof techniques; see below. Also, in its current form, the paper seems to end abruptly. A separate Conclusions section summarizing the main contributions along with scope for future work / citing open problems can be very useful.

**Questions:**

* What is a high-level intuition for variable-size compressibility for the tail bounds? Likewise, for block covering does compressing multiple instances simultaneously allow for smaller distortion levels? It would definitely help the reader if some of the high-level proof ideas are laid out more clearly either in the main text or supplementary.

**Limitations:**

I do not foresee any potential negative societal impact of this work.

---

> ### Author Rebuttal · Authors · 2023-08-09
>
> We thank the reviewer for their interest in our work and their constructive comments. We will apply them to improve the quality of our work.
>
> **Regarding the presentation:** Thanks for all suggestions related to the presentation of our work. We will definitely take them into account for the next version of our paper. We will also focus on giving more intuition about the concepts and proof techniques. In particular, we will improve Section 2.1 and explain more clearly how starting from a combinatorical approach of [BL03], we established an information-theoretic framework using the "block-coding" technique. This technique, combined with lossy compression, can better explain the concept of "geometrical compression". Furthermore, we will expand the discussion about the connection between the block-coding technique and the Donsker-Varadhan's variation representation lemma that is used in the proofs. To this end, we will explain better also the "Covering Lemma" in information theory. This also helps to better explain the intuition behind IB (lines 45-49): Consider $n$ i.i.d. realizations $(X_i,U_i),i\in[n]$ of a discrete input $X \sim P_X$ and its corresponding discrete latent variable $U \sim P_{U|X}$. The covering lemma says for large enough $n$ there exists roughly $l_n\approx e^{nI(U;X)}$ vectors $\tilde{u}^n(j),j\in[l_n]$ such that with high probability for every $(X^n,U^n)$, $\exists j\in[l_n]$ such that the empirical distributions of $(X^n,U^n)$ and $(X^n,\tilde{u}^n(j))$ are close enough (and close to $P_X P_{U|X}$). Hence, loosely speaking, if block coding were allowed, an "equivalent" version of the latent variable could be described with roughly $I(U;X)$ nats.
>
> To give more intuition about the proof techniques, we will discuss more the intuition behind the choices of our "data-dependent" priors, which is new in this context. As an example, we will explain in Section 2 that in the simplest form inspired by [BL03], intuitively, we consider independent "shuffling" of $n$ pairs of $((Y_{i,1},\hat{Y_{i,1}}),(Y_{i,2},\hat{Y_{i,2}}))$ in a way that it becomes fully ambiguous whether any pair belongs to a training or test (ghost training) dataset. For this result, in fact, only a "partial" symmetry suffices, since considering a vector of $(Y^{n\times 2},\hat{Y}^{n,2})$, the elements $(Y_{i,1},\hat{Y_{i,1}})$ may be only swapped by $(Y_{i,2},\hat{Y_{i,2}})$. However, in Theorem 2, we consider shuffling the whole super-sample to assign $n$ instances to the training set and the remaining $n$ instances to the ghost training set. This requires a "complete symmetry" as described in lines 201-206. The idea of Section 2, however, is not directly applicable to Section 3, as the decoder part also depends on data, and hence by shuffling only pairs $((Y_{i,1},U_{i,1}),(Y_{i,2},U_{i,2}))$, $P_{W|S}$ and hence $P_{\hat{Y}|W_d,U}$ may change also. For this reason, we keep the training and test sets unchanged (hence, $P_{W|S}$ remains also unchanged) and only "shuffle" the assigned latent variables (distributed according to $\mathbf{Q}$) to a pair of test and data samples having the same label. This is the intuition behind the symmetry notion used in Section 3.
>
> We will add a conclusion to summarize the main questions addressed by our work and the future directions. We believe that our framework could be useful to study theoretically or experimentally different architectures of neural networks (e.g., two-layer FCN vs CNN) in order to understand how and why some of them can generate "better" representations. Another interesting direction would be to understand how to learn a "semi-optimal" lossy version of the latent variables and how to estimate the corresponding KL-divergence term using the recent advances in Rate-distortion estimation. Finally, while we performed some convincing experiments which partially showcase the practical implications of our results, we believe that our framework can be exploited further. For example, one may introduce more suitable priors which can better account for the "simplicity of the encoder" or the dependency of all latent variables $U^n$ (as induced by the encoder).
>
> **Regarding variable-size compressibility:**  Perhaps the main idea behind variable-size compressibility could be intuitively better explained by its contrast to the "fixed-size" compressibility: in the latter, one considers a fixed-size codebook (as defined in the paragraph starting from line 187) and then for "quantization" chooses any suitable vector $\hat{\mathbf{y}}[j]$. Technically, since in the proof we apply the union bound over the vectors of the codebook, the result of this approach would depend on the "fixed-size" length of the codebook (i.e., $\lim_{m\to \infty} \frac{1}{m}\log(l_m)$). In the high probability or tail bound (or, more precisely, in "data-dependent" tail bounds), in general, we allow the bound to depend on particular instances $S$, $S'$, and probably also on $W$. Hence, one cannot readily use fixed-size codebooks. To make this compatible, we consider a codebook with sufficiently large size. Then, depending on $(S,S',W)$, we only search for a suitable $\hat{\mathbf{y}}[j]$ only in the **first** $e^{R_{(S,S',W)}}$ elements of this codebook. For instance, the term $R_{(S,S',W)}$ in Proposition 1 is exactly the KL-divergence term appearing in the bound. We add these explanations.
>
> **Regarding block covering:** Thanks for pointing this out. Indeed, this is one of the main reasons why block covering is considered in lossy compression (rate-distortion) in information theory. In general, to guarantee the distortion level of "one-shot" covering to remain below a certain threshold, one usually needs to consider the "worst-case" scenarios. In contrast, the block-covering allows to consider the "average cost" which can be considerably smaller. We will add this discussion.

---

### Official Review · Reviewer_2eZy · 2023-07-08

**Soundness:** 3 good
**Presentation:** 2 fair
**Contribution:** 2 fair
**Rating:** 6
**Confidence:** 4

**Summary:**

This submission considers supervised learning problems and derives two types of generalization gap bounds. The first type states that generalization gap can be upper bounded by approximately $\sqrt{\mathrm{KL}(P || Q) / n}$, where $P$ is the average (wrt data and training stochasticity) of predicted label distribution on $n$ training examples and $n$ test examples conditioned on the labels of the $2n$ examples, while $Q$ is a fixed **symmetric** prior distribution for predicted labels on $2n$ examples given $2n$ labels. $Q$ is symmetric when $Q(\hat{y_1},\ldots,\hat{y_{2n}} | y_1,\ldots,y_{2n}) = Q(\hat{y_{\pi(1)}},\ldots,\hat{y_{\pi(2n)}} | y_{\pi(1)},\ldots,y_{\pi(2n)})$ for any permutation $\pi$. In a sense, the symmetricity means that $Q$ puts a distribution on $\hat{\mathcal{Y}}^{2n}$ given $y_1,\ldots,y_{2n}$ without "knowing" which $n$ of the $2n$ examples belong to the training set.

The second type of generalization bounds are designed for classifiers which have a representation part (an encoder) and a classification head. The main term in these type of bounds is also of the form $\sqrt{\mathrm{KL}(P || Q) / n}$, but where representations play the role of predicted labels. Again there are $n$ training, $n$ testing examples, and a symmetric prior $Q$ but now over representations. Notably, this type of bounds do not depend on the classification head, they only depend on the encoder.

For both type of generalization bounds, both in-expectation and in-probability bounds are derived. Furthermore, the authors derive tighter bounds where the mismatch between empirical and population risks is measured by a convex function $h:[0,1] \times [0,1] \rightarrow [0,2]$ instead of the $|x-y|$ function. Finally, to avoid KL divergence becoming infinite when working with continuous labels or continuous representations, the authors derive bounds where one seeks a "quantized" algorithm that has a similar generalization gap but non-infinite KL divergence.








**Strengths:**

**Strength #1: Relevance.** Understanding how some properties of learned representations affect generalization is an important and highly relevant direction for the NeurIPS community. The authors present a good summary of the criticism of information-bottleneck compression ideas in explaining generalization and propose to consider instead the joint compressibility of training and test representations.

**Strength #2: Originality.** To my best knowledge, the bounds that depend on representation complexity (results of Section 3) are novel. In some sense, these results suggest a new type of information bottleneck for representation learning.

**Weaker strength: Soundness.** The results of this submission are sound as far as I checked. However, in some places I couldn't follow the proofs very carefully as some steps were not detailed enough or the notation was confusing (please see my comments below).


**Weaknesses:**

**Weakness #1: Hard to judge the significance.** While the overall direction of obtaining generalization guarantees via joint compressibility is interesting and this submission makes a good step towards exploring that direction, it is hard to judge how significant are the main results of Section 3.
- [Estimation] I don't see a good way of estimating the bound of Theorem 5. In the case of continuous representations, how can one take an infimum over learning algorithms that have similar generalization gap?
- [Experiments] The experiments do not help to judge whether the KL term in the bound can be successfully used as a regularization.
	* [E1] Line 608, "As in the VIB approach, inevitably, we only restrict the optimization to the training set.": This seems to go against the spirit of the approach employed throughout the paper. Namely, as I understand the symmetricity of $\boldsymbol{Q}$ can be interpreted as $\boldsymbol{Q}$ not being aware of the train-test set split. Forcing the encoder to be close to such a distribution $\boldsymbol{Q}$ forces the encoder to behave similarly for seen and unseen examples. From this one gets a generalization guarantee.
	- [E2] Even if one splits an original training set into a new training set and a ghost set to follow the logic of the main text, the KL regularization term will probably be fitted by "cheating", in the sense that representations of the new training set and ghost set examples will be different from those of unseen examples. The nature of ghost examples (to be not used during the training) makes it impossible to use the KL divergence term of Theorem 5 as a regularization.
	* [E3] Lines 612-617: $\boldsymbol{Q}$ will not be symmetric in this construction, as it is constructed using statistics of *training data*. Consequently, the KL divergence can become optimistically small (when training examples are "memorized").
	- [E4] Lines 667-669, "It can be observed that our scheme outperforms VIB in terms of both generalization error and population risk, for all values of the Lagrange multiplier $\beta$": Since the two objectives are different one should not compare then for a fixed $\beta$. Instead, we can consider tuning $\beta$ for each of them independently to minimize test error. We see that in that case there is no significant difference between the two approaches. Similarly, there is no significant difference if we compare generalization gaps for a fixed level of small test error.
- [Applications] The submission would benefit significantly from an analytical evaluation/bounding of the bound of Theorem 5 in some tractable settings (maybe a two-layer neural network on Gaussian data). This would allow to see if the bound is powerful enough to produce meaningful results in some settings.

**Weakness #2: Clarity & Presentation.** This submission would benefit from improvements in terms of clarity and presentation. It was hard for me to read and understand the details. There are many small mistakes, the notation is cumbersome, and some parts of derivations are not detailed enough. Please see more detailed comments and suggestions below.

**Weakness #3: Missing important related work.** This submission does not cite well the literature of information-theoretic generalization bounds initiated by the works Russo and Zou [1] and Xu and Raginsky [2], and developed further by many others. This line of work is relevant for multiple reasons.
1. The general technique of deriving bounds in this work and in the mentioned line of work is the same (using Donsker-Varadhan inequality and bounding a cumulant-generating function).
2. The general idea of measuring how different a learned function acts on training and testing examples has been used in many follow-ups on [3].
3. The idea of symmetric priors appears in [4] under the name "almost exchangeable data-dependent priors". In their bound there is also a term of the form $\mathrm{KL}(P||Q)$, where $P$ is the output distribution (over hypotheses) after training on $n$ training examples, while $Q$ is a symmetric prior over hypotheses given $2n$ examples. **Important:** when considering in-expectation bounds, the optimal $Q$ is the average of output distributions over all possible $2^n$ training/test splits. The KL divergence for this optimal prior becomes the conditional mutual information (CMI) $I(W; J | \tilde{Z}_0, \tilde{Z}_1)$, where $J \sim \mathrm{Uniform}\\{0,1\\}^n$ is the training-test split, $W$ is the output of the learning algorithm when splitting with $J$, and $\tilde{Z}_0$ and $\tilde{Z}_1$ are the $n$ training and $n$ testing examples respectively.
4. While in works [2], [3], and [4] the bounds depend on some kind of information captured by the output of a learning algorithm (i.e., the output hypothesis), in [5] there are bounds that measure information in *predicted labels* in the random subsample setting of Steinke and Zakynthinou [3]. Given the important part of the last point, the result of Theorem 1 can be seen as an f-CMI bound written differently. The only minor difference is that in f-CMI bounds there is conditioning on $\tilde{Z}_0$ and $\tilde{Z}_1$ instead of $\tilde{Y}_0$ and $\tilde{Y}_1$ (i.e., only the label parts), but it is easy to see that the proof does not use the input parts ($\tilde{X}_0$ and $\tilde{X}_1$).
5. It is also worth to mention that CMI (or f-CMI or e-CMI) bounds are usually improved by measuring information with individual pairs separately (variations of the technique of Bu et al. [6]), and by conditioning on an individual pair of train/test examples (e.g., [7]). These two techniques together produce f-CMI bounds of form $\sum_{i=1}^n \sqrt{I(\hat{Y_{0,i}}, \hat{Y_{1,i}}; J_i | Y_{0,i}, Y_{1,i})}$ that can be much tighter than the bound of Theorem 1 (the equivalent of the f-CMI bound when measuring information in all predictions at once).


**References**

[1] D. Russo and J. Zou. How much does your data exploration overfit? controlling bias via information usage. IEEE Transactions on Information Theory, 2019.

[2] A. Xu and M. Raginsky. Information-theoretic analysis of generalization capability of learning algorithms. NeurIPS 2017.

[3] T. Steinke and L. Zakynthinou. Reasoning About Generalization via Conditional Mutual Information. COLT 2020.

[4] P. Grunwald, T. Steinke, L. Zakynthinou. Pac-bayes, Mac-bayes and conditional mutual information: Fast rate bounds that handle general VC classes. COLT 2021.

[5] H. Harutyunyan, M. Raginsky, GV. Steeg, and A. Galstyan. Information-theoretic generalization bounds for black-box learning algorithms. NeurIPS 2021.

[6] Y. Bu, S. Zou, and V. V. Veeravalli. Tightening mutual information-based bounds on generalization error. IEEE Journal on Selected Areas in Information Theory, 2020.

[7] B. Rodríguez-Gálvez, G. Bassi, R. Thobaben, and M. Skoglund. On random subset generalization error bounds and the stochastic gradient Langevin dynamics algorithm. IEEE ITW 2021.

**Questions:**

1. Lines 45-48: What exact result is being referred here? The mutual information between two variables cannot be seen directly as some sort of description length (just consider two variables with 0 mutual information).
2. Section 2.1 Compressibility framework: The concepts introduced in this section are not used in the main text (specifically the block quantities). I recommend moving this subsection to Appendix C. This way it becomes easier to read the main text and the already complicated notation does not become more complicated. The only idea that I recommend keeping from this section is that if one can encode $\hat{Y}^{2n}$ with short code on average, then one should expect good generalization. Similarly, the sentence of lines 209-211 should be moved too.
3. Theorem 1: $\text{gen}(S,W)$ should be defined.
4. Equation (3): The training algorithm can act differently on training examples with different indices. It is not necessarily true that the RHS of eq. (3) can be written as $n$ times the quantity for a single example.
5. Theorem 2: $\mathcal{L}(W)$ and $\hat{\mathcal{L}}(S,W)$ should be defined.
6. Line 295: $\boldsymbol{Q}$ instead of $P$ for the prior.
7. Theorem 4: Why is the idea of a posterior $Q$ introduced here, can't it be just $P_{W|S}$ and $P_{\hat{W}|S}$? As it has been done before this point, the letter $Q$ would be good to use only for priors. In the equation after line 311, it should be $\hat{W} \sim P_{\hat{W}|S}$ instead of $\hat{W} \sim Q$. Also, in eq. (6) it would be good to write it as a difference of two expectations. In that case it will be clear what distributions $W$ and $\hat{W}$ follow.
8. Line 802: It should be $\boldsymbol{Q}$ (the symmetric prior) instead of $Q$ (the posterior over $W$). Similarly in all $\Delta(S,Q)$ terms. Also $\Delta(S,Q)$ should be $\Delta(S,S',Q)$ as it depends on $S'$ too.
9. Line 350: Why is the property of preserving labeling important for the permutation $\pi$ in the definition of a symmetric prior? I recommend adding more details of the transition at line 839 where the symmetricity of $\boldsymbol{Q}$ is used.
10. Line 831-832, "it can be easily verified that this distribution is symmetric": Which distribution is referred here?
11. Line 833: It should be $k : [n] \rightarrow [n]$.
12. Line 834: $||\hat{p_s} - \hat{p_{s'}}||_1 \in [0, 2]$, so $T$ can become negative. It should be divided by 2.
13. Lines 833-834: The construction of the permutation $k$ is not always possible. It might not be possible to map the *first* $T$ samples in $Y$ to another $T$ equal samples in $Y'$.
14. The technique of taking an infimum over close enough "quantized" algorithms is a step that can be applied no matter whether the bound depends on joint compressibility or $\hat{I}(X; U)$ or something else. Therefore, I feel that one should not claim advantage over other results in terms of being applicable also to continuous variables when the advantage is a result of applying this general technique (not due to something inherent to the approach).
15. It would be good to have a small discussion on the fact that representation learning has many goals (e.g., learning representations that are "useful" for many tasks). Learning representations that guarantee small generalization gap for a *given* task is not of high importance in common representation learning settings. I don't think that one would be happy to find out that after a heavy pretraining round with a pretraining task $Y$ the network learned a representation $R=\phi(X)$ such that $R=g(Y)$ for some function $g$.


**Limitations:**

The limitations are addressed adequately.

---

> ### Author Rebuttal · Authors · 2023-08-09
>
> We thank the reviewer for their detailed investigation of the paper, their interest, and their constructive comments.
>
> **Weakness #1:** As mentioned by the reviewer, our work is the first theoretical work that characterizes $gen(S,W)$ in terms of the complexity of latent variables, achieved by considerably extending the framework of [BL03] to characterize the MDL of predictions/latent variables. In Sections 2 and 3, we address "conceptually" several existing challenges and criticisms about measuring the MDL using mutual information, in particular by implicitly taking into account the encoder's simplicity and the geometric compression. Furthermore, we demonstrated some practical benefits of our result and the role of "data-dependent" priors, which we believe can be exploited further in future works.
>
> - Estimation: Firstly, the bound in Theorem 5 holds for **any** choice of $P_{\hat{W}_e|S}$ satisfying (7) and taking the infimum to achieve a "valid bound" is not needed. Thus, any simple technique that adds noise/quantization to parameters or models can be considered, as described in lines 381-388, which is especially useful for deterministic encoders. Besides these "blind" approaches, an interesting direction could be to leverage a combination of Mine (arXiv:1801.04062) and "SFRL" (arXiv:1801.04062), similar to "Rate-Distortion via Constrained Estimated Mutual Information Minimization" (ISIT2023), to intuitively "learn" a "semi-optimal" choice of "noise-injection" using another NN. Nonetheless, these approaches would need to be (non-trivially) adapted to our setup. We add these discussions as future directions.
>
> - Experiments: Regarding [E1-E3], we do agree that to "correctly" use the KL-divergence term of Theorem 5 as a regularizer and to satisfy the symmetry condition of the prior, one needs a "fresh" test mini-batch in each iteration, which is impractical. To partially address this concern, we made several changes in our experiments and by considering the best choice of $\beta$ for each algorithm ([E4]), indeed now we observe a better improvement of about 3\% upon VIB. Please refer to the response to all reviewers for the changes: moving-average and multiple (per label) priors and the CNN choice.
>
> - Application: We agree that analytically evaluating the bound for a two-layer NN will be a significant complement to our work, which in our opinion merits a separate publication. As for the current state, we believe, we made significant theoretical and experimental contributions that show the importance of our work.
>
> **Weakness #2:** Thanks for the provided feedback in the Questions section, addressed below. We will proofread the paper and the calculations carefully.
>
> **Weakness #3:** Our motivation was to understand the "complexity" of the predictions/latent variables and their connection to $gen(S,W)$, as "assumed" in IB. This is conceptually and practically different than results in terms of complexity of $W$. To study this, we built a framework based on the combinatorical approach of [BL03]. The proofs using Donsker-Varadhan's lemma appeared later as an abstraction of our initial proofs.
>
> We missed the connection with CMI-based works as the original CMI-based work was mainly focused on the complexity of $W$. We sincerely thank the reviewer for bringing our attention to them, in particular the f-CMI-based follow-ups. We will make the connection between the CMI-based works (and IT-based bounds and their proof techniques) and ours, which also highlights the connection between [BL03] and the CMI bounds, which had not been elaborated in CMI-based works, AFAWK. For further discussions, please see the parts **Regarding CMI** and **Regarding almost exchangeable priors** in Reviewer Lcin's response.
>
> **Questions**
>
> 1.Covering lemma in a sense explained in "Regarding the presentation" part in Reviewer FoJr's answer. This will be clarified.
>
> 2.This section is provided to give an intuition about *compressibility* nature of MDL and our results and also how IT terms arise by combining [BL03] and block-coding. However, we agree it may be hard to follow. We will rephrase/reduce this section and keep only the needed intuitions in the main text.
>
> 3, 5,6, 8, 11. Thanks, will be added/corrected.
>
> 4.Per our understanding, the question is about the first term in (3) which is related to $S$. If so, we agree and the text already reflects this by considering $X\sim \hat{\mu}_{X|S}$ in (3) and by denoting empirical MI in line 218 as $\hat{I}(X;\hat{Y})$.
>
> 7.Thanks. Will be adapted and corrected. Using another distribution is to emphasize that the expectation is not needed to necessarily be taken w.r.t. $P_{W|S}$ induced by learning algorithm; as used e.g. in arXiv:1707.09564.
>
> 9.Thanks, details of the proof will be added. For the intuition, please refer to "Regarding the presentation" part in Reviewer FoJr's answer. Just please note a typo at line 350 and its correct version at line 820.
>
> 10.We meant $P_{\mathbf{Q}}$ in (31). Will be rephrased.
>
> 12.Thanks. Indeed $T=n-\frac{n}{2}\|\hat{p_s}-\hat{p_{s'}}\|$, which slightly improves the bound.
>
> 13.Thanks for such a precise verification! We implicitly assumed this holds w.l.o.g. since the proof does not rely on the position of these indices, but to be more precise, we will consider another permutation $f\colon [n] \to [n]$ s.t. $Y_{f_i}=Y_{k_i}^{'}$, $i\in[T]$ (for simplicity, the dependence of $f$ on vectors $(Y^n,Y^{'n})$ are dropped in the notation.). The rest of the proof follows similarly as before by replacing $Y_i$ by $Y_{f_i}$ and $\hat{Y_{2,i}}$ by $\hat{Y_{2,f_i}}$.
>
> 14.We agree and this will be precised to avoid confusion. As mentioned, the idea comes from the RD theory that is indeed applied to MI. Here, it is leveraged in particular to explain the experimentally-observed "geometrical compression" concept that was "perceived" to be different from IT compression.
>
> 15.Thanks. We add the discussion that we believe can be partially addressed by our work.

---

> > ### Comment · Reviewer_2eZy · 2023-08-16
> > **Reviewer response**
> >
> > Thank you for the detailed response. The inclusion of a discussion on the overlooked related work and the refinements of presentation improve this submission. I have adjusted the score accordingly.

---

### Author Rebuttal · Authors · 2023-08-09

We sincerely thank all the reviewers for their valuable time and their pertinent comments that helped to improve our work. We are excited to see that all reviewers appreciated the value of our work. We carefully read the comments of all reviewers and addressed them separately. Please do let us know if any further clarification is needed.

Here, we present our new experimental methods and findings that further reveal the potential practical implications of our work.

**New experimental results.**  To improve the experimental results, we introduce two enhanced variations of the previously described CDVIB approach: a Moving-average CDVIB (MCDVIB) and Multiple-centers Moving-average CDVIB (MMCDVIB).

**MCDVIB:** The first approach (MCDVIB) is similar to CDVIB with the only difference that the mean and covariance matrix of the priors $Q_K$ do not depend solely on the data in a single mini-batch. Instead, we consider a *moving-average* of such quantities over mini-batch sequences. Please note that although MCDVIB is just a special case of MMCDVIB (see below) with $M=1$, we presented them separately to better illustrate the effect of each enhancement.

By taking the moving average of mean and covariance, the data in the previous batches plays "partially" the role of the "ghost data" (that should be from the test set and is not available). This synthetic reproduction of the effect of ghost data is more pronounced during the first epochs where the neural network had not yet "memorized" the whole data. Note that the learning rates at the first epochs are larger and in general the early epochs of the training are important and sometimes "decisive" during which the regularization has significant importance, as shown in arXiv:2002.09572, arXiv:1609.04836, arXiv:2002.05366, arXiv:1711.08856. As can be observed in the attached PDF file, indeed this helps to increase the accuracy.

**MMCDVIB:** Now, we consider a prior choice that allows *multiple* priors for each label. For each label $k\in[K]$, we consider $M\in \mathbb{N}$ priors

$$Q\_{k,r}^{(t)}=\mathcal{N}\left(\bar{\mu}\_{k,r}^{(t)},diag\left(\bar{\sigma}^{(t)^2}\_{k,r}\right)\right),\quad r\in[M], k\in [K],$$

defined over $\mathbb{R}^m$, where the superscript $t\in\mathbb{N}$ represents the optimization iteration. The vectors $\bar{\mu}^{(t)}\_{k,r}$ and $\bar{\sigma}^{(t)^2}\_{k,r}$ are initialized as $m$-dimensional zero vector and updated at each iteration as follows. First, we introduce two notations. For any sample $z$ with label $k$ and for any $t\geq 1$, denote
$$r\_z^{(t)} = argmin\_{r\in[M]} \| \mu\_{x}-\bar{\mu}\_{k,r}^{(t-1)} \|.$$

Suppose that the picked mini-batch at iteration $t$ is $\mathcal{B}\_t=\{z_1,\ldots,z_b\}$. For each $k\in[K]$ and $r\in[M]$, let
$$\mathcal{I}\_{k,r}^{(t)}=\\{ z_i: i\in[b],y\_i=k, r\_{z\_i}^{(t)}=r \\},$$
and let $b\_{k,r}=|\mathcal{I}\_{k,r}|$. Now, if $b\_{k,r}\neq 0$, update $\bar{\mu}^{(t)}\_{k,r}$ and ${\bar{\sigma}^{(t)}\_{k,r}}$ as
$$\bar{\mu}\_{k,r}^{(t)} \coloneqq  (1-\alpha b\_{k,r})\bar{\mu}\_{k,r}^{(t-1)} + \alpha \sum_{z_i\in \mathcal{I}\_{k,r}} \mu\_{x_i},$$
$$\bar{\sigma}^{(t)}\_{k,r,j} \coloneqq \sqrt{(1-\alpha b\_{k,r})  \bar{\sigma}^{(t-1)^2}\_{k,r,j}+ \alpha\sum\_{z\_i\in \mathcal{I}\_{k,r}} \sigma_{x_i,j}^2},\quad j=1,\ldots,m.$$
where $\alpha \in [0,1]$ is a coefficient smoothening the evolution of the mean and variance, and also implicitly takes into account the effect of the ghost dataset $S'$ appearing in the bounds of Theorems 5 and 7. The optimal value of $\alpha$, among others, depends on $b$ and $M$.
Finally, the considered objective function at iteration $t$ is
$$\frac{1}{b} \sum\_{i=1}^b \\bigg\\{  \beta  D\_{KL}\bigg[\mathcal{N}\left(\mu\_{x_i},diag(\sigma\_{x\_i}^2)\right) \\| \mathcal{N}\left(\bar{\mu}^{(t)}\_{y_i,r_{z_i}^{(t)}},diag(\bar{\sigma}\_{y_i,r_{z_i}^{(t)}}^{(t)^2})\right)\bigg]-\mathbb{E}\_{U_i\sim P\_{U|X,W_e}(U_i|x_i,W_e)}\left[\log P_{\hat{Y}|U,W_d}(y_i|U_i,W_d)\right] \\bigg\\}.$$

**Simulation setup:** For the simulations, we consider the CIFAR10 dataset and a two-step prediction model, where the encoder consists of 4 conv. and 2 fully-connected layers (CONV4+FC2) and the decoder is a single FC layer. This choice is motivated by the fact that in general, CNN are better adapted to process image data. We set mini-batch size $b=128$, the moving average coefficients $\alpha = 5e-4$ in (M)MCDVIB, and the number of centers in MMCDVIB $M=5$. We trained the model for 200 epochs.

**Simulation findings:** As can be observed in the attached PDF, both MCDVIB and MMCDVIB help to improve the performance of the learning algorithm. The latter improves the accuracy by around 3\% upon VIB.

We have further plotted the 2-dimensional random projection of the latent space for both training and testing data for MMCDVIB, and for $\beta = 0.001, 0.01, 0.1$ (for these values of $\beta$, the advantage of MMCDVB over VIB is the most pronounced, as seen in Fig. 1a). It could be noticed that the distribution of testing samples is closely aligned with the distribution of training samples. This suggests in turn that the trained models did not simply "memorize" the training data, and instead, they learned to generalize. Furthermore, although it is hard to judge on this 2D projection whether the produced learned samples are far apart, we may notice that when $\beta=0.1$ the latent space (somewhat intuitively) seems to be more "structured" and compact than when $\beta=0.001$.

---

### Decision · Program_Chairs · 2023-09-21

**Decision:**

Accept (poster)

**Comment:**

The paper shows new generalization bounds based on information-theoretic quantities and techniques. The contribution builds on the classical work of Blum and Langford for PAC-MDL bounds; the bounds are phrased in terms of a KL divergence with symmetric priors, and leverage a rate-distortion theoretic perspective. The work also has connections to ideas in information bottleneck, and to some degree formalize the benefit of "bottlenecked" representations, in a sense the authors suitably formalize.

The reviewers universally agreed the paper is written well, places the bounds in good context given prior work, and provides interesting insights.